

# Spruce bark beetle (*Ips typographus*) infestation cause up to 700 times higher bark BVOC emission rates from Norway spruce (*Picea abies*)

Erica Jaakkola[1], Antje Gärtner[1], Anna Maria Jönsson[1], Karl Ljung[2] Per-Ola Olsson[1], Thomas Holst[1]

[1]Department of Physical Geography and Ecosystem Science, Lund University, Lund, 223 62, Sweden
[2]Department of Geology, Lund University, Lund, 223 62, Sweden

*Correspondence to*: Erica Jaakkola (erica.jaakkola@nateko.lu.se)

**Abstract**

Emissions of biogenic volatile organic compound (BVOC) from the bark of Norway spruce (*Picea abies*) trees can be affected by stress, such as infestation of spruce bark beetles (*Ips typographus*). We studied the difference in emission rates from healthy spruce bark and infested spruce bark, the influence of time since spruce bark beetle infestation started and the difference in emission rates from bark beetle drilled entry holes and exit holes. Bark chamber measurements on both healthy trees and infested trees were performed during the summer of 2019 at two sites in Sweden. To consider the seasonal pattern of the spruce bark beetle, we divided the emission rates from infested trees into two seasons, an early season dominated by entry holes and a late season with mainly exit holes. Our findings show a significant difference in emission rates from healthy and infested trees, independent of season. The seasonal average standardized emission rate from healthy trees was $31.89 \pm 51.67$ µg m$^{-2}$ h$^{-1}$ (mean ± standard deviation), while the average standardized emission rates from infested trees were 6385 µg m$^{-2}$ h$^{-1}$ and 2102 µg m$^{-2}$ h$^{-1}$ during early and late season respectively. We also found an exponentially decreasing relationship with BVOC emission rates and time since infestation started where the emission rates reached the same level as the constitutive BVOC emission rates from bark after around one year. When comparing bark monoterpene BVOC emission rates with emission rates from needles, we found that the constitutive needle emission rates were 11 times higher than the constitutive bark emissions. However, the emission rates from infested Norway spruce tree bark were instead 6 to 20 times higher than the constitutive needle emissions, causing substantial increases in the total tree BVOC emission rate (550 % to 1900 % increase). This study adds evidence that spruce bark beetle induced bark BVOC emissions are higher than previously thought and highlights the need for further research with more samples more frequently throughout the season to fully understand the impact, which is required to quantify spruce bark beetle infestations impacts on the atmospheric chemistry and climate change.

## 1 Introduction

In Europe, forest damage caused by outbreaks of the European spruce bark beetle (*Ips typographus*) is the third largest disturbance after storm-felling and forest fires (Jönsson et al., 2012; Schelhaas et al., 2003). In Sweden, the drought in the summer of 2018 led to increased bark beetle outbreaks which, in 2020, were estimated to affect about 8 million m$^3$ (standing timber volume) Norway spruce (*Picea abies*) forest (Wulff and Roberge, 2020). This is the largest stock of forest volume killed by spruce bark beetles recorded in a single year in Sweden; in the period of 1990-2010, around 150 000 m$^3$ forest in southern Sweden was damaged on average per year (Wulff and Roberge, 2020). Climate change amplifies the risk of bark beetle outbreaks as the elevated risk of storm felling



and drought favor the bark beetles with easier access to weakened trees (Jönsson et al., 2012). Higher temperatures

and a longer growing season can also lead to an additional generation of spruce bark beetles per year (Jakoby et al., 2019; Jönsson et al., 2012). A larger bark beetle population, triggered by weather extremes, is associated with an increased risk of attacks on healthy spruce trees, with outbreaks leading to extensive damage to the forests (Jakoby et al., 2019; Seidl et al., 2014).

Biogenic volatile organic compounds (BVOCs) emitted from trees function as a defense system against heat and oxidative stress (Loreto and Schnitzler, 2010). They are highly volatile and chemically reactive and can react directly with oxidizing species or act as membrane stabilizers (Brilli et al., 2009; Kleist et al., 2012; Sharkey et al., 2001). The efficiency of BVOCs to form oxidation products depends on the specific BVOC's molecular structure (Bonn and Moortgat, 2002; Roldin et al., 2019; Thomsen et al., 2021). Stress-induced BVOC emissions alter the

oxidation capacity as some BVOC species are more efficient to act as secondary organic aerosol (SOA) precursors and foster particle growth (Roldin et al., 2019; Thomsen et al., 2021). Boreal forests experiencing abiotic or biotic stress due to large-scale forest disturbances might increase the production of BVOC species highly efficient as precursors of SOA to the atmosphere. As BVOCs are emitted, they can enhance chemical reactions that in turn can lead to increased tropospheric ozone concentration or be oxidized and foster the formation of SOA. This results

in high uncertainties regarding the contribution to either a negative feedback loop (cloud formation and radiation scattering (Paasonen et al., 2013)) or a positive feedback loop (increased tropospheric ozone) for the climate (Arneth et al., 2010; Jia et al., 2019). The impact of aerosol formation including the BVOC–SOA feedback still remains the largest uncertainty in our understanding of their radiative forcing (IPCC, 2014; Jia et al., 2019). Emission of BVOCs due to plant stress further increase the uncertainties of their impact in a changing climate.


Increased BVOC concentrations in plant tissue can also fight off predators (Laothawornkitkul et al., 2009; Li et al., 2019; Rieksta et al., 2020), and conifer trees use BVOCs as a defense mechanism against spruce bark beetles (Raffa and Berryman, 1982). The parental bark beetles attack spruce trees by drilling entry holes into the bark and form eggs galleries in the phloem. After about eight weeks, the new generation starts to leave the tree by boring

exit holes (Öhrn et al., 2014). To prevent a successful attack, the spruce increase the resin flow which submerge the parental bark beetles and egg galleries, potentially killing beetles or pushing them out of the entry hole (Raffa, 1991). The resin serves as a storage pool containing BVOCs that volatilize when the resin is flowing out of the tree, making the resin harden and close the wound. BVOCs are emitted constitutively from the trunk of the spruce, but when the emissions are induced as a stress-response, they have been shown to be toxic to spruce bark beetles,

especially certain compounds like myrcene and α-terpinene (Everaerts et al., 1988). Studies on conifers attacked by bark beetles found evidence of increased monoterpene (MT) content at the attacked location (Amin et al., 2013; Eller et al., 2013; Ghimire et al., 2016; Zhao et al., 2011) and occurrence of the oxygenated MT eucalyptol has been found to indicate induced defense and higher survival rates from Norway spruce attacked by spruce bark beetles (Schiebe et al., 2012). Comparing BVOC emission sources from different parts on conifer trees, trunk

emissions are suggested to potentially contribute a lot more to the whole tree emissions than previously thought, even when not attacked by bark beetles (Greenberg et al., 2012).

The defense mechanism of Norway spruce is poorly understood, as few studies have analyzed the induced BVOC emission from the trunk following an attack of the European spruce bark beetle (Ghimire et al., 2016; Zhao et al.,

2011). The aim of this study was to study the BVOC emissions from Norway spruce trunks and the impact of spruce bark beetles and to investigate the connections of BVOC emission rate, number of bark beetle holes and





time. The aim was also to put our study in a broader perspective by connecting the spruce bark beetle induced BVOC emission rates to needle emissions and other stresses like heat. Based on previous findings we formulated three hypotheses. Our first hypothesis was that infested trees have higher bark emission rates than healthy trees,

and that infestation changes the emission blend. Our second hypothesis was that BVOC emission rates would be highest at the start of infestation, and decrease over time in response to declining vitality and eventual death of the tree. We also wanted to investigate the connections of emission rates and number and type of bark beetle holes, where our third hypothesis was that there is a relationship between the number of entry holes and emission rate rather than the total number of holes. We reasoned that after a successful infestation, the number of holes increase

after some weeks when the new generation leave the tree through exit holes, but the emission rates would decrease as the tree is dying due to bark damage and blue stain fungi preventing transport of water and nutrients. In a sub-study, tree individuals with different initial health status were selected and followed throughout the growing season by repeating the measurements during the successful attack and infestation of bark beetles. From this sub-study we were hoping to see if the difference in initial health status of the trees would result in different emission rates

and emission blends, and also analyze how the individual emission blend changed over time after a successful infestation.

## 2 Methods

### 2.1 Site description

Five measurement campaigns were carried out from May to August 2019 at the ICOS (Integrated Carbon

Observation System, ICOS-Sweden.se) research station Hyltemossa (HTM, 56°06′N, 13°25′E; Fig. 1b) and one additional campaign at the ICOS research station in Norunda (NOR, 60°05′N, 17°29′E; Fig. 1b). The forest in HTM is dominated (>97% of the species composition) by Norway spruce (*Picea abies*) with a small fraction (<3%) of Scots pine (*Pinus sylvestris*) and deciduous trees. The understory vegetation is sparse, containing mostly mosses (Heliasz et al., n.d.). The forest in NOR is dominated by Norway spruce (54%) and Scots pine (37%) with a small

fraction (9%) of deciduous trees and an understory vegetation with shrubs of mostly blueberries, cranberries, mosses and flowers (Mölder et al., n.d.). Both facilities are located inside managed forests, but the age and height of the trees differ. In HTM the trees are around 40 years old with an average height of 19 m and NOR has a forest stand of mixed ages around 60-80 years and up to 110 years with a height of around 25 m for the dominating trees in 2019 (Heliasz et al., 2021; Mölder et al., 2021).

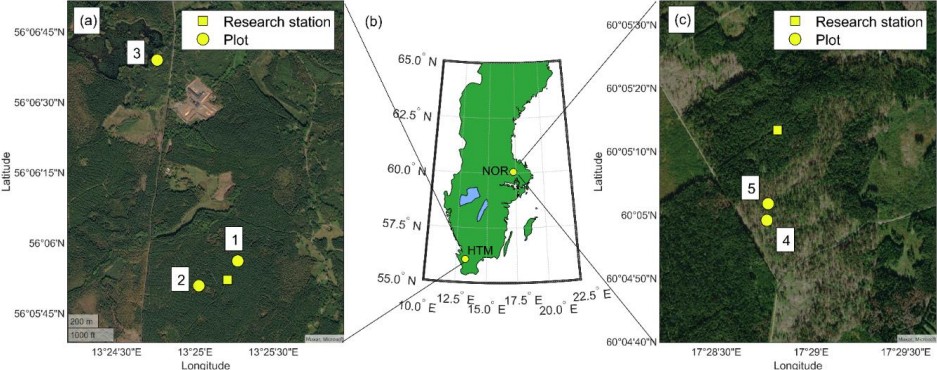


**Figure 1: The location of the study sites in Sweden (b), with Hyltemossa (HTM; a) displayed to the right and Norunda (NOR, c) displayed to the left. Measurement plots at HTM (1-3) and NOR (4-5) are shown in the site-specific maps and**



**their location relative to the ICOS station. The Figure is created in MATLAB and Mapping Toolbox release 2021a (The MathWorks, Inc., Natick, MA, USA).**


Three plots in HTM were selected for the study (Fig. 1a), and two plots in NOR (Fig. 1c). Two plots in HTM were located inside the Norway spruce plantation used by ICOS, while the third plot was located around 1.6 km north of the ICOS station in an older (about 100 years) forest stand. In NOR the locations were chosen based on availability of bark beetle-affected trees inside the forest plantation. Four Norway spruce trees were selected at

each plot in HTM, and three trees at each plot in NOR. A total of 18 trees were measured, whereof 12 were measured repeatedly during the growing season in HTM. For all plots in HTM healthy trees were selected where the tree health was determined by visual examination in close contact with the forest manager at Gustafsborgs Säteri AB, in May 2019. In addition, two trees seemingly stressed from late bark beetle infestation in 2018 were chosen, one at plot 1 only measured in the early season and one at plot 3 which later got infested again and used

for the comparison of healthy and stressed trees when infested. Only infested trees were selected in NOR, as bark beetle outbreaks were occurring in NOR during 2019.

To enable the measurements of healthy trees which later got infested, a tree at plot 3 in HTM (Fig. 1a) was baited using a bark beetle slit trap with pheromones. The trap was installed between the $1^{st}$ and $2^{nd}$ campaign. The

pheromones used in the trap were a combination of 2,3,2 Methylbutenol, cis-Verbenol and Ipsdienol (Typosan IPS), and one standard bag was inserted in the slit trap. Two trees were successfully infested and measured repeatedly to track the emission pattern after infestation.

The weather during the measurement periods varied from cold and humid to warm and dry conditions. The average

temperature during the growing season (May-August) was 14.6 °C (± 4.6 °C) in HTM and 14.2 °C (± 4.2 °C) in NOR with the sum of the precipitation over the growing season being 168 mm in HTM and 151 mm in NOR. The daily average temperatures during the measurement periods ranged from 5 °C to 22 °C for both sites, with a daily total rainfall up to 3 mm (Fig. 2). In HTM, it was warmest (12-28 °C) during measurement campaign 2 in June and coldest (1-11 °C) during the $1^{st}$ campaign in May.






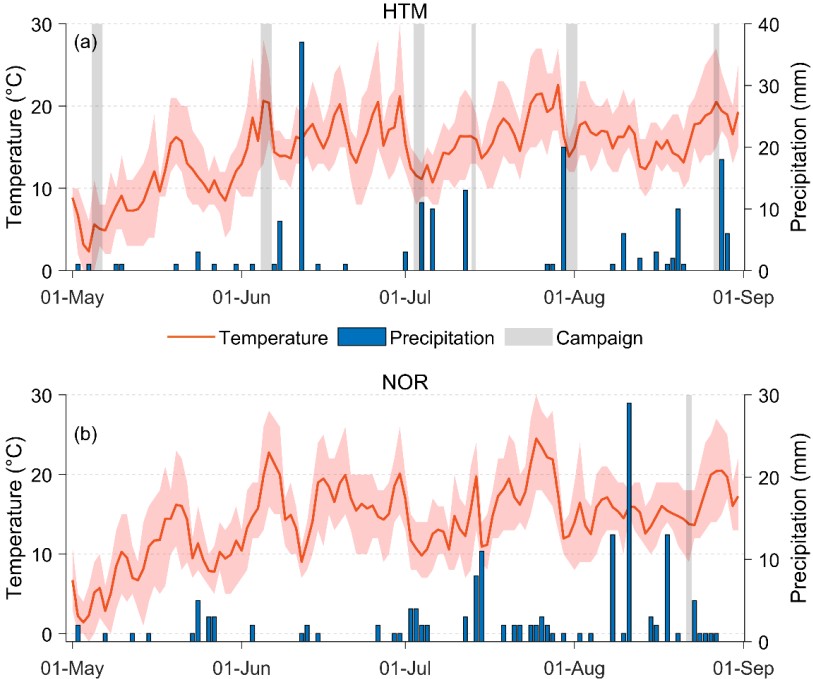

**Figure 2: The total daily precipitation (blue bars) and daily average temperature (red line) with daily minimum and maximum temperature (red shade) for the study sites in (a) Hyltemossa and (b) Norunda. The times for the campaigns are marked in grey. Data: ICOS (Heliasz, 2020; Mölder, 2021).**

**2.2 Experimental design**

The bark emissions from the trees were measured using a tree trunk chamber connected with PTFE tubing (Teflon, Swagelok, Solon, OH, USA) to a pump box system to provide purge air with a flowrate of 0.7 lpm (liters per minute) to the trunk chamber with a volume of 0.6-0.9 litres (Fig. 3). Adsorbent tubes were used to take air samples from the chamber. A hydrocarbon trap (Alltech, Associates Inc., USA) containing activated carbon and $MnO_2$-

coated copper nets was mounted between pump box and chamber to remove VOCs and $O_3$ to ensure that only clean air entered the chamber. The chamber consisted of a metal frame and a flexible polyethylene foam base to fit the tree trunk and was fastened with straps around the tree trunk. The inside of the chamber had been carefully wrapped with pre-conditioned (oven-cleaned 40°C, 3 hours) polyamid bags (Toppits, Cofresco Frischhalteprodukte GmbH, Germany) to avoid contamination with BVOC from the chamber foam base. During

measurements, the chamber was closed with a metal lid with in- and outgoing PTFE-lines (1/4") for purge and sample flow. Temperature of the air within the chamber was measured with a temperature probe (HI 145, Hanna Instruments, RI, USA) during sampling, and an infrared thermometer (IRT260, Biltema, Sweden) was used to take bark surface temperature before and after each sample from bark inside the chamber.

For each campaign, trees from one plot were sampled per day, typically starting at 08:00 (LT) and ending at 19:00 (LT) alternating sampling between the trees. The chamber bases were secured in place onto the tree trunks every morning and were left open during the day to avoid built up concentrations inside the chambers. Prior to the





sampling the bark temperature was measured at four different points inside the base after which the lid was fastened and the chamber was flushed for 15 minutes before sampling started.


The start of the infestation was determined by the beetles' swarming time in relation to when the tree infestation was detected. For plot 3, the start of the infestation was witnessed on site, for the other plots the swarming time was retrieved using data from Skogsstyrelsen Statistical Database (Skogsstyrelsen, n.d.).

The bark inside the chambers was controlled visually before each measurement to count the number of bark beetle holes as seen in Table 1. By looking at bark photographs, the holes were later separated into entry or exit holes for the infested trees. The separation depended on the characteristics of the hole where entry holes were determined to have more resin bleed compared to exit holes, which also had a rounder shape as seen in Fig. 4. Table 1 describes the number of visual holes inside the chamber and the extrapolated number per square meter of bark area.


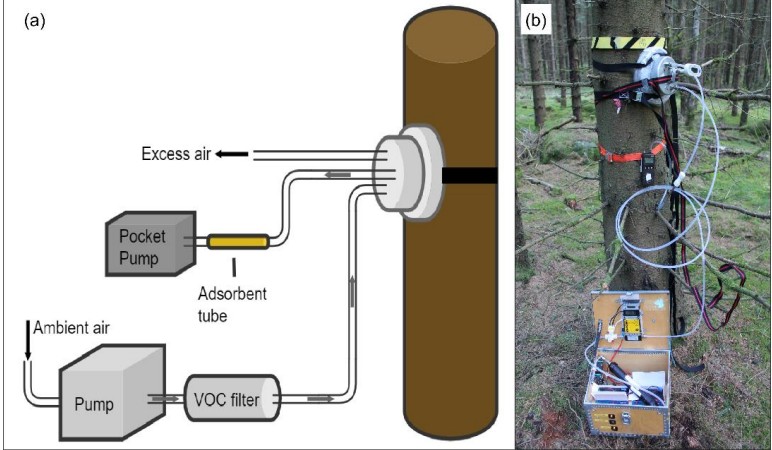

**Figure 3: The experimental schematics (a) contained a tree trunk chamber mounted on a tree, attached to a pump box used to provide BVOC- and O₃-free purge air. The BVOC samples were collected with adsorbent tubes connected to the chamber. A photograph of how the setup looked in the field is displayed to the right (b).**

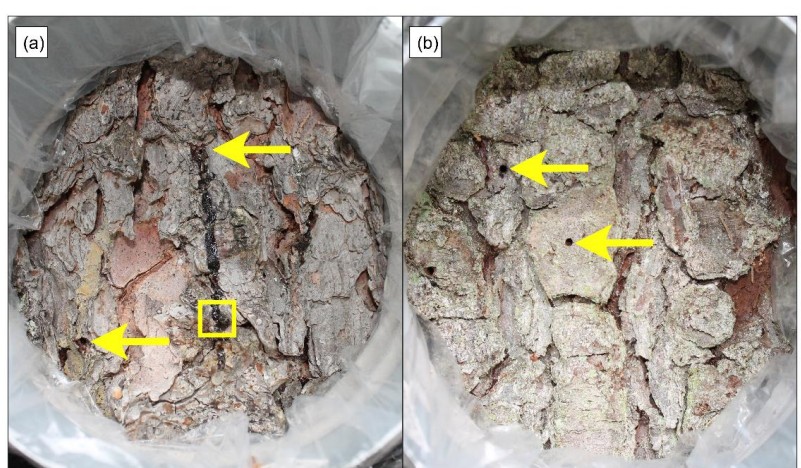






**Figure 4: Examples of infested trees with (a) entry holes and (b) exit holes, where the arrows points towards examples of bark beetle drilled holes and the square box shows a bark beetle. More holes can be found in the picture than pointed to.**

**Table 1. The infested trees and the number of holes counted inside the respective chamber at the given date. The counted holes were extrapolated to holes per square meter and the majority of the hole type was determined to either mostly exit holes or mostly entry holes. The tree with the ID S1S1 was infested in late season 2018 and had thus a majority of exit holes early in 2019.**

| Tree ID | Date | Number of holes inside chamber | Upscaled to holes per m$^2$ | Type majority |
|---------|------|-------------------------------|-----------------------------|---------------|
| S1S1 | 2019-05-04 | 12 | 1062 | exit |
| S1S1 | 2019-06-05 | 8 | 708 | exit |
| S3S2 | 2019-06-04 | 4 | 354 | entry |
| S3S3 | 2019-06-04 | 5 | 442 | entry |
| S3S2 | 2019-07-03 | 3 | 286 | entry |
| S3S3 | 2019-07-03 | 5 | 465 | entry |
| S3S2 | 2019-08-01 | 6 | 531 | entry |
| S3S3 | 2019-08-01 | 15 | 1273 | entry |
| S4S1 | 2019-08-21 | 4 | 354 | exit |
| S4S2 | 2019-08-21 | 5 | 442 | exit |
| S4S3 | 2019-08-21 | 5 | 442 | exit |
| S5S1 | 2019-08-22 | 5 | 442 | exit |
| S5S2 | 2019-08-22 | 4 | 340 | exit |
| S5S3 | 2019-08-22 | 7 | 619 | exit |

**2.3 BVOC sampling and analysis**

Stainless steel cartridges (Markes International Limited, Llantrisant, UK) packed with adsorbents Tenax TA (a porous organic polymer) and Carbograph 1TD (graphitized carbon black) were used to sample BVOC. The BVOCs were sampled from the chambers using flow-controlled pocket pumps (Pocket Pump, SKC Ltd., Dorset, UK) which extracted the air through the steel cartridges at a flow rate of 200 ml min$^{-1}$ and a sampling time of ca. 30 minutes,

the collected volume for each sample was between 5 to 6 liter. Blank samples were collected from air entering the chamber twice a day to capture possible background contamination of the filtered purge air. Temperature was measured inside the chamber at the start of the BVOC sampling and at the end to note potential temperature differences during the sample period. After sampling was finished, the bark temperature was measured again following the same procedure as mentioned above. The same method was repeated throughout the day until all trees

of that plot were measured three times.

After collecting the BVOC samples, the adsorbent cartridges were capped and stored in a refrigerator (at ~3 °C) before being analysed using a two-stage automated thermal desorption apparatus coupled to a gas-chromatograph mass-spectrometer. Desorption was done on a Turbomatrix ATD 650 (PerkinElmer, Waltham, MA, USA).

Cartridges were initially heated to 280 °C in a flow of purified helium for 10 minutes, in order for the VOCs to volatilize. After the primary desorption, VOCs were cryo-focused downstream on a Tenax TA cold trap maintained at –30 °C. The cold trap was flash-heated (40 °C sec$^{-1}$) to 300 °C for 6 minutes to perform a second desorption. The volatilized VOCs were passed via a heated transfer line using He as carrier gas, to a gas chromatograph-mass spectrometry system (GC-MS, Shimadzu QP2010 Plus, Shimadzu Corporation, Japan). The BVOCs were separated





using a BPX5 capillary column (50m, I.D. 0.32mm, film thickness 1.0 µm, Trajan Scientific, Australia) and the oven temperature was initially held at 40 °C for 1 minute, raised to 210 °C at a rate of 5 °C min⁻¹ and further increased to 250 °C at a rate of 20 °C min⁻¹ and lastly held for 2 minutes. Pure standard solutions of isoprene, α-pinene, β-pinene, p-cymene, eucalyptol, limonene, 3-carene, linalool, α-humulene, β-caryophyllene, longifolene and myrcene were pre-prepared in methanol (Merck KGaA, Darmstadt, Germany) and injected onto adsorbent

cartridges in a stream of helium and analyzed with the same conditions as samples. When quantifying BVOCs for which no standards were available, α-pinene was used for MTs, and α-humulene for sesquiterpenes (SQT). The peaks of longifolene and β-caryophyllene where coeluted in the chromatography and are therefore presented together as a sum of two compounds in this study. The chromatogram peaks were identified based on comparison with retention times and mass spectra of standards and the mass spectra in the NIST08 library.  LabSolutions GCMS

post run analysis program was used for data processing (Version 4.30, Shimadzu Corporation). Detection limit was set to 0.4 ng in the analysis software based on the analysis of blank samples.

**2.4 Emission rate calculation and standardization**

The BVOC concentrations obtained from the sample air analysis were converted to emission rate (ER) (µg m⁻² h⁻¹) according to Eq. (1), following Ortega & Helmig (2008):

$$ER = \frac{[C_{out} - C_{in}]Q}{A},$$     (1)

where $C_{out}$ (µg l⁻¹) is the concentration of each compound within the chamber, and $C_{in}$ (µg l⁻¹) is the concentration of the compound in the filtered inlet air, $Q$ is the flow rate through the chamber (l min⁻¹) and $A$ is the bark surface area (m⁻²) covered by the chamber.

The ER per hole was calculated as the average by dividing the ER derived from Eq. (1) for the respective sample with the number of holes (the number of holes can be found in Table 1):

$$ER\ per\ hole = \frac{ER}{\#\ holes},$$     (2)

Finally, the ER for one square meter of bark surface, $ER_{sqm}$, was extrapolated based on the number of holes within the chamber and the chamber's bark area:


$$ER_{sqm} = ER \cdot \#holes \cdot \frac{1}{A},$$     (3)

To remove the influence from the variations of the emission rates due to the difference in amount of bark beetle holes, the emission rates were scaled to represent the same number of holes using an average number of bark beetle holes per square meter calculated using the counts from this study in Table 1. The average holes per square meter

was applied to the emission rates for the infested trees, enabling comparison of emission rates from all infested trees.

To separate entry and exit holes in the data set, measurements taken up to 100 days after infestation was determined to have a majority of entry holes and measurements later than 100 days since infestation had a majority of exit holes.

The timeframe of 100 days was determined by confirming the hole type majority from bark photographs.



As the bark surface temperature varied over the season and between the days, the emission rates were standardized using the algorithm for stored, temperature dependent BVOCs (G93) by Guenther et al., (1993):

$\quad M = M_S \cdot e^{(\beta(T - T_S))}$,  (4)

where $M$ is the emission rate ($\mu$g m$^{-2}$ h$^{-1}$) at a given bark temperature, $T$, and $\beta$ (0.09 K$^{-1}$) is an empirical coefficient establishing the temperature dependency (Guenther et al., 1993). $M_s$ is the emission rate at standard temperature $T_s$ of 30 °C.


The temperature sensitivity of compound emission rates was calculated using a Q$_{10}$ relationship (Lloyd and Taylor, 1994) following Seco et al. (2020) where the Q$_{10}$ coefficient represents the factor by which the compound emission rate increases for every 10 °C temperature increase from a reference emission rate, F$_0$. Only compounds appearing in more than three individual samples were selected for further analysis. Log transformed emission rates were binned 260 into 1 °C bins and the mean emission rate per bin was calculated except for bins with only one value. An orthogonal distance regression was applied to the binned mean emission rates weighed by their standard deviation to determine Q$_{10}$ and F$_0$ using:

$F = F_0 \cdot Q_{10}^{(T - T_0)/10}$,  (5)


where $F_0$ is the reference emission rate at temperature $T_0$ (=30 °C), F is the flux rate at bark surface temperature $T$ (°C), and $Q_{10}$ is the temperature coefficient.

Based on the Guenther algorithm (G93, Guenther et al., 1993; Eq. (4)) and the Q$_{10}$ temperature dependency 270 calculation (Q$_{10}$, Lloyd and Taylor, 1994; Eq. (5)), an estimation of the constitutive total bark VOC emission rate and the leaf VOC emission rate throughout the season was calculated. Both algorithms were used to calculate bark BVOC emissions, while only G93 was used to calculate the leaf emission, as well as bark SQT emission rate. The modelled emissions for bark was based on the measured tree trunk temperature from the ICOS ecosystem data in HTM (Heliasz, 2020), taken at 3 meter height. An average of the trunk temperature measurements was taken in 275 the north and east orientation of the trunk because the trunk BVOC sampling was taken in these orientations.

The leaf emissions for MT and SQT were calculated according to Eq. (4) using the standardized seasonal average emission rate (M$_s$), 1.25 $\mu$g g(dw)$^{-1}$ h$^{-1}$ for MT and 0.34 for SQT $\mu$g g(dw)$^{-1}$ h$^{-1}$, taken from van Meeningen et al. (2017). The temperature input was the canopy-level air temperature measured at 24 meter agl. taken from the HTM ICOS station (Heliasz, 2020). The output of Eq. (4) was scaled from g(dw) to m$^2$ by using SLA of 38.4 cm$^{-2}$ g$^{-1}$ 280 calculated from Wang et al. (2017).

Two outliers in the BVOC samples were found and after examination with bark photography, it was discovered that the chamber in both cases had been placed upon a small emerging branch with some spots of resin as well as with one single needle stuck on the bark. These samples were considered unusable and excluded from further analysis. 285 All samples from one spruce at plot 2 were also excluded from the analysis after discovering placements on top of a bark hole likely not originating from spruce bark beetles, and thus not suitable in this study.



### 2.5 Statistical analysis

The significance of the difference between the healthy tree sites, the emission rates from control trees and infested trees, the difference in emission rates from the two infested trees S3S2 and S3S3, and the difference between $Q_{10}$

and $F_0$ for healthy and infested trees was analyzed using a Kruskal-Wallis test. The level of significance was $p < 0.05$.

### 3 Results

### 3.1 Constitutive and induced BVOC emissions from Norway spruce bark

For the healthy spruce trees in HTM, the average total temperature-standardized bark emission rate from all

samples (n=113) was $31.89 \pm 51.67$ µg m$^{-2}$ h$^{-1}$ (mean ± standard deviation). The most dominant BVOC group was MTs ($29.37 \pm 51.01$ µg m$^{-2}$ h$^{-1}$) followed by SQTs ($2.12 \pm 3.17$ µg m$^{-2}$ h$^{-1}$). Isoprene emissions were detected in 58% of total samples from the healthy tree bark with an average emission rate of $0.40 \pm 0.85$ µg m$^{-2}$ h$^{-1}$.

The variability of the emission rates differed little between the sites where plot 1 had a daily average total

temperature standardized bark emission rate of $23.26 \pm 42.53$ µg m$^{-2}$ h$^{-1}$, plot 2 of $30.38 \pm 32.35$ µg m$^{-2}$ h$^{-1}$ and plot 3 of $47.54 \pm 79.22$ µg m$^{-2}$ h$^{-1}$. The standardized emission rates were ranging from 0-145.7 µg m$^{-2}$ h$^{-1}$, for plot 1, 0-94.2 µg m$^{-2}$ h$^{-1}$ for plot 2 and 1.5-234 µg m$^{-2}$ h$^{-1}$ for plot 3 where the medians are 4.1, 17 and 7.9 µg m$^{-2}$ h$^{-1}$ respectively (Fig. 5). No statistically significant difference (p-value > 0.3) was found in standardized emission rate between the sites and no clear pattern of diurnal variation was found in the samples.

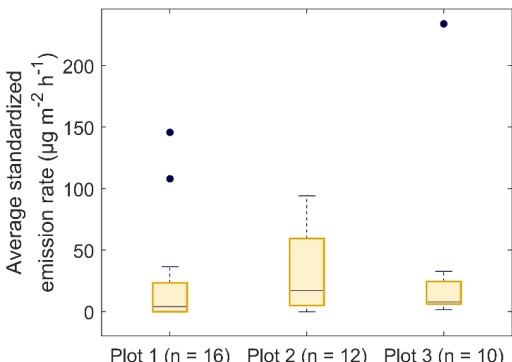

**Figure 5: The temperature-standardized emission rates of the control trees for the plot 1-3 in Hyltemossa, where plot 3 is located furthest away from the station in an older forest stand. There is no statistically significant difference for the daily average of the total temperature-standardized emission rates (p > 0.3).**


For the bark beetle infested trees, the calculations of seasonal average emission rate was separated into early season and late season as the bark beetle typically drill entry holes earlier, and exit holes later. The seasons were separated based on infestation start, where the early season was less than 100 days since infestation start and the late season after more than 100 days of infestation. The average total temperature standardized bark emission rate from the

early season bark beetle infested spruce trees (n = 6) was $6690 \pm 6860$ µg m$^{-2}$ h$^{-1}$ (mean ± standard deviation), while the average for the late season (n = 8) was $1970 \pm 1310$ µg m$^{-2}$ h$^{-1}$. MTs was the most dominant BVOC group throughout the season with an average of $6630 \pm 6740$ µg m$^{-2}$ h$^{-1}$ for the early season and $1950 \pm 1350$ µg m$^{-2}$ h$^{-1}$ for the late season, followed by SQTs (early: $53 \pm 74$ µg m$^{-2}$ h$^{-1}$, late: $18 \pm 24$ µg m$^{-2}$ h$^{-1}$). Throughout the season,



isoprene was also found in 42% of the samples with an average emission rate of 3.36 ± 6.69 µg m⁻² h⁻¹ during the
early season and 0.14 ± 0.20 µg m⁻² h⁻¹ for the late season.

For all trees, a total of 74 individual VOCs were found throughout the measurement period for all samples
(n = 151) whereof 32 were MTs, 5 were SQTs and 37 other BVOCs including isoprene. For the healthy spruce
tree samples (n = 113), 44 individual compounds were found in total, where 12 were MTs, 2 were SQTs and 30
other BVOCs including isoprene. Despite the lower sample count for the infested spruce tree samples (n = 38)
compared to the healthy tree samples, a higher number of individual compounds was found with 52 compounds
in total where the majority of the compounds were MTs (n = 30) which was more than the double compared to the
healthy trees. There were also more SQTs (n = 5) found in the infested tree samples, but less other BVOCs
including isoprene, (n = 17) compared to the healthy tree samples. For the infested trees, there was also a difference
in how many compounds were found early in the season compared to later, in total 40 and 33 individual compounds
were found for the early and late season respectively. For MTs and SQTs, more individual compounds were found
in the early season (27 MTs & 5 SQTs) compared to the late season (17 MTs & 2 SQTs), but for the other BVOCs
more were found in the later season which had 14 individual compounds identified compared to 8 in the early
season.


A significant difference was found for the daily average of the total standardized bark BVOC emission rate when
comparing healthy trees and infested trees, for both early and late season (p < 0.00; Fig. 6). The median of the
daily average emission rates is 8.6 µg m⁻² h⁻¹ for the healthy control trees, and 6385 µg m⁻² h⁻¹ and 2102 µg m⁻² h⁻¹
for the infested trees during early and late season respectively, i.e. around 740 and 240 times higher for the infested
individuals early and late in the season, respectively.

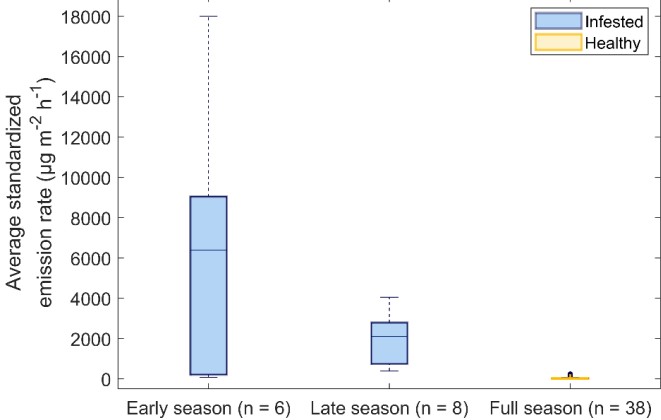

**Figure 6: Comparison between the daily average total standardized emission rate per measured tree and time for
healthy trees measured in Hyltemossa and infested trees measured both in Hyltemossa and Norunda. To consider the
seasonal pattern of the spruce bark beetle, the infested trees were divided into early and late season, where the early
season is dominated by entry holes and the late season by exit holes. The emission rate from the control trees are
clearly lower than of the infested trees for both the early and late season, with a significance of p < 0.00 for both. The
emission rates are standardized to temperature (30°C).**

A difference between the healthy trees and infested trees was also apparent in the occurrence of the compounds
throughout the samples (Table 2). The most common MT compounds among the healthy spruce trees were α-
pinene (76 %, relative occurrence in all samples), β-pinene (55 %), 3-carene (48 %) and limonene (44 %). For



infested spruce trees in both seasons, the mentioned MTs were also the most occurring compounds, however, they occurred in more samples (88-100 %). The late season also had 100 % occurrence of (1S)-camphene while that compound occurred in 75 % of the samples for the early season. For SQTs, α-humulene occurred most among the

healthy trees (47 %) followed by longifolene+β-caryophyllene (17 %). For the infested trees in the late season, the SQTs occurring most were longifolene+β-caryophyllene (55 %) followed by α-humulene (18 %). For the early season, the occurrence was similar for longifolene+β-caryophyllene (56 %) but α-humulene occurred in more samples (31 %) compared to the late season. The SQTs germecrene D, isoledene and β-cubene were found to be emitted from one of the infested trees in the early season, but was not discovered at any other time. Isoprene was

found to be mostly occurring among the other BVOCs for both the healthy (58 %) and infested spruce during the early and late season (63 % and 27 % respectively). After isoprene, decanal (45 %), benzene (45 %), nonanal (38 %) and toluene (21 %) were occurring most for the healthy spruce. For the early season infested spruce trees, 2-methyl-1-phenylpropene (38 %) and 2-methyl-3-buten-2-ol (19 %) were occurring most after isoprene, however, this was not the case for the infested trees in the late season where 2-methyl-3-buten-2-ol was not emitted and 2-

methyl-1-phenylpropene occurred in 9 % of the samples.

Compared to the constitutive emission rates from healthy trees, the emission rates of infested trees were shown to increase for all individual compounds with increases ranging from 200 % to 250 000 % for both early and late season (Table 2). The group of MTs had the highest increase of 22678 % during the early season while SQTs

increased by 2400 % and isoprene increased with 740 %. The emission rates during the late season also contributed to an increase of 6608 % for the MTs and 749 % for the SQTs, however, isoprene was found to be decreasing from the infested tree emission rates in the late season (-65 %). The compound (+)-sabinene had the highest increase of all individual compounds (257863 %) during the early season when comparing healthy and infested tree emission rates (Table A1). The compounds: tricyclene, eucalyptol, 4-carene, zeta-fenchene, α-phellandrene, trans,trans-

alloocimene, norbornane, gamma-terpinene, α-fenchene, 2-carene, α-thujene, α-terpinene only occurred in infested trees and indicate a change in the chemical composition of the emitted BVOCs when a tree is infested (Table A1).

**Table 2. The compounds occurring most from all samples throughout the season for healthy trees and infested trees.**
**Presented is the temperature standardized seasonal average emission rate (µg m⁻² h⁻¹ ± one standard deviation) for each**
**compound and the groups of MT, SQT and other BVOCs and the occurrence (%) of each compound in all samples.**
**The increase (%) is presented for the infested trees as an increase from healthy to infested. The compounds that were**
**identified but unable to quantify is presented as n.q. (no quantification). A full list of all identified compounds is found**
**in Table A1.**

| Compound name | Healthy | | Infested early season | | | Infested late season | | |
|---|---|---|---|---|---|---|---|---|
| | *average ± std (µg m⁻² h⁻¹)* | *occurrence (%)* | *average ± std (µg m⁻² h⁻¹)* | *increase (%)* | *occurrence (%)* | *average ± std (µg m⁻² h⁻¹)* | *increase (%)* | *occurrence (%)* |
| **Monoterpenes** | 29.37 ± 51.01 | | 6630 ± 6740 | 22678 | | 1950 ± 1350 | 6608 | |
| alpha-Pinene | 11.49 | 76.11 | 911.14 | 7830 | 100 | 824.64 | 7077 | 100 |
| beta-Pinene | 8.22 | 55.75 | 954.17 | 11508 | 100 | 225.28 | 2641 | 100 |
| 3-Carene | 2.48 | 48.67 | 285.16 | 11398 | 100 | 33.07 | 1233 | 95 |
| Limonene | 1.89 | 44.25 | 320.82 | 16875 | 88 | 85.43 | 4420 | 100 |
| p-Cymene | 0.49 | 39.82 | 241.45 | 49176 | 63 | 53 | 10716 | 77 |
| beta-Myrcene | 0.32 | 17.70 | 159.76 | 49825 | 79 | 6.3 | 1869 | 86 |
| beta-Phellandrene | 2.7 | 10.62 | 673.13 | 24831 | 44 | 189.47 | 6917 | 68 |
| (1S)-Camphene | 1.7 | 6.19 | 1516 | 89076 | 75 | 388.82 | 22772 | 100 |





| | | | | | | | | |
|---|---|---|---|---|---|---|---|---|
| (+)-Sabinene | 0.08 | 0.88 | 206.37 | 257863 | 44 | 2.93 | 3563 | 5 |
| **Sesquiterpenes** | 2.12 ± 3.17 | | 53.0 ± 74 | 2400 | | 18 ± 24 | 749 | |
| Longifolene+beta-Caryophyllene | 0.7 | 17.70 | 37.65 | 5279 | 56 | 13.6 | 1843 | 55 |
| alpha-Humulene | 1.42 | 47.79 | 4.86 | 242 | 31 | 4.52 | 218 | 18 |
| Germacrene D | - | - | 3.86 | - | 19 | - | - | - |
| Isoledene | - | - | 3.25 | - | 19 | - | - | - |
| beta-Cubebene | - | - | 3.2 | - | 19 | - | - | - |
| **Other BVOCs** | 0.40 ± 0.85 | | 3.36 ± 6.69 | | | 0.14 ± 0.20 | | |
| Isoprene | 0.4 | 58.41 | 3.36 | 740 | 63 | 0.14 | -65 | 27 |
| Decanal | n.q. | 45.13 | n.q. | - | 13 | n.q. | - | 14 |
| Benzene | n.q. | 45.13 | - | - | - | n.q. | - | 14 |
| Nonanal | n.q. | 38.94 | n.q. | - | 6 | n.q. | - | 14 |
| Toluene | n.q. | 21.24 | n.q. | - | 6 | n.q. | - | 9 |
| 2-Methyl-1-phenylpropene | - | - | n.q. | - | 38 | n.q. | - | 9 |
| 2-methyl-3-buten-2-ol | - | - | n.q. | - | 19 | - | - | - |

### 3.1.1 Scaling the infested tree bark emission with number of bark beetle holes

The temperature standardized emission rates (TS) for the total BVOCs from the bark beetle infested trees had seasonal averages ranging from about 500 to 13000 µg m$^{-2}$ h$^{-1}$ for all trees throughout the measurement period (Fig. 7). The daily average TS emission rate per bark beetle hole for infested trees during the early season was 22.04 ± 28.69 µg hole$^{-1}$ h$^{-1}$ which is emissions from mainly entrance holes. The bark beetle holes from infested trees during the late season and with mainly exit holes had a daily average standardized emission rate of 4.41 ± 3.51 µg hole$^{-1}$ h$^{-1}$. The average number of holes per square meter bark area found in this study was 554 based on the values in Table 1. When applying the average number of holes to emission rates with bark beetle holes (BBH), a comparison with the TS emission rates showed that the total average of BBH emission rates for the trees measured in the early season were about 2500 µg m$^{-2}$ h$^{-1}$ higher (9200 ± 6230 µg m$^{-2}$ h$^{-1}$) compared to TS emission rates (6690 ± 5140 µg m$^{-2}$ h$^{-1}$). For the trees measured in the late season the BBH emission rates were 150 µg m$^{-2}$ h$^{-1}$ higher (900 ± 650 µg m$^{-2}$ h$^{-1}$) compared to TS emission rates (750 ± 400 µg m$^{-2}$ h$^{-1}$). For the individual trees and for the group of MT, the BBH emission rates increased compared to TS emission rates for all trees but two (tree S1S1 and tree S5S3) where the TS emission rates were about 300 µg m$^{-2}$ h$^{-1}$ higher (Fig. 7). The inconsistent variation in emission rates scaled with BBH or TS can be explained by the difference in number of bark beetle holes found per tree (Table 1). The TS emission rates only consider the bark beetle holes inside the bark chamber while the BBH emission rates are calculated based on holes extrapolated to average holes per square meter, and as the same average of bark beetle holes were used for all trees, any variation due to amount of holes can be disregarded. The results from the infested trees were thus from here on presented as BBH emission unless stated otherwise.





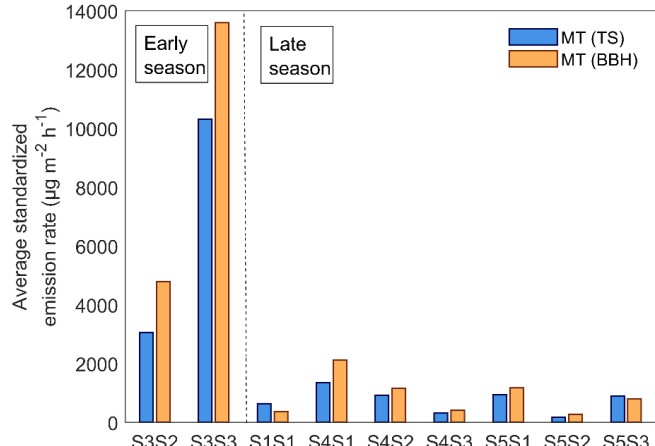

**Figure 7: The seasonal average standardized emission rate for all infested trees for the group of monoterpenes (MT), separated into early season (< 100 days since infestation start) and late season (> 100 days since infestation start). The temperature standardized emission rates (TS) are presented in blue, while the emission rates also scaled to the same number of bark beetle holes (BBH) is presented in orange. The emission rates are a lot higher in the early season where the emissions occur mainly from entry holes compared to the late season with mainly exit holes. This is evident even if the emission rates were re-calculated to represent equal number of holes.**

**3.2 The influence of time since infestation on emission rate from infested trees**

After a tree was infested by bark beetles, its emission rates were found to decrease with time passed since the start of the infestation. When the measurement started after 12 days since the start of infestation, the average emission rate for all VOCs where around 1850 µg m$^{-2}$ h$^{-1}$, when excluding the tree with lowered defence (as presented in Sect. 3.2.3; Fig. 8a; excluded tree is marked in yellow). An exponential function ($f(x) = a \times e^{b \times x}$, where $x$ is the emission rate in µg m$^{-2}$ h$^{-1}$) was fitted to all data points. Three compounds were selected for further analysis, β-phellandrene, eucalyptol and β-pinene, using the same exponential function. The emission rates after 12 days were different for the individual compounds compared to the total average, β-phellandrene and β-pinene have emission rates of around 3000-3500 µg m$^{-2}$ h$^{-1}$ and 2000-2500 µg m$^{-2}$ h$^{-1}$ respectively (Fig. 8b and d). Eucalyptol was emitted at slightly lower rates of around 4 and 25 µg m$^{-2}$ h$^{-1}$, depending on if the tree had lowered defence or not, where the low emission rates came from the lowered defence tree (Fig. 8c). Some compounds were not emitted from all infested trees: eucalyptol was only observed from 4 individual trees, and β-phellandrene from 7 trees, while β-pinene was emitted from all infested trees (n = 9).

After about 100 days since start of infestation, the trees were showing signs of browning and loss of needles. The emission rates for the total BVOCs had decreased with about 80 % on average from the start to around 300 µg m$^{-2}$ h$^{-1}$, but still at levels higher than the seasonal constitutive emissions from healthy trees (around 30 µg m$^{-2}$ h$^{-1}$). Compared to the emission rate of the total BVOCs after 100 days since infestation, the emission rates from the compounds β-phellandrene and β-pinene were at about the same level, however, the decrease on average since the start of infestation was higher (88 % and 89 %, respectively). Eucalyptol did not have as distinct decrease, after 100 days the emission rates had only decreased with 10 % on average. The tree measured after more than 300 days since the start of infestation had lost almost all of its needles and some bark, and the emission rates of the total BVOCs were down to around 40 µg m$^{-2}$ h$^{-1}$, which was at the same level as the constitutive emissions at that time (on average 38 µg m$^{-2}$ h$^{-1}$). No emissions of eucalyptol were found after more than 300 days,





but the emission rates of β-phellandrene and β-pinene after 315 days post infestation were around 70 and 268 µg m$^{-2}$ h$^{-1}$ respectively, however, after 350 days the emission rates went down to around 32 and 58 µg m$^{-2}$ h$^{-1}$ respectively, also comparable with the constitutive emissions at that time (around 45 µg m$^{-2}$ h$^{-1}$).

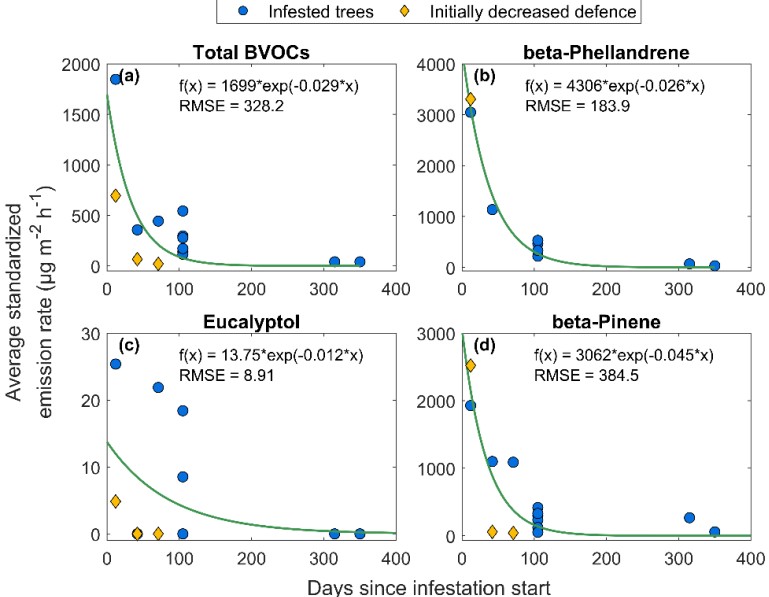

**Figure 8:** The relationship between average standardized emission rate from all infested trees and the number of days passed since start of infestation for (a) all BVOCs and the compounds: (b) beta-phellandrene, (c) eucalyptol and (d) beta-pinene. All trees are included in the exponential fitted curve, however one tree had initially lowered defence from a late bark beetle attack previous season and is in the figure marked in yellow and diamond-shape for visualization.

**3.3 The difference in BVOC emission rates from bark beetle entry holes and exit holes**

No clear relationship was found between the total number of holes and emission rates, likely due to a mixed signal from the type (entry or exit) and time since infestation. The total BVOC temperature standardized emission rates were generally lower from exit holes compared to entry holes when the number of holes were similar (Fig. 9a). The individual compounds emitted from both the entry and exit holes were dominated by β-phellandrene, β-pinene, α-pinene and (1S)-camphene. The compounds found from entry holes but not from exit holes were: 2-carene, 4-carene, α-fenchene, α-phellandrene, α-terpinene, α-thujene, β-cubebene, ϒ-terpinene, germacrene D, isoledene and trans-alloocimene. Generally lower emission rates from exit holes were also seen for the compounds β-phellandrene and β-pinene (Fig. 9b,d). However, for the compound eucalyptol (Fig. 9c) emissions were only found from four individuals, which had similar emission rates regardless of entry or exit holes. The oxygenated compounds myrtenal and bornyl acetate were only found in entry holes but could not be quantified.



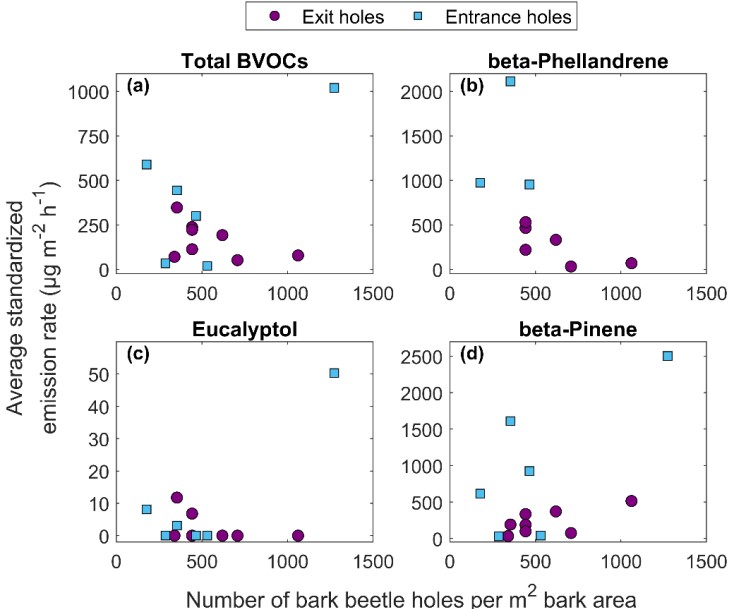

**Figure 9:** The relationship between average standardized emission rate from infested trees and the number of bark beetle holes per m$^2$ bark area for (a) all BVOCs and the compounds: (b) beta-phellandrene, (c) eucalyptol and (d) beta-pinene. There is a distinction between the entry holes (green squares) and the exit holes (purple circles). The exit holes were appearing later during the season (> 100 days since infestation start) which could explain the lower emission rates as the vitality of the trees decrease with higher number of holes. High emission rates from exit holes could indicate signals from blue stain fungi.

### 3.4 Bark beetle infestation impact over time from two trees with different initial health status

At plot 3, where the bark beetle trap was installed, two trees were infested by bark beetles. One of the trees had no visual stress signs and was assumed to be healthy (S3S3). The other tree (S3S2) had visual signs of stress already before the beetle attack with old resin flow located high on the stem, most likely due to a late summer attack the previous season. The different status of the trees can be identified in Fig. 10 (a-b) where S3S2 had significantly higher (p > 0.02) total emission rate of bark BVOCs in May, before the infestation, compared to the healthy spruce (S3S3). An average of the two control trees at plot 3 is also presented in Figure 8, however, only four compounds were found in May, longifolene+β-caryophyllene, α-humulene and isoprene, which had similar emission rates as the other trees. The control trees remained at low emission rates for the remaining months (Fig. 10a-h) and are thus not included in further comparison of the trees. The total emission rates for both trees were induced in June when the bark beetle infestation started (Fig. 10c-d) but there was no significant difference (p > 0.2) in emission rates between the trees. However, a difference was seen in the compound blend. S3S3 had a higher emission rate from the compound 1S-camphene (about 800 μg m$^{-2}$ h$^{-1}$) and was also emitting zeta-fenchene, trans,trans-alloocimene, norbornane and α-thujene, compounds which were not emitted from S3S2. The samples from S3S2 in June (Fig.10c-d) did however contain the bark beetle pheromones germacrene D, isoledene and β-cubene which were not found in the compound blend from S3S3. In July, the emission rate from S3S2 was significantly lower (p < 0.025) and close to zero compared to S3S3, which still had high emission rates of β-phellandrene, α-pinene, β-pinene and 1S-camphene. A similar difference between the trees was apparent in August as well (Fig. 10e-f) where the emission rates from S3S2 were still significantly lower (p < 0.003) compared to S3S3. The spruce S3S3 was also found to emit the compound verbenone in August, which was not found in S3S2.





The individual compound blend was also found to change over time for the healthy tree, S3S3, when it got infested and as the infestation continued (Fig. A1). In May, when the tree was healthy, in total 10 compounds were identified, with dominant BVOC being decanal (28 %), nonanal (20 %), toluene (16 %) and 1,3,5-triflourobenzene (15 %) by mass. After bark beetle infestation in June, the number of detected compounds raised to 27, dominated

by MTs (1S-camphene (18%), β-phellandrene (9%)) but also other BVOCs (5-vinyl-m-xylene (15%)). The emissions during the campaign in July consisted mainly of MTs with largest contributions from β-phellandrene and β-pinene (23 % and 22 % respectively) followed by 1S-camphene and α-pinene (19 % and 16 % respectively). The compound composition in August was similar as in June, with the majority of the blend consisting of MTs dominated by 1S-camphene, α-pinene and β-pinene (22 %, 13 % & 11 % respectively) and other BVOCs (2-

methyl-1-phenylpropene (6%)).

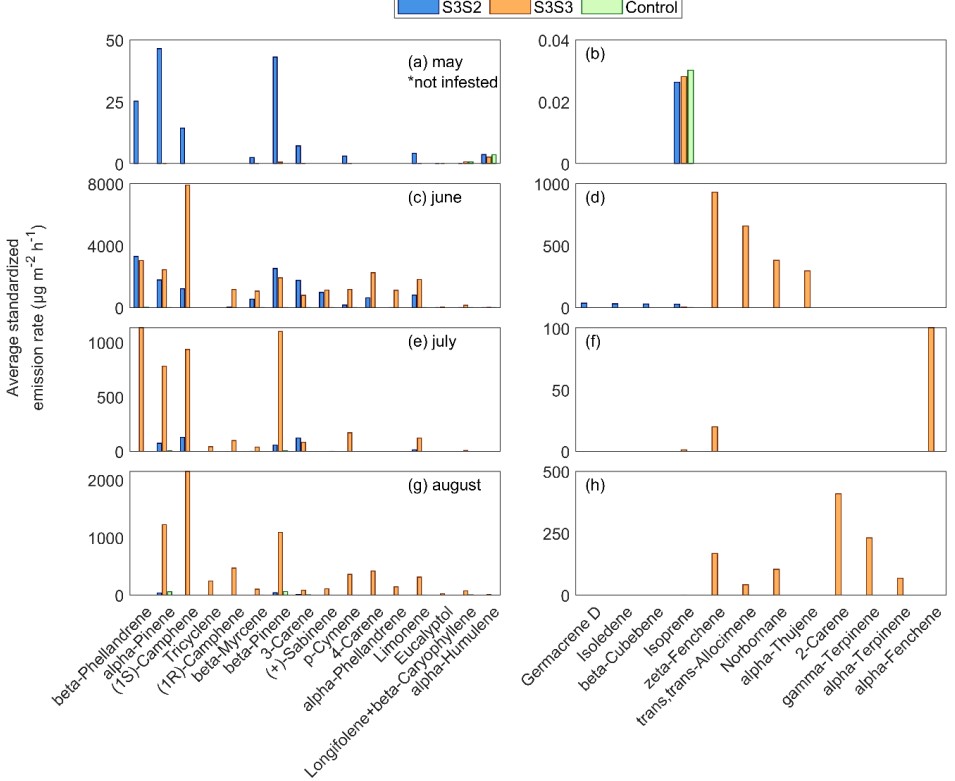

**Figure 10: The average standardized emission rates for all compounds for the two spruces S3S2 (blue), S3S3 (orange) and control trees at site 3 (green) for May (a-b), June (c-d), July (e-f) and August (g-h). The graphs are horizontally**
**separated for visibility due to large differences in scale. The control trees are included in all graphs but the emission rates are not visible on the same scale as the infested trees in June (c-d) or July (e-f). The bark beetle infestation had not started in May (a-b), however, the spruce S3S2 was already subjected to stress from late bark beetle attacks previous season before the bark beetle infestation started again in June (c-d), leading to higher emission rates in May.**

**3.5 Reference emission rate at 30°C and calculated $Q_{10}$ coefficient**

From Eq. (5), the reference emission rate at 30°C ($F_0$) and the increase in emission rate with every 10°C ($Q_{10}$ coefficient) were calculated for both the healthy spruce trees as well as for the infested trees. Only compounds





found in at least three samples were included in the calculation. The result for emitted compounds from both healthy trees and infested trees shows a $Q_{10}$ coefficient spread from 0.1 to 56 for healthy trees and 1.3 to 981 for

infested trees where the $Q_{10}$ coefficient was higher when the trees where infested for all compounds but one, p-cymene (Table A2). The $F_0$ value indicates the emission rate that compound would have at 30°C and for this value there was also a difference between the healthy and infested trees. The spread of $F_0$ for healthy trees was ranging from 0.005 to 92 µg m$^{-2}$ h$^{-1}$ and from 0.5 to 34900 µg m$^{-2}$ h$^{-1}$ for the infested trees, but in this case the $F_0$ was higher for all compounds from the infested trees compared to the healthy (Table A2). The average $Q_{10}$

coefficient for all compounds for healthy trees was 13 while it was 96 for infested trees, leading to an increase of the $Q_{10}$ coefficient by 634 %. The same average for $F_0$ was 20.8 µg m$^{-2}$ h$^{-1}$ for healthy trees and 2648 µg m$^{-2}$ h$^{-1}$ for infested trees, an increase of 12600 %, which is in line with the increased emission rates when standardized according to G93 (Table A1). The highest increase in both $Q_{10}$ and $F_0$ was seen in the compounds β-pinene and longifolene+β-caryophyllene (two SQTs quantified together) with $Q_{10}$ increasing with 12500 % and 22400 %

respectively and $F_0$ with 316000 % and 209800 % respectively. The lowest change for $Q_{10}$ was seen in α-pinene and p-cymene, where α-pinene was increasing with 0.7 % from healthy to infested and the coefficient for p-cymene was actually decreasing with 34 % for infested trees compared to healthy. Despite the lower $Q_{10}$ coefficient for p-cymene in infested trees, its $F_0$ was still higher for the infested trees, however, it had the lowest increase with 449 % compared to the other MT compounds. Isoprene was seen to have the overall lowest increase in $F_0$, increasing

with 89 % from healthy to infested. A significant difference was found for the $Q_{10}$ coefficients for healthy and infested trees ($p < 0.032$) as well as for $F_0$ ($p < 0.006$).

There were four compounds for which the requirements for the calculations where only fulfilled for infested trees. Those where eucalyptol, tricyclene, (1R)-camphene and (+)-sabinene, for which an increase or comparison

between healthy and infested cannot be made, but this might indicate that these compounds could be limited to emissions from infested trees only.

### 3.6 Calculated constitutive BVOC emissions from bark over the season and comparison with leaf emissions

Leaf emissions over the growing season 2019 varied between an average of around 60 to 170 µg m$^{-2}$ h$^{-1}$ for leaf

MT in July and August and an average of around 25 to 100, and 50 to 120 µg m$^{-2}$ h$^{-1}$ in May and September, respectively (Fig. 11b). Bark emissions have been based on measured tree trunk temperature at 3m agl, averaged from 2 directions (North and East) and the average standardized emissions (Ms), 29 µg m$^{-2}$ h$^{-1}$ for MT and 2 µg m$^{-2}$ h$^{-1}$ for SQT, and the $Q_{10}$ approach for healthy trees. The calculated emission rates from bark reached a maximum around 16 µg m$^{-2}$ h$^{-1}$ in July, which is ten times lower than the calculated leaf emissions at the same

time. The bark emission rates remained below 10 µg m$^{-2}$ h$^{-1}$ for most of the growing season. The estimated bark emission rates from healthy trees using the $Q_{10}$ approach were generally about 5 µg m$^{-2}$ h$^{-1}$ lower than the calculated emission rates using the G93 approach, but steeply increased during the warmest days to match the G93-emissions (Fig. 11c).

For SQTs, the leaf emissions peaked at 30 µg m$^{-2}$ h$^{-1}$ in late July at the same time when MT emissions were high, and also showed emissions up to 20 µg m$^{-2}$ h$^{-1}$ earlier in June. For most of May and September, SQT emissions from leaves were calculated to be below 5 µg m$^{-2}$ h$^{-1}$. Bark emissions of SQT for healthy trees estimated with G93





were well below 1 µg m$^{-2}$ h$^{-1}$ throughout the season (maximum 0.75 µg m$^{-2}$ h$^{-1}$ in late July), and below 0.3 µg m$^{-2}$ h$^{-1}$ most of the time.


However, when comparing estimated bark emission from healthy trees with actual measurements of infested trees, bark emissions from infested trees were much higher. The measured bark MT emission rate from the infested tree reached up to around 18000 µg m$^{-2}$ h$^{-1}$ as a daily average for one day, making the total MT emission rate (including leaf emissions) increase by almost a 100-fold when the tree was infested. The lowest measured infested tree

emission rate (around 3900 µg m$^{-2}$ h$^{-1}$) for MT was found during the July campaign, however, this was still considerably higher than the constitutive MT emission rate of that day, which was calculated to be around 70 µg m$^{-2}$ h$^{-1}$ including both leaf and bark VOC emission. Emission rates from the bark of the infested trees were about 55 times higher than the total constitutive MT emission rate from both leaves and bark, even at the lowest measured emission rates.


For the SQT emission rates, the difference was not as distinct. The SQT emission rate reached maximum around 0.75 µg m$^{-2}$ h$^{-1}$, for bark emissions (Fig. A2b) and around 30 µg m$^{-2}$ h$^{-1}$ for leaf emission (Fig. A2a) when the measured bark emission rate from the infested tree peaked around 40 µg m$^{-2}$ h$^{-1}$ as a daily average for one day, indicating an 1.3-fold increase when a tree is infested. The lowest measured infested tree emission rate was also

in July for the SQTs, at around 1.4 µg m$^{-2}$ h$^{-1}$, which was still higher than the calculated constitutive bark emission rate at around 0.2 µg m$^{-2}$ h$^{-1}$, but lower than the constitutive leaf emission rate of about 5 µg m$^{-2}$ h$^{-1}$.

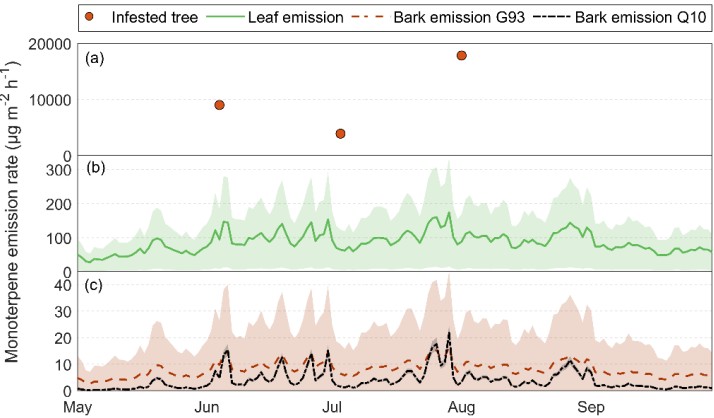

**Figure 11. The calculated constitutive (c) bark VOC emission rates from the group monoterpenes based on the tree**
**temperature taken at 3 meters height in the north and east orientation, data taken from the HTM ICOS station (Heliasz, 2020). The bark emission (black) is calculated based on the Guenther algorithm (Guenther et al., 1993) and the Q$_{10}$ temperature dependency (orange) based on measured emission rates in this study. The (b) leaf emission rates (green) are calculated based on the Guenther algorithm and the measured emission rates are taken from van Meeningen et al., (2017) and specific leaf area (SLA) was taken from Wang et al., (2017), using the air temperature at 24 meters taken**
**from the HTM ICOS station (Heliasz, 2020). For comparison, (a) the actual measured bark VOC emission rates from one infested tree over time, from this study, is included in the Figure (red dot).**





## 4 Discussion

### 4.1 Constitutive and induced bark BVOC emissions from Norway spruce

Both emission rates and composition blend of bark BVOCs from Norway spruce trees were found to change when infested by spruce bark beetles, which is in line with previous studies on bark beetle infestation of conifer trees (Amin et al., 2013; Ghimire et al., 2016). In this study, we identified 29 compounds unique to infested trees where the majority were MTs (n = 19). Several of the identified compounds were only emitted from infested trees, for example, eucalyptol, isoledene, (+)-camphor, tricyclene, α-phellandrene, which is consistent with the findings of

Ghimire et al. (2016). Isoprene was also found to be emitted from both healthy trees and infested trees, which was believed to originate from potential lichen cover (Zhang-Turpeinen et al., 2021), however, there seemed not to be a relationship when assessing the lichen cover from bark photographs, making it uncertain if the isoprene emission did originate from the bark or not. This is also consistent with the study by Ghimire et al. (2016) in which they did not find any significant relationship with isoprene emission from bark and lichen or algal cover. With regard to

the quantity of BVOC emission however, the results of this study indicate a much greater difference between emission rates from healthy trees and infested trees than previous findings. We found the total constitutive bark BVOC emission rates from healthy trees to have a seasonal average of $31.89 \pm 51.67$ µg m$^{-2}$ h$^{-1}$ (mean ± standard deviation), while the infested trees had an average of $6630 \pm 6740$ µg m$^{-2}$ h$^{-1}$ (mean ± standard deviation) for the early season and $1950 \pm 1350$ µg m$^{-2}$ h$^{-1}$ for the late season. This implies that bark from infested trees emits 63 to

215 times more BVOC to the atmosphere than healthy trees, depending on how long the infestation has been ongoing where the emissions are higher earlier and decrease with time. The measured emission rates in this study are 3 to 9 times higher than the emission rates in the study by Ghimire et al. (2016) in which they found 15-fold increased emission rates from all BVOCs when comparing healthy Norway spruce trees with trees infested by the spruce bark beetle. A possible reason for the difference in measured emission rates might be the time of the

infestation in relation to the measurements. However, the measured emission rates in this study from exit holes during the late season is still higher than the emission rates found in June for Ghimire et al. (2016). In our study, we conducted measurements throughout the growing season, starting before the bark beetle infestations. This allowed us to capture emission rates from early infestation to later stages, something that, to our knowledge, has not been done before. This finding makes it very important to consider the influence of time since infestation when

modelling emission rates of BVOCs from infested Norway spruce. We could see a trend with exponential decrease in emission rates with time for both the total BVOCs as well as selected compounds. The spruce bark beetles typically have a first swarm in May, followed by a sister brood in June and the initiation of a second generation in July (Jönsson et al., 2009).The trend with decreased emission rates we found is only related to the start of infestation, regardless of the time in the season and from which swarming period the infestation started.


The timing of the infestation plays an important role in the increased emission rates. When comparing the total number of holes to emission rates for the individual spruce trees, no distinct pattern was found. The hypothesis was that emission rates would be higher at the start of infestation and decrease over time with declining vitality of the trees, which could be explained by a relationship between emission rates and the type of bark beetle hole rather

than the total number of holes. When separating the holes into entry and exit, it was apparent that entry holes generally have higher emission rates compared to exit holes (Fig. 9), which can also be supported by the relation to time since infestation where the emission rates were highest in the beginning (Fig. 8) and entry holes generally appear at the start of an infestation. For the total VOCs there is a large spread in emission rates, and the second highest emission rate came from an individual with less than 200 holes per square meter, which was determined



to have mainly entry holes (Fig. 9). Some of the lowest emission rates came from an individual with more than 1000 holes per square meter with mainly exit holes. The same is true when looking at the compounds β-phellandrene, eucalyptol and β-pinene. The importance of hole type could be further supported when the emission rates were scaled with temperature only and with additional scaling from the averaged bark beetle holes from our study (Fig. 7). Comparing all trees, there is a larger difference in emission rates comparing the two scaling

approaches from tree S3S2 and S3S3, which both had a majority of entry holes. This indicates that when only taking temperature into account, the emission rates were lower, but when all trees had been scaled to have the same number of bark beetle holes, the difference in emission rates is more distinct – especially from the two trees with a majority of entry holes. This supports the conclusion that high emissions are rather explained by the type of holes, where entrance holes are more relevant than exit holes, than the total amount of holes.


The number of identified compounds was also higher in emissions from entry holes. This is consistent with Birgersson and Bergström (1989) who looked at volatiles emitted from entry holes in bark beetle infested spruce. They did however not look at exit holes, but their findings show that during the early stage of an attack the MT emissions are high and the concentration of the collected MTs during the first day is consistent with what we found

in our study 12 days after infestation. Two oxygenated compounds were found only from the entry holes which is consistent with the findings of Birgersson and Bergström (1989) and indicates emissions from the phloem. The bark beetle pheromones: germacrene D, isoledene, β-cubebene and 2-methyl-3-buten-2-ol were also only found emitted from entry holes – however, only from one tree and one day throughout the study. This could be indications of beetles present in the galleries during the measurements of that individual tree.

**4.1.1 Indications of differences in emissions from healthy or stressed trees during infestation**

At plot 3 in HTM we selected trees with various health status to study if the health status might have an impact on induced tree emissions from bark beetle infestations. Prior to the infestation, we found that emission rates for two spruces were different; one emitted significantly more BVOCs compared to the other, a state which was indicating stress (Fig. 10). This was not only visible in the BVOC emissions, but also from resin flow on the bark, supporting

the claim that the high emission rates were from stress, something which was caused by a late summer attack from spruce bark beetles during the previous season. Despite the old infestation, the stressed tree was attacked by spruce bark beetles again during our measurement period. When both trees had been infested by bark beetles, and were 12 days into the infestation they both showed induced emission rates. There was no significant difference in their respective emission rates; however, the trees were emitting slightly different compound blends, which might have

been a cause from the initial status of the trees. The initially healthy tree showed induced emissions of mostly MTs where (1S)-camphene, β-phellandrene, α-pinene and 4-carene were dominating the emissions, but there were also emissions of ζ-fenchene, trans,trans-alloocimene, norbornane and α-thujene which were not found emitted from the initially stressed tree. The initially stressed tree was emitting several of the same MT with the majority being emissions of β-phellandrene and β-pinene. In addition to this, the bark beetle pheromones previously mentioned

being emitted once came from this initially stressed tree, and was only found 12 days into the infestation. The pheromones could indicate that there was ongoing blue stain fungi infection caused from the infestation that happened in the previous season, something that could be supported as the tree was cut down at the end of the measurement period. As the infestation continued, the emission rates from the already stressed spruce got significantly lower than from the initially healthy spruce, which continued at high emission rates until August.

However, the last measurement in August revealed occurrence of verbenone from the initially healthy tree, which have been found to be emitted with successful fungal establishment and has been shown to repel bark beetles




(Bakke, 2009; Cale et al., 2019). The findings of verbenone could indicate that the bark beetles had successfully overtaken the spruce in August, however, this could not be confirmed as the forest owner had to take down the trees which made further measurements impossible. As bark beetle outbreaks have been seen to increase in number, there might be an increase in the number of healthy trees being attacked and killed in addition to the typical attacks on already stressed trees. The results revealed different blends of compounds when a tree was already stressed from previous infestation and attacked again compared to when the tree was healthy before the attack. In addition to this, we could also see that the one with previous infestation indicated induced emission rates until the start of the next season, during which the tree was infested again with further induced emissions. The healthy tree did however have higher emission rates for longer when it was infested compared to the stressed one. These results indicate that the second generation of spruce bark beetles might lead to induced emission rates continuing until the next season, when new attacks can occur on the same tree, as well as on healthy trees. The increased attacks on healthy trees as well as the initiation of a second generation of spruce bark beetles might have a large impact on the total bark BVOC emission rates from Norway spruce.

The high BVOC emission rates from infested trees does not only affect the trees themselves, but the emissions also impact atmospheric processes. Induced emission rates of BVOCs due to insect herbivory have been found to potentially increase SOA yields when modelling an increase in emission rates (Bergström et al., 2014), which would support the claim that bark beetle induced emission rates could be important to consider when modelling or measuring SOA formation. Taking not only the quantitative aspects of bark beetle induced emission rates, but also the qualitative effects into account, the SQTs α-humulene, longifolene and β-caryophyllene have been found to have highest SOA yields compared to 16 other BVOCs, where α-pinene had the 9th highest SOA yield (Lee et al., 2006). In our study we found increases of longifolene+β-caryophyllene (quantified together) emission rates at around 5300 % and 1850 % from infested trees depending on the time since infestation (Table 2). This could lead to potentially increased SOA yields when forests are subjected to bark beetle outbreaks. The MTs limonene and myrcene were slightly below the SQTs in ranking of SOA yield, and according to our findings, they were seen to increase with an average of 4400 to 16900 % and 1900 to 49900 % respectively, depending on time since infestation, where the highest increase was in the early season. Myrcene had the third highest percentage increase for the seasonal average of all compounds, and appeared in around 80 % of all samples from infested trees, compared to about 20 % from the healthy tree samples. This change in compound blend could potentially lead to large impacts on SOA yield from bark beetle infested trees overall.

In the comparison of the initially stressed tree and the initially healthy tree, it was apparent that there initially was no significant difference in the total emission rate, but different compound blends were emitted. Linking this to SOA, the higher emissions of limonene and myrcene from the initially healthy tree early during the infestation indicates that higher SOA yields might come from healthy spruce trees when infested. The high emission rates were also seen to continue until the trees were taken down in August which implies that potential increase in SOA yield might continue for longer. Compounds unique to both infested trees were emitted as well, where the stressed tree emitted bark beetle related pheromones and the initially healthy tree emitted a broader blend of MTs, these individual compounds might play a role in the SOA yield as well. As the bark beetle infested trees generally would impact SOA yields, more attacks on healthy trees might further affect the atmospheric processes, specifically production of SOA.



**4.2 Bark beetle induced BVOC emissions in relation to other stresses, leaf emissions and modelling**

Significant increases of the temperature standardized BVOC emissions of Norway spruce bark of up to around

22000 % for the total group of MTs were seen when trees were infested by bark beetles early in the season. This high increase in emission rate from insect stress for Norway spruce has not previously been observed according to the review by Yu et al. (2021), in which the highest recorded increase was around 2000%, including previous studies on *Ips typographus*. Heat stress from higher temperatures did also not increase BVOC emissions as much as stress from bark beetles. A study on Norway spruce with higher air temperatures of 40 °C found that BVOCs

increased by 175 % compared to emission rates at air temperature of 30°C (Esposito et al., 2016), not comparable to the increase found from bark beetle infestation. However, the impact of combined stresses from temperature and insect attacks might further increase the BVOC emissions. In our study we found the temperature sensitivity to change when trees become infested. Our analysis of the $Q_{10}$ coefficient showed that the coefficient increased for all compounds but one (of 15, namely p-cymene) as temperatures increased, indicating that the emission rates

would increase if the temperature increased. The combination of bark beetle stress with increased temperature might thus lead to even higher increase in emission rates. This is however not the focus of this study, but as we found the BVOC compounds temperature sensitivity to increase, and as we found bark beetle infestations to increase bark BVOC emissions more than any other comparable stress, there might be high influences of BVOC emissions from combined stress, making it important to account for when modelling the emissions.


The increase of bark emission rates we found in this study are high enough to considerably add to the emission rates of a full tree when comparing with emission rates from needles – which is considered the part of the tree with the highest emission rates. When modelling the emission rates, two approaches were used for the bark MT emissions, the G93 algorithm and the $Q_{10}$ approach. The results showed similarities in pattern but the $Q_{10}$ approach

had larger increases in emission rates with higher temperature increase, something that was expected. The G93 modelled emission rates were constantly higher than the $Q_{10}$, and had less variability, which might be explained by the empirical coefficients used in the G93 compared to only taking temperature into account. For the needle emission rates, only G93 was used because of the light dependent nature of some BVOCs emitted from the needles that could not be explained by the temperature in the $Q_{10}$ approach only. The seasonal average emission rates from

spruce needles were measured during 2017 in Hyltemossa in a study by van Meeningen et al. (2017). As the study was conducted at the same site, their results were applied to our study as a comparison of bark BVOC emission to needle emission. It was clear that constitutive MT emissions from bark does not compare to the needle emission (Fig. 11), where the seasonal average of the MT emissions where 11 times higher from needles than bark. However, when comparing seasonal average bark MT emission from infested trees with needle emissions, it was the other

way around, the bark emissions from infested trees where 6 to 20 times higher than the needle emissions depending on the time of season. The constitutive MT emission from bark of healthy spruce trees accounted for 8 % of the total emission rates from bark and needles. However, if there was an ongoing infestation from bark beetles, the bark emission rates would account for 95 % of the total emission rates during the early season, and 85 % during the late season. The infestation would increase the total emission rates from bark and needles by 550 to 1900 %,

depending on the time in the season, when comparing with the seasonal average of the constitutive emission rates.

When a tree is infested, the emission rates increase significantly which can cause large local effects both for tree health but also SOA production. The BVOC emission increase also cause more widespread effects, if the outbreaks are sustained at high levels, there would be large impacts regionally. During 2020 in Sweden, 8 million m³ forest

was affected by spruce bark beetles (Wulff and Roberge, 2020). This represents about 0.7 % of the total volume



of Norway spruce trees with a diameter larger than 15 cm in Sweden (Skogsstyrelsen, n.d.). Using the seasonal average from the early season and late season of the bark beetle infested emission rates of MT found in this study and the needle emission rates from van Meeningen et al. (2017), the infested trees during 2020 would contribute to an increase of about 4 to 13 % of total MT emission rate from Norway spruce trees in Sweden, including

emissions from canopy and stem. The effects from insect herbivory and specifically spruce bark beetles might thus be underestimated both in emission and vegetation models (MEGAN, LPJ-GUESS; Guenther et al., 2006; Schurgers et al., 2009) and atmospheric chemistry models estimating BVOC impacts on oxidation capacity and SOA formation (ADCHEM; Roldin et al., 2011).

**5 Conclusion**

Norway spruce trees are emitting BVOCs from the bark as a stress response to spruce bark beetles, and as the number of spruce bark beetle outbreaks increase, it will impact the total emission of BVOCs. The aim of our study was to examine how spruce bark beetles affect the BVOC emission rates from Norway spruce bark by looking at the difference between healthy and infested trees, the time passed since infestation and the difference in emissions from different bark beetle hole types. We also wanted to provide an insight into how the BVOC emissions change

from non-infested to infested, and following the infestation over time. Our study shows that there is a significant difference in BVOC emission rates from healthy spruce bark and infested spruce bark, but also a relationship between BVOC emissions from infested trees and the time passed since infestation start, which can be supported by our result that indicated a difference in emissions from bark beetle drilled entry holes and exit holes. We also saw that the initiation of a second generation of bark beetles, which can lead to late summer attacks, can potentially

have prolonged impacts on the BVOC emissions as we found emission rates to be induced until the start of the next season. When the tree was infested again, the emission rates was further induced to reach the same levels as the induced emissions of a tree that was healthy before infestation. As the infestation proceeded, we saw a difference in the emission rate and compound blend when comparing the initially stressed tree with the initially healthy tree, where the emission rates were induced to high levels until August for the initially healthy tree, but

not for the initially stressed tree. The entire impact of spruce bark beetles on Norway spruce trees would require further studies, and the importance of such studies is supported by our findings that the bark beetle induced BVOC emission rates can be considerably higher than previously thought and could potentially increase the total MT emissions from Norway spruce in Sweden with 4 to 13 %. Even further work would be needed in investigating the impact of coupled stress factors. We found a potential link between temperature stress and bark beetle stress, where

trees seems to become more sensitive to temperature with a potential to have even higher emission rates when temperatures increase in conjunction with bark beetle infestations. We believe that bark beetle infestations can have higher impacts on the atmosphere and climate change than previously thought and samples from more trees and more frequently throughout the season is needed in order to fully understand the impact.






**Appendix A**

**Table A1. All identified compounds from all samples throughout the season separated into healthy trees and infested trees. Presented is the seasonal average emission rate (µg m⁻² h⁻¹ ± one standard deviation) for each compound and the groups of MT, SQT and other BVOCs and the occurrence (%) of each compounds for all samples. The increase (%) is presented for the infested trees as an increase from healthy to infested. The compounds that where identified but unable to quantify is presented as n.q. (no quantification).**


| Compound name | Healthy | | Infested early season | | | Infested late season | | |
| --- | --- | --- | --- | --- | --- | --- | --- | --- |
| | *average ± std ($\mu g\ m^{-2}\ h^{-1}$)* | *occurrence (%)* | *average ± std ($\mu g\ m^{-2}\ h^{-1}$)* | *increase (%)* | *occurrence (%)* | *average ± std ($\mu g\ m^{-2}\ h^{-1}$)* | *increase (%)* | *occurrence (%)* |
| **Monoterpenes** | 29.37 ± 51.01 | | 6690 ± 6860 | 22678.35 | | 1970 ± 1310 | 6607.52 | |
| alpha-Pinene | 11.49 | 76.11 | 911.14 | 7829.85 | 100.00 | 824.64 | 7077.02 | 100.00 |
| beta-Pinene | 8.22 | 55.75 | 954.17 | 11507.91 | 100.00 | 225.28 | 2640.63 | 100.00 |
| 3-Carene | 2.48 | 48.67 | 285.16 | 11398.39 | 100.00 | 33.07 | 1233.47 | 95.45 |
| Limonene | 1.89 | 44.25 | 320.82 | 16874.60 | 87.70 | 85.43 | 4420.11 | 100.00 |
| p-Cymene | 0.49 | 39.82 | 241.45 | 49175.51 | 62.50 | 53 | 10716.33 | 77.27 |
| beta-Myrcene | 0.32 | 17.70 | 159.76 | 49825.00 | 78.95 | 6.3 | 1868.75 | 86.36 |
| beta-Phellandrene | 2.7 | 10.62 | 673.13 | 24830.74 | 43.75 | 189.47 | 6917.41 | 68.18 |
| (1S)-Camphene | 1.7 | 6.19 | 1516 | 89076.47 | 75.00 | 388.82 | 22771.76 | 100.00 |
| 2-Cyclopentylcyclopentanone | n.q. | 4.42 | - | - | - | n.q. | - | 4.54 |
| alpha-Terpineol | n.q. | 2.65 | - | - | - | - | - | - |
| 5-Ethyl-m-xylene | n.q. | 0.88 | - | - | - | - | - | - |
| (+)-Sabinene | 0.08 | 0.88 | 206.37 | 257862.50 | 43.75 | 2.93 | 3562.50 | 4.54 |
| (1R)-Camphene | - | - | 186.16 | - | 18.75 | 112.32 | - | 77.27 |
| Tricyclene | - | - | 173.42 | - | 50.00 | 24.52 | - | 36.36 |
| Eucalyptol | - | - | 10.25 | - | 43.75 | 2.32 | - | 27.27 |
| (+)-Camphor | - | - | - | - | - | n.q. | - | 45.45 |
| Pinocarvone | - | - | n.q. | - | 43.75 | n.q. | - | 4.54 |
| 4-Carene | - | - | 348.15 | - | 37.50 | - | - | - |
| zeta-Fenchene | - | - | 116.96 | - | 25.00 | 0.92 | - | 4.54 |
| alpha-Phellandrene | - | - | 114.43 | - | 31.25 | - | - | - |
| (1R)-(-)-Myrtenal | - | - | n.q. | - | 31.25 | - | - | - |
| trans,trans-Allocimene | - | - | 50.9 | - | 25.00 | - | - | - |
| 5-Vinyl-m-xylene | - | - | n.q. | - | 25.00 | - | - | - |
| 3-Pinanone | - | - | - | - | - | n.q. | - | 13.63 |
| Norbornane | - | - | 60.27 | - | 18.75 | - | - | - |
| gamma-Terpinene | - | - | 88.84 | - | 12.50 | - | - | - |
| alpha-Fenchene | - | - | 14.07 | - | 12.50 | - | - | - |
| 2-Carene | - | - | 156.65 | - | 12.50 | - | - | - |
| alpha-Thujene | - | - | 15.83 | - | 12.50 | - | - | - |
| Verbenone | - | - | n.q. | - | 12.50 | - | - | - |
| Myrtenal | - | - | n.q. | - | 6.25 | - | - | - |
| alpha-Terpinene | - | - | 26.06 | - | 6.25 | - | - | - |





| | | | | | | | | |
|---|---|---|---|---|---|---|---|---|
| **Sesquiterpenes** | 2.12 ± 3.17 | | 53.0 ± 74 | 2400.00 | | 18 ± 24 | 749.06 | |
| Longifolene+beta-Caryophyllene | 0.7 | 17.70 | 37.65 | 5278.57 | 56.25 | 13.6 | 1842.86 | 54.54 |
| alpha-Humulene | 1.42 | 47.79 | 4.86 | 242.25 | 31.25 | 4.52 | 218.31 | 18.18 |
| Germacrene D | - | - | 3.86 | - | 18.75 | - | - | - |
| Isoledene | - | - | 3.25 | - | 18.75 | - | - | - |
| beta-Cubebene | - | - | 3.2 | - | 18.75 | - | - | - |
| **Other BVOCs** | 0.40 ± 0.85 | | 3.36 ± 6.69 | 740.00 | | 0.14 ± 0.20 | -65.00 | |
| Isoprene | 0.4 | 58.41 | 3.36 | 740.00 | 62.50 | 0.14 | -65.00 | 27.27 |
| Decanal | n.q. | 45.13 | n.q. | - | 12.50 | n.q. | - | 13.63 |
| Benzene | n.q. | 45.13 | - | - | - | n.q. | - | 13.63 |
| Nonanal | n.q. | 38.94 | n.q. | - | 6.25 | n.q. | - | 13.63 |
| Toluene | n.q. | 21.24 | n.q. | - | 6.25 | n.q. | - | 9.09 |
| 1,3,5-Trifluorobenzene | n.q. | 14.16 | - | - | - | - | - | - |
| Benzaldehyde | n.q. | 11.50 | - | - | - | n.q. | - | 9.09 |
| Butyl formate | n.q. | 7.96 | - | - | - | n.q. | - | 4.54 |
| Caprolactam | n.q. | 7.08 | - | - | - | - | - | - |
| Cyclopentanone | n.q. | 5.31 | - | - | - | n.q. | - | 9.09 |
| Methanesulfonic anhydride | n.q. | 5.31 | - | - | - | n.q. | - | 4.54 |
| Trimethylbenzol | n.q. | 1.77 | - | - | - | - | - | - |
| m-Xylene | n.q. | 1.77 | - | - | - | - | - | - |
| Ethylhexanol | n.q. | 1.77 | - | - | - | - | - | - |
| Acetic acid | n.q. | 1.77 | - | - | - | - | - | - |
| tert-Butylamine | n.q. | 0.88 | - | - | - | n.q. | - | 9.09 |
| m-Ethyltoluene | n.q. | 0.88 | - | - | - | - | - | - |
| o-Ethyltoluene | n.q. | 0.88 | - | - | - | - | - | - |
| Methyl 3-hydroxy-2,2-dimethylpropanoate | n.q. | 0.88 | - | - | - | - | - | - |
| 1-Pentene | n.q. | 0.88 | - | - | - | - | - | - |
| Butanal | n.q. | 0.88 | - | - | - | - | - | - |
| 1-Nonene | n.q. | 0.88 | - | - | - | - | - | - |
| Isobutenyl methyl ketone | n.q. | 0.88 | - | - | - | - | - | - |
| Diacetone alcohol | n.q. | 0.88 | - | - | - | - | - | - |
| Furfural | n.q. | 0.88 | - | - | - | - | - | - |
| 1,6-Anhydro-beta-d-talopyranose | n.q. | 0.88 | - | - | - | - | - | - |
| dl-3,4-Dehydroproline methyl ester | n.q. | 0.88 | - | - | - | - | - | - |
| 6,10,14-Trimethyl-2-pentadecanone | n.q. | 0.88 | - | - | - | - | - | - |
| Undecanal | n.q. | 0.88 | - | - | - | - | - | - |
| Carbon disulfide | n.q. | 0.88 | - | - | - | - | - | - |
| 2-Methyl-1-phenylpropene | - | - | n.q. | - | 37.50 | n.q. | - | 9.09 |
| alpha,alpha-Dimethylallyl alcohol | - | - | n.q. | - | 18.75 | - | - | - |
| Benzoic acid | - | - | - | - | - | n.q. | -65.00 | 9.09 |
| Acetophenone | - | - | - | - | - | n.q. | - | 9.09 |
| Methyl acetate | - | - | - | - | - | n.q. | - | 9.09 |
| (-)-Bornyl acetate | - | - | n.q. | - | 12.50 | - | - | - |
| Bornyl acetate | - | - | n.q. | - | 12.50 | - | - | - |



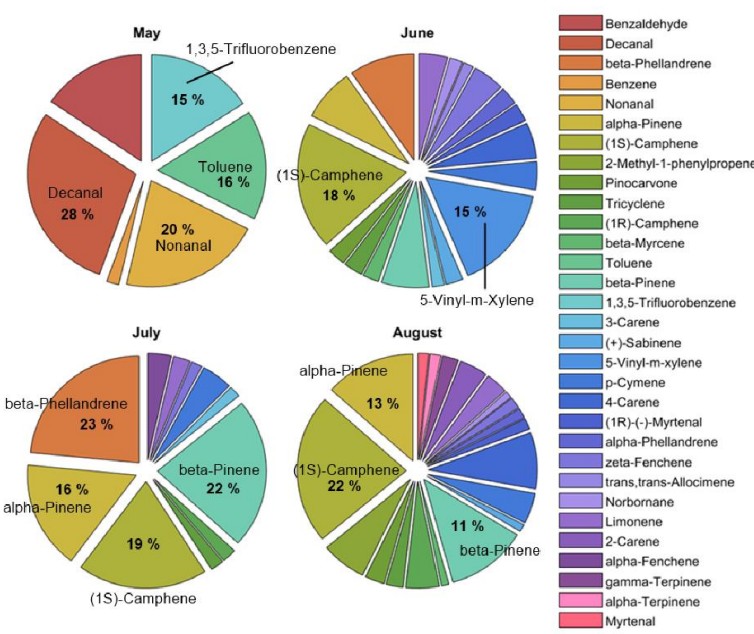


**Figure A1: The daily average blend from the spruce S3S3 and percentage contribution in mass throughout the summer (May, June, July and August), showing only compounds with a mass contribution of at least 1 %.**

**Table A2. The difference between healthy and infested trees when applying the calculations for the $Q_{10}$ temperature dependency. The reference emission rate at 30°C ($F_0$) entails higher emission rates at 30°C with a higher number. The $Q_{10}$ coefficient indicates the emission rate sensitivity to temperature, leading to higher emission rates at higher temperatures when the coefficient is larger.**

| | $F_0$ (µg m$^{-2}$ h$^{-1}$) | | $Q_{10}$ | |
| --- | --- | --- | --- | --- |
| **Compound name** | *Healthy* | *Infested* | *Healthy* | *Infested* |
| **Monoterpenes** | | | | |
| beta-Pinene | 11.0 | 34901.4 | 7.8 | 981.6 |
| (1R)-Camphene | - | 1503.0 | - | 80.2 |
| beta-Phellandrene | 20.6 | 1240.0 | 2.8 | 37.8 |
| alpha-Pinene | 55.0 | 879.5 | 18.7 | 18.8 |
| (1S)-Camphene | 92.6 | 470.1 | 6.3 | 14.3 |
| beta-Myrcene | 15.8 | 248.2 | 56.7 | 167.6 |
| Limonene | 12.7 | 116.4 | 11.6 | 16.9 |
| 3-Carene | 4.6 | 111.4 | 6.6 | 25.8 |
| p-Cymene | 15.7 | 86.3 | 22.4 | 14.6 |
| Tricyclene | - | 76.3 | - | 13.6 |
| (+)-Sabinene | - | 66.3 | - | 7.5 |
| Eucalyptol | - | 2.5 | - | 3.5 |
| **Sesquiterpenes** | | | | |
| Longifolene+beta-Caryophyllene | 0.01 | 23.6 | 0.1 | 33.1 |
| alpha-Humulene | 0.01 | 0.5 | 0.7 | 1.3 |
| **Other BVOCs** | | | | |



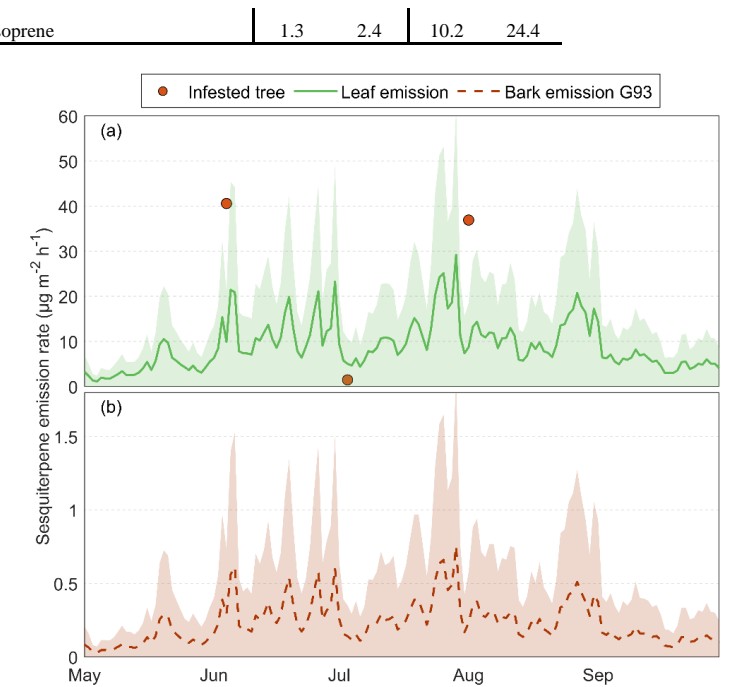


**Figure A2. The modelled (b) constitutive bark VOC emission rates from the group sesquiterpenes based on the tree temperature taken at 3 meters height in the north and east orientation, data taken from the HTM ICOS station (Heliasz, 2020). The bark emission (black) is calculated based on the Guenther algorithm (Guenther et al., 1993) based on measured emission rates in this study. The (a) leaf emission rates (green) are calculated based on the Guenther**
**algorithm and the measured emission rates are taken from van Meeningen et al., (2017) and specific leaf area (SLA) was taken from Wang et al., (2017) using the air temperature at 24 meters taken from the HTM ICOS station (Heliasz, 2020). For comparison, the actual measured bark VOC emission rates from one infested tree over time, from this study, is included in the Figure (red dot).**


**Author contribution**

EJ and TH designed and planned the campaigns. EJ performed the measurements. EJ performed the data analysis with contributions from KL, AG and AMJ. Funding was acquired by TH. EJ prepared the manuscript draft with
contributions from all co-authors.

**Competing interests**

The authors declare that they have no conflict of interest.

**Acknowledgements**

The authors would like to thank the ICOS Hyltemossa and Norunda staff for logistical support and Gustafsborg Säteri AB and Mats de Vaal for support with forest sites and tree selection. We would also like to thank Julia Iwan,




Marieke Scheel, Tanja Sellick and Emily Ballon for assistance with field measurements and Cleo Davie-Martin for valuable discussions on sample analysis. The research presented in this paper is a contribution to, and was supported by the Strategic Research Area Biodiversity and Ecosystem Services in a Changing Climate, BECC (BECC.LU.SE), funded by the Swedish government.

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
