# Peer review of "Spruce bark beetles (*Ips typographus*) cause up to 700 times higher bark BVOC emission rates compared to healthy Norway spruce (*Picea abies*)"

_Biogeosciences, 2022_

## Referee Comment (RC2)

**Spruce bark beetle (*Ips typographus*) infestation cause up to 700 times higher bark BVOC emission rates from Norway spruce (*Picea abies*)**

Biogeosciences: bg-2022-125

This study is interesting and has its own important value to supplement what has been done previously and giving new information about bark beetle infestation effect to BVOC emission from Norway spruce bark. A lot of confusing information and weak Discussion is not helping to reveal this experiment value. Important part is also the new emerging volatiles, but these get so little attention or no attention in Discussion part. Discussion part is weak and needs more improving, more references and discussion that is connected with result and putting the results in bigger picture. I suggest leaving out the comparison with leaf emission; this does not give an extra value to the study. This has opposite effect: to diminish your efforts.

Through MS: commas are there where they are not supposed to be and at the same time missing where they should be. I suggest to ask some native speaker to check English (my English is also not so good, but based on my experience I guess I understand when English needs improving). A lot of using %, this does not make the MS better. Try to find better alternative for this to describe results. Important part is also the new emerging volatiles, but these get so little attention or no attention in Discussion part. Discussion part is weak and needs more improving, more references and discussion that is connected with result and putting the results in bigger picture. I suggest leaving out the comparison with leaf emission; this does not give an extra value to the study.

1) Title says: "...cause up to 700 times higher emission..." – but compared to what?
2) L14 – drilled – used one style trough MS: drilled or boring; instead "entry holes and exit holes" use "entry and exit holes", check whole MS
3) L15 – instead "healthy trees and infested trees" used "healthy and infested trees"; write out site names in abstract
4) L21 – decreasing relationship – is it statistically true?
5) L28 – " This study…." This kind if sentence fits better to Conclusions
6) L54 – "..SOA." – add reference
7) L55-56 – better use "negative or a positive feedback loop..."
8) L57-58 – form a better sentence
9) L63 – any references from last 10 years? If yes, add.
10) L10 - any references from last 10 years? If yes, add.
11) Trough MS – I suggest to use Norway spruce instead of (just) spruce when related to your experiment and *xyz* spruce when related to specific reference, if possible
12) L78  - I guess it's not poorly understood, because every experiment gives valuable information, but of course depending of the aims
13) L81 – Connections with what? Some words missing?
14) L82 – For me the bark emission comparison with needle emission is not giving this MS much more value. Needle emission and bark emission measurements from spruce should be standalone experiment.

15) L84-90 – I suggest not to use we/our etc for this part (check also trough MS). For the reader would be helpful to find hypotheses easily when they are numbered 1,2,3 etc and hypotheses presented clearly and properly.
16) L87 –"...eventual death of the tree" – I assume you did not measure the tree until it's eventual death. How long does it take when the tree is dead or considered dead after bark beetle infestation?
17) L89 – "total number of holes" – what do you mean in "total number of holes"? Entry+exit holes?
18) L91 – You did not investigate blue stain fungi in this experiment – I suggest to use blue stain fungi in discussion part.
19) L92 – "tree individuals" – I suggest "two trees"
20) L99 – I suggest that in this sentence indicate table 1 for campaign dates
21) L96 –Based on this sentence I would expect that in a sub-study where more than one campaign but it's written over time. For me it indicates more than one campaign.
22) L101 – use one style: sub-study or additional campaign trough MS
23) Figure 1 – how are the rules for figure legends, is there a dot or semicolon after Figure 1? What is the rule for figure legend text - needs to be in **bold**? I like the map for showing, but describing b-c-a is not common in figure legends also in text (like Fig.1b). If possible, change maps etc. starting with abc and in figure legends also describing abc. Missing proper explanations what yellow squares and circles mean. "The Figure..." –why is there **F**igure not **f**igure? If possible, write out in the legends which site had healthy and infested trees, and where was sub-study done.
24) L122 – "by visual examination" – meaning no entry holes?
25) L123 – Indicate site name
26) L120-125 – write out here which was here considered as sub-study
27) In Methods part, please consider clearly explaining healthy, infested and control, because later you show also control measurements and use the same style trough MS.
28) L130 – check names and ALL NAMES in MS must be correct, many mistakes: alpha-pinene, α-pinene, beta-pinene, β-pinene etc and other names written not well.
29) L130 – Typosan- company name?
30) L132 – repeatedly: measurements dates are indicated where?
31) L136 – indicate figure
32) L137 – use one style trough MS 5°C to 22°C or 5° to 22°C
33) L138 – start with May and then with June
34) Figure 2 – in figure legends start describing with temperature as you have on the figure temp, precip, campaign. Figure legend text in bold? I suggest writing out in a better way where data was taken.
35) L147 – pump company? I guess L/min is visually better than lpm. 0.6-0.9 L.
36) L148 – "Ad sorbent tubes…" goes to VOC collecting part
37) L153 - 40°C or 40 °C – use one style trough MS
38) L155 – I suggest here to use diameter, not inches; PTFE-tubing or PTFE-lines?
39) L158 – "each sample" – do you mean here each VOC collection?
40) L160 – so the first VOC measurement was at 8:00 in the morning or you started to put the chamber bases on tree?
41) L166 – Can you add the info when the infestation (swarming time) was detected. "For plot 3" indicate plot name. "…witnessed on site." – when?

42) L168 – based on data from Skogsstyrelsen database, when was the proposed swarming time?
43) L170 – Did you mark the entry/exit holes to distinguish new and previous entry/exit holes?
44) "By looking at bark photographs…" – I suggest to use "photos" and it's the first time to indicate figure 4, so, in the end of sentence, indicate figure, not in the next sentence.
45) L173 – Rewise sentence – I suggest not to start the sentence with "table 1".
46) Figure 3: On figure you have VOC filter, but in Methods part there is no info about VOC filter. Is it the same as nets? "A photograph of how the setup looked in the field is displayed to the right (b)." – I suggest revising. "The BVOC samples were.." - add here the pocket pump also. Add tree name or latin name.
47) Figure 4: which tree – add name or latin name. Correc to " entry and exit holes". Photos where done by who? Drilled or bored?
48) Table 1 – text in bold? Which infested tree –add name. Which site/area trees belong? Table paragraph spacing. "exit and entry holes". There is no explanation what S1S1 etc mean? "Type majority"-> "Hole type majority". If possible add spruce initial health status also – healthy, infested etc.
49) L190 – I suggest to add in this paragraph how much healthy and infected tree BVOC measurements were done in total
50) L191 –Tenax company? Carbograph company?
51) L194 – Why was the collecting time 5 to 6 L not exactly 5 L or 6 L?
52) L196 – "twice a day" – can you say at least before mid-day or after mid-day? Or when?
53) L197 – "at the start of the BVOC sampling and at the end"-> "at the start and at the end of the BVOC sampling"
54) L198 – "After sampling.." -  does this sentence repeat the previous sentence? "During sampling period"?
55) L205 – He, which company He?; VOC or BVOC – use one style trough MS
56) L210 – space between number and m/mm
57) L220 – Shimadzu… - country?
58) 2.4 – ALL equations need numbers and indicated in the MS where necessary
59) L225 – in equation the A is normal, but in describing text it is *A* (italic)
60) L243 – For me you have three ways to describe 1) 100 days before/after; 2) majority entry and /or exit holes; 3) early and late season. And all these three are used trough MS. I suggest to explain this well in this paragraph and use one style trough MS and also on figures and figure legend text.
61) L256 - Q in italic?
62) L258 – F in italic?
63) L262 – F and Q in italic?
64) 269 –Eq 4 – where, which one?
65) L270 – Eq 5 – where, which one?
66) L270 – I suggest explaining in Methods part: does constitutive mean healthy or control tree emission. And what is considered as control, because later on the figures you have a control.
67) L274 – 3 m
68) L275 – North, East? north? east?  - Look trough MS that weather arc are correctly. I suggest that weather arcs info goes in BVOC collecting paragraph.
69) L276 – I'm not so fond of comparing bark emission with leaf emission. This should be extra study to compare with more measurements campaigns. For me this comparison does not give an extra value to this study (overall). For me the core of the study is bark emissions from infested and

healthy trees and then finding important results to discuss on Discussion part and with SOA and climate change and modelling (MEGAN etc).

70) L282 – outliers from which sites/area? I suggest moving this paragraph closer to BVOC sampling paragraph. "Photography"->"photos"

71) 2.5 Statistical studies – Needs a lot improving. In here also clarify which measurements, from what area/sites, what about constitutive etc, what about sub-study? Look my comment before. P in italic (*p*)? Same with F and Q. Clearly write out what did you compare and with what statistical test and which program did you use for it.

72) Should there be dot in each main/second headline, like 3. Results/ 3.1. zyx ?

73) 3.1. – Revise the tile if clarifying "constitutive" in previous MS parts

74) L294 – constitutive=healthy? Is it temperature-standardized or temperature standardized? Use one style.

75) L296 – indicate the results in table X if they are written out in table.

76) L301 - The numbers here and on the figure are not well trackable/comparable and at the same time quickly understood on the figure. If I read the sentence and look at the same time on the figure, cannot distinguish quickly what is what and figure is hard to read for a reader who does not read the results part about this figure.

77) L303 – p-value? *p*>0.3. Make changes through MS.

78) L304 – which sites in what area?

79) Add on Figure 5 HTM, so the reader can understand what area without reading figure-describing text. In describing text, indicate statistical test. What n means? What black circles mean?

80) L311 – infested trees in which area?

81) L312 – drill or bored? "The season…" – you were describing it before in Methods part, in here use one style to indicate, but not repeating the same as in Methods. Look my comment before.

82) L314 – I suggest to add total emission also to Table 1.

83) L315 – "..µg m-2 h-1 . MTs…" check space. " MTs…" indicate for table 2.

84) L319 – 42% or 42 %? Use one style thorough MS.

85) L322 – Which trees: name? Which area? Name? BVOC or VOC?

86) L323 – healthy from which area?

87) L328 – infested tree: I suggest to use infested bark samples, bark emissions, infested tree bark etc through MS, one style through MS.

88) 329 – "isoprene, (n = 17) compared " check the mistake. Infested trees in which area?

89) L331 – indicate table

90) L337 – Which areas/sites? P value – how come 0.00???? "The median…" - Consider writing this sentence according to this what is shown on figure or change figure according to the sentence. I advise to start with healthy and then infested (early, late). I suggest to use less "respectively" through MS, not so easy to follow.

91) Figure 6 - please add also HTM or NOR on the figure. Which tree? Again use one style for early/late season, exit/entry. "The emission rate.." sentence doesn't belong to figure describing text. P-value?? Standardized to temperature – indicate equation in XX part. Explain yellow and blue. Statistical test used?

92) L349 – healthy  and infested. Which sites/area???

93) L315 – I suggest rounding the numbers to proper value in table, visually better. Overall, it's kind of disturbing using so much % through MS.

94) Check VOC names trough MS.

95) L287 – Again, using % and even so big numbers….do you use 22678 or 22 678? Which area, which site trees? Which trees?

96) L374 – when it's the first time indicating Appendix, write out Appendix, otherwise what Table A1 means? trans, trans-alloocimene - > maybe (4E,6E)-alloocimene  better?

97) L375 – gamma-terpinene or γ-terpinene? "in infested…"- revise end of the sentence, double repeat

98) Table 2 – paragraph spacing. Which tree? Which area? STDEV is only for groups, not for each compound. Explain MT, SQT. What about decrease (-65%), not explained, only increase. Again numbers: 22678 or 22 678? Check all the names in table, all the names need to be used correctly trough MS (tables, figures, etc). If possible add total emission.

99) 386 –Which area? Which trees? Number 13000?

100)      L390 - "...daily average standardized emission.." – check

101)      L391 – "The average number of holes per square meter bark area found in this study was based on the values in Table 1." Revise or delete.

102) L396 – Whole sentence "For…." – check and correct. Are here the results of sub-study?

103) L400 – "The TS emission..." – I guess this sentence is to long for explanation, I suggest to divided in two parts and describing clearly. I guess this kind of description goes to Methods part in Statistics.

104) Figure 7 – Which trees? What area? S3S2, S3S3, etc meaning? Early/late, 100 days before/after. "The emission rates…" and "This is evident…" sentences do not belong to figure describing text.

105) 3.2 – Did you to statistics to evaluate influence of time?

106) L413 – Which site/area? Indicate table/figure.

107) L415 – Where is 3.2.3? I suggest to put function to Statistics part and indicating on the figure describing text where to find it.

108) L420 – use one style trough MS: 8b and 8d or 8b,d etc.

109) L421 – I guess "slightly" is not correct to use here. "..depending..." revise, double repeat.

110) L425 – How long does it take from infestation to total damage/death to the tree? Maybe something to discuss in Discussion and some info for introduction part?

111) L428/429 – indicate figure. Again constitutive/healthy emission? Area?

112) L432 – "The tree measured.."- did it have a name SxSx? What area?

113) L434 – indicate figure

114) L435 – Too long sentence. Revise. Make it easier to read. And indicate figure.

115) Figure 8 – VOC names!!! Which trees? What area? "..days  since..". Correclty put abcd to figure describing text, (a) is in a wrong place and (d). Blue is for what? Did you include yellow marked tree also in the fitting? Revise sentence for clearer understanding. What test is used here? In Methods part there is no indication about measurements after 300 days.

116) L445 - entry

117) L450 – indicate tableX?

118) L452 – check names!

119) L453 – 9b,d – look my comments before.

120) L455 – indicate table

121) Figure 9 – check names. Which area. Change the order top of the figure: first are entry holes and then exit holes. Relationship p value? Which test was used? Which trees? Again (a), (b), etc to correct order. All or total BVOC. The "2" on the x-scale title is to big. "High…" sentences does not belong to figure describing. Again: exit/entry, late/early, 100 days before/after. "There is a disctinion.." – revise.

122) 3.4  - sub-study?

123) L466 – Revise sentence "At plot 3…" and indicate figure. On another way L466-L469 is describing what you said before, I suggest revise this sentence. Which area?

124) L470- P in italic.

125) L471 - Control trees – what does the control trees mean in MS? Control - not at all infested during campaign or beginning of campaign not infested and later experiencing bark beetle infestation? Indicate figure 8 correctly.

126) L473 –Which other trees? Other trees in plot 3 or somewhere else? Indicate a tableX?

127) L576 –" a difference.." indicate table if possible

128) L477 – check names; I suggest to remove 800 ug or use another number for comparing.

129) L479 -  check names

130) L480 – indicate figure, I suggest that p<0,03. Delete "and close to zero", check names

131) L483 – p italic

132) L484 – indicate figure

133) L485-495 – Again less % or even when presenting choose important results, because this part is interesting.  I suggest this part to discuss in Discussion part. Check names. "..by mass" – what mass?

134) Figure 10 - Months names ith capital letter. On the figure, where is asterix-what does it indicate? Can't be seen. Check names! All figure plot need to have month names. Describing text what trees, what area. Why no SE? Add tick marks on the figure, other figures have. Remove the outer box from top of the figure, where are indicated control, S3 and S2 and add some space between them for clearer separation. Start with control in describing text, then healthy or infested etc. Add year in describing text. "For", I guess "from". In this figure, there is important info, but this does not come forward clearly because of figure negative sides. Maybe another alternative?

135) L506 – F in italic? Q in italic? Look through MS

136) L507 – which area?

137) L508 – indicate a table

138) L510-511 – double repeat

139) L513 – 34900

140) L510-520 – too much %

141) L521-  coefficient?

142) L526 – p<0.05 and p<0.01

143) 3.6 – As is said before: for me leaf emission does not give an extra value for the study. Maybe to leave this part out and present the result what are in this chapter? Again which area?

144) L537 – Ms or $M_s$?

145) L558 – well?

146) Figure 11 – pleases start with abc, revise figure then. Which tree, which area? Again: north, east, North, East? Meters -> m. Which ONE infested tree (SxSz?), which area? What green and purple shades are representing? Add tick marks on y-scale.

147) 4. Discussion – needs a lot improving. Right now it's weak. Indicate table and figures where proper. Check names. What areas? More references and discussions. Somehow Discussion part repeats Results part, would be better to have more discussion about results.

148) L587 –what relationship? How did you calculate this? Assessing how? Based on what? No info in Methods. Any references about lichen cover related to isoprene emissions. Ghimire – how did they assess? Statistically or also visually?

149) L591 – healthy and infested trees – what previous findings? References? Some examples for describing? Indicate figure.

150) L591 – repeating of the results. I suggest to put these number into comparison with some references if possible or to discuss about emissions. I suggest not to use "we".
151) L549 - And so what it is higher at the beginning? Connect the higher emission with atmospheric processes, add references. What is different in the beginning? Indicate your figure/table if proper.
152) L583 – 605: all compared to one reference.
153) L599 - time...what about temperatures, weather conditions? And what time, measuring time during day (morning, lunch time, evening) or campaign times (certain dates).
154) L600 – "study…" indicate figure
155) L601 – Ghimire – which holes in this reference?
156) L602 – "this allowed…" then what is done before? References? What stage to what stage?
157) L604 – "This finding.." – what finding exactly? Why it is important in bigger picture?
158) L605 – "We could see..."- is it really a trend or rather something normal to be expected when more time passes since infestation.
159) L608 – check space
160) L609 – references, more to discuss with few sentences?
161) L611 – indicate figure
162) L612 – which total holes, majority of entry or exit holes? Or entry+exit? To discuss what could be the possibilities for no pattern. Any references where this kind of pattern occurs?
163) L614 – indicate figure
164) L616-617 – references. Too long sentence, I suggest to make into two sentences. Double repeat.
165) L618 –VOC`
166) L619 – which individual tree, which area?
167) L620-629 – References.
168) L624 – All trees – what area?
169) L628 – any reference to support your results and conclusion?
170) L631 – indicate table/figure if proper
171) L630-639 – add references and something to discuss more than these two references
172) L635 – oxygenated monoterpenes?
173) L639 – any references to add?
174) L640 – 4.1.1.- Only 3 references per this part?
175) L682 – what other 16 BVOCs? Revise, no need to add BVOC names exactly.
176) L700-  "As the…" double repeat.
177) 684 and 690 – both sentences are a repeat. " This could.." and "This change.."
178) L685 – Myrcene sentence goes rather to Results part.
179) 4.2 – again % and numbers. Now here Ips typographnus – why not bark beetle as trough MS? Only 5-6 references per this part?
180)  Table A1 – Names: numbers; MT, SQT explanation – check comment for table 1.
181)

---

## Author Comment (AC1)

**Response to the comment of Anonymous Referee #1**

We appreciate the time and effort from Referee #1 to provide comments and great suggestions on our paper. We address each comment below, where the reviewer's comments are shown in italics. The line numbers refer to the original document.

*Comments*

*This study investigates the effect of bark beetle attack on Norway spruce biogenic volatie organic compounds. While I think it is a interesting study subject, I found the manuscript difficult to evaluate because I could follow the experimental design: what was the n for all the different treatments, plots, sites, ect.? when was sampling performed i.e dates and time? I think it would be very helpful to make an experimental design figure.*

Thank you for your comment and helpful suggestion. We would like to avoid confusion and will include an experimental design figure/table, rather than having it spread out in the text.

*My other major comment is about the statistics. I saw that there was a statistics paragraph in the methods, but I would like to have more details about how the statistics were perform. Also, no statistic were included in the result or figures.*

We agree that it can be clarified. All measurements were included in the statistics, from all sites and plots. Clarifications have been made accordingly: Using a Kruskal-Wallis test (MATLAB R2021a, The MathWorks, Inc., MA, US) with a level of significance set to $p < 0.05$ we compared the following scenarios: 1) emission rates between the healthy tree plots, plot 1, 2 and 3 in HTM, 2) the difference in emission rates from healthy trees and infested trees, from all plots and sites, 3) the difference in emission rates from the two infested trees in the sub-study and 4) the difference between $Q_{10}$ and $F_0$ for healthy and infested trees.

We decided to only perform a Kruskal-Wallis test because of our small sample set. Our data was also not normally distributed which further limited our choice in statistics to perform. We decided that in order to test our hypothesis, we simply needed to know if there was a significant difference between the groups or not which we could find our using the Kruskal-Wallis test for our different scenarios.

Statistics are included throughout the Results section where the results of the statistical test are presented as the p value that was given. The figures only include statistics in the form of some of them being boxplots, which includes minimum, lower quartile, median, upper quartile and maximum as well as outliers.

*Finally, in the first sentence of the last paragraph of the introduction it is stated "The defense mechanism of Norway spruce is poorly understood." I don't think this is a fair assessment of the field. We know quite a bit about the induction of terpenes, phenolics, and traumatic resin ducts (e.g Krokene, 2015 Conifer Defense and Resistance to Bark Beetles in Bark Beetles: Biology and Ecology of Native and Invasive Species ; Celedon and Bohlmann, 2019 Oleoresin defenses in conifers: chemical diversity, terpene synthases and limitations of oleoresin defense under climate change, New Phytologist). Although, I agree that we still have a lot to learn.*

As this was also commented on from Referee #2, we agree and understand the poor choice on our wording. We changed the sentence to "There is still a lot to learn about the defense mechanism of Norway spruce, only a few studies have analyzed the induced BVOC emission….". And would like to thank you for contributing with references on the matter.

---

## Author Comment (AC2)

**Response to the comment of Anonymous Referee #2**

We appreciate the time and effort from Referee #2 for their extensive work to provide detailed comments and great suggestions on our paper. We address each comment below, where the reviewer's comments are shown in italics. The line numbers refer to the original document.

*General comments*

*This study is interesting and has its own important value to supplement what has been done previously and giving new information about bark beetle infestation effect to BVOC emission from Norway spruce bark.*

Thank you for the positive comments regarding the value of this study.

*A lot of confusing information and weak Discussion is not helping to reveal this experiment value.*

Thank you for pointing this out. We will work on revising and re-arranging the MS to avoid confusion and work on strengthening our Discussion – see detailed responses below.

*Important part is also the new emerging volatiles, but these get so little attention or no attention in Discussion part.*

Thank you for your input, we will elaborate more about this in the Discussion.

*Discussion part is weak and needs more improving, more references and discussion that is connected with result and putting the results in bigger picture.*

Thank you for your comment. We found difficulties in increasing our reference list for our Discussion section as there are only very few studies available to discuss with. We will expand and elaborate our Discussion to include a broader perspective and clarity to when we discuss and speculate around our own results to provide hypothesis to be tested and added on in follow-up studies. See detailed responses on the Discussion below.

*I suggest leaving out the comparison with leaf emission; this does not give an extra value to the study. This has opposite effect: to diminish your efforts.*

Thanks for the suggestion on leaving out the comparison with leaf emissions. We value the perspective you have given us and will consider leaving this part out in a revised manuscript.

*Through MS: commas are there where they are not supposed to be and at the same time missing where they should be. I suggest to ask some native speaker to check English (my English is also not so good, but based on my experience I guess I understand when English needs improving).*

Thank you for pointing this out, we will consider running our MS through a professional copy editor.

*A lot of using %, this does not make the MS better. Try to find better alternative for this to describe results.*

Thank you for this input, we appreciate the comment and will make changes to improve readability. We use % mainly to describe the increase/decrease in emission rate, the relative occurrence of each compound in all samples and the mass contribution of each compound in the samples. We will convert % to xx-fold increase/decrease in the first part. Using % is necessary for the latter two as we cannot quantify all of the detected compounds but we can compare the GCMS peak area to find the contribution of each compounds and compare the percentage within and between samples. We don't see a better way of displaying.

*1) Title says: "...cause up to 700 times higher emission..." – but compared to what?*

This is compared to the emission rates from healthy trees. Suggested title change for clarity: "Spruce bark beetle (*Ips typhgraphus*) infestation cause up to 700 times higher bark BVOC emission rates compared to healthy Norway spruce (*Picea abies*)"

*2) L14 – drilled – used one style trough MS: drilled or boring; instead "entry holes and exit holes" use*

*"entry and exit holes", check whole MS*

Corrected!

*3) L15 – instead "healthy trees and infested trees" used "healthy and infested trees"; write out site*

*names in abstract*

Corrected!

*4) L21 – decreasing relationship – is it statistically true?*

We test this by the 'goodness of fit' in terms of RMSE from which we found an exponentially decreasing relationship to best describe our data with the lowest error. Because of the few data points we did not further test this statistically and our intention is mainly to show the type of relationship in our data..

*5) L28 – " This study...." This kind if sentence fits better to Conclusions*

We agree and have removed this part of the Abstract.

*6) L54 – "..SOA." – add reference*

Reference inserted. (Kulmala et al., 2003)

*7) L55-56 – better use "negative or a positive feedback loop..."*

Corrected.

*8) L57-58 – form a better sentence*

Removed sentence, we regarded it as repetitive and must have been overlooked previously.

*9) L63 – any references from last 10 years? If yes, add.*

Yes, references added. (Celedon and Bohlmann, 2019; Krokene, 2015)

*10) L10 - any references from last 10 years? If yes, add.*

Assuming this refers to L70, no additional reference than (Everaerts et al., 1988) was found looking at the specific compounds mentioned. However, the references mentioned above was added after the section before "they have been shown to be toxic to spruce bark beetles".

*11) Trough MS – I suggest to use Norway spruce instead of (just) spruce when related to your*

*experiment and xyz spruce when related to specific reference, if possible*

Thanks for the suggestion. We understand the need of clarification and will change accordingly.

*12) L78 - I guess it's not poorly understood, because every experiment gives valuable information, but*

*of course depending of the aims*

As this was also commented on by Reviewer #1, we agree and understand the poor choice on our wording. We changed the sentence to "There is still a lot to learn about the defense mechanism of Norway spruce, only a few studies have analyzed the induced BVOC emission….".

*13) L81 – Connections with what? Some words missing?*

We changed the sentence to clarify: ".. the connection **between** BVOC emission rate, number of bark beetle holes and time."

*14) L82 – For me the bark emission comparison with needle emission is not giving this MS much more value. Needle emission and bark emission measurements from spruce should be standalone experiment.*

We appreciate the new perspective that you bring on this and are willing to consider this part to be removed from the MS with the motivation you mention, needle emission and bark emission are in our case hard to compare as they are not measured at the same time. However, we still think it is important to compare emission rates from different parts of the tree to fully understand the whole tree dynamics. Our main reason for including this in our study was to highlight the importance of induced emission rates from infested bark, where the emission rates are so high that they greatly exceed those of leaf emissions, which normally is considered to be the dominant ones.

*15) L84-90 – I suggest not to use we/our etc for this part (check also trough MS). For the reader would be helpful to find hypotheses easily when they are numbered 1,2,3 etc and hypotheses presented clearly and properly.*

,Thanks for pointing this out, we will remove we/our throughout the text, and we agree that re-structuring the hypotheses for clarity is needed.

*16) L87 –"…eventual death of the tree" – I assume you did not measure the tree until it's eventual death. How long does it take when the tree is dead or considered dead after bark beetle infestation?*

We consider this a matter of definition. From our perspective, we consider the tree dead when the needles have shed off because of the disruption in the vascular system of the tree, which can happen at different time scales depending on the extent of the infestation. To achieve tree death, the beetles kill the cambium which is the part capable of emitting VOCs as a defense (Krokene, 2015), so when we measure close to 0 emissions or at least lower than the emission from healthy trees, we would consider this emissions from a dead tree. To answer your first assumption, we did not measure one specific tree until its death, but we did measure different trees at different stages where one was for certain considered dead after almost a year since infestation.

*17) L89 – "total number of holes" – what do you mean in "total number of holes"? Entry+exit holes?*

Yes, "total number of holes" is meant as entry+exit holes. For clarity we changed to: "rather than a high amount of holes".

*18) L91 – You did not investigate blue stain fungi in this experiment – I suggest to use blue stain fungi*

*in discussion part.*

We agree, the mention of blue stain fungi is removed in this sentence.

*19) L92 – "tree individuals" – I suggest "two trees"*

Agreed and corrected!

*20) L99 – I suggest that in this sentence indicate table 1 for campaign dates*

We have changed accordingly.

*21) L96 –Based on this sentence I would expect that in a sub-study where more than one campaign*

*but it's written over time. For me it indicates more than one campaign.*

The sub-study we refer to in this case is indeed more than one campaign. We followed two tree individuals from May to August, from before to after infestation. So the indication of it being more than one campaign is accurate. We will clarify this in the revision.

*22) L101 – use one style: sub-study or additional campaign trough MS*

Based on comment 21 above, we believe there to be a misunderstanding. The sub-study is the study in which we compare two trees with different health status over time before and after infestation. The additional campaign was a campaign carried out in Norunda on top of the campaigns in Hyltemossa, it was an additional campaign as we did not plan it but acted on the opportunity to travel there to measure more bark beetle infested trees. We will this keep the separate phrases as it refers to two different things, and will try to make it more explicit in the revision.

*23) Figure 1 – how are the rules for figure legends, is there a dot or semicolon after Figure 1? What is*

*the rule for figure legend text - needs to be in bold? I like the map for showing, but describing b-ca is not common in figure legends also in text (like Fig.1b). If possible, change maps etc. starting*

*with abc and in figure legends also describing abc. Missing proper explanations what yellow*

*squares and circles mean. "The Figure..." –why is there Figure not figure? If possible, write out in*

*the legends which site had healthy and infested trees, and where was sub-study done.*

The manuscript was conducted using the Copernicus_Word_template.docx in which the figure caption is in bold and has a semicolon after Figure number. We agree to change the orderof description to abc in the Figure caption, however we prefer to keep references like "Fig.1b" in text as it brings clarity to what is referred to in the text. We also appreciate that you noticed the lack of explanation in the figure caption, and the use of a capital F in "The Figure", this is not on purpose and will be added/changed accordingly.

*24) L122 – "by visual examination" – meaning no entry holes?*

Yes, in the visual examination there would be no entry holes, but we also looked at the general health of the tree. The tree could for example have been stressed due to forest machinery ripping up the roots, some other pests like aphids and adelgids. If we saw signs of this the tree was not selected. We will clarify this during the revision.

*25) L123 – Indicate site name*

Gustafsborg säteri AB is the company that owns and manages the forest, thank you for pointing out that this information is missing here and may cause confusion.

*26) L120-125 – write out here which was here considered as sub-study*

The sub-study is described below these lines in a new paragraph starting at L128. We will clarify that this is where the sub-study was conducted.

*27) In Methods part, please consider clearly explaining healthy, infested and control, because later you*

*show also control measurements and use the same style trough MS.*

We agree, it is currently inconsistent and it is now corrected in the MS. Control trees and healthy trees are the same and we already decided on using healthy trees only but obviously missed to change this through the MS.

*28) L130 – check names and ALL NAMES in MS must be correct, many mistakes: alpha-pinene, αpinene, beta-pinene, β-pinene etc and other names written not well.*

Thank you for highlighting this, we agree, consistency is important and this has been corrected for.

*29) L130 – Typosan- company name?*

The company name must have been missed, it is now corrected "(Typosan IPS, Plantskydd AB, Ljungbyhed, Sweden)

*30) L132 – repeatedly: measurements dates are indicated where?*

Dates indicated by Table 1, tree ID's are "S3S3" and "S3S2". This has now been indicated in MS.

*31) L136 – indicate figure*

Corrected

*32) L137 – use one style trough MS 5°C to 22°C or 5° to 22°C*

This has been corrected to the latter.

*33) L138 – start with May and then with June*

Agree, this has been corrected.

*34) Figure 2 – in figure legends start describing with temperature as you have on the figure temp,*

*precip, campaign. Figure legend text in bold? I suggest writing out in a better way where data was*

*taken.*

Figure caption has been corrected to follow the structure of the legend. The figure legend text is kept in regular to decrease the number of fonts used in the figure to one according to online instructions (https://www.biogeosciences.net/submission.html#templates). The description of data has been corrected.

*35) L147 – pump company? I guess L/min is visually better than lpm. 0.6-0.9 L.*

The pump box was custom made, but agree that the pump and flow controller information needs to be given here.. Corrections have been made accordingly. We decided to keep lpm as this is our preference. Changed liter to L.

*36) L148 – "Ad sorbent tubes…" goes to VOC collecting part*

Corrected.

*37) L153 - 40°C or 40 °C – use one style trough MS*

Corrected, a space between number and unit is intended as described in online instructions.

*38) L155 – I suggest here to use diameter, not inches; PTFE-tubing or PTFE-lines?*

Changed to metric system, 6.35 mm (1/4"). Corrected to PTFE-tubing, not lines.

*39) L158 – "each sample" – do you mean here each VOC collection?*

Yes. For clarity "each sample" can be changed to "each VOC collection".

*40) L160 – so the first VOC measurement was at 8:00 in the morning or you started to put the*

*chamber bases on tree?*

Yes, as indicated the sampling started at 8:00 in the morning. The chambers were placed onto the tree trunks prior to this to allow for flushing.

*41) L166 – Can you add the info when the infestation (swarming time) was detected. "For plot 3"*

*indicate plot name. "…witnessed on site." – when?*

Info on swarming times can indeed be added asthis adds important information, thanks for pointing this out.. The swarming time had been used later to calculate how long the infestation has been ongoing, so we missed to explicitly mention it at this point.. Plot 3 is the name of Plot 3, the site is HTM which we can indicate for clarity. The date of the infestation start can be indicated more clearly in Table 1 where the date is already there (2019-06-04). We will add supplementary information to the swarming period time series to increase transparency on how accurately we were able to estimate the swarming date as we determined this from peaks during swarming periods taken from measurement stations located nearest to our measurement sites.

*42) L168 – based on data from Skogsstyrelsen database, when was the proposed swarming time?*

See answer to comment 41.

*43) L170 – Did you mark the entry/exit holes to distinguish new and previous entry/exit holes?*

No, the entry/exit holes were not marked, so it is possible that "old" entry/exit holes where measured, but we still believe it to be the time since the infestation start that matters in our case, as well as the distinguishing between entry and exit holes.

*44) "By looking at bark photographs…" – I suggest to use "photos" and it's the first time to indicate*

*figure 4, so, in the end of sentence, indicate figure, not in the next sentence.*

Thank you for this input, we have corrected accordingly.

*45) L173 – Rewise sentence – I suggest not to start the sentence with "table 1".*

Corrected: "The number of holes visually counted inside the chamber area is listed in Table 1 along with an extrapolation of the number of holes that would represent per square meter bark area."

*46) Figure 3: On figure you have VOC filter, but in Methods part there is no info about VOC filter. Is it*

*the same as nets? "A photograph of how the setup looked in the field is displayed to the right (b)." – I*

*suggest revising. "The BVOC samples were.." - add here the pocket pump also. Add tree name or latin*

*name.*

You are correct, in the Methods we do not mention "VOC filter" specifically but have only failed to be consistent in wording. In the Methods we named it hydrocarbon trap (L149). We will change this to VOC filter to keep the consistency. Added mention of the pocket pump as well as tree name. The text was revised to mention (b) in the beginning: "The experimental schematics (a) and field photo (b) indicating the tree trunk chamber mounted on a tree…".

*47) Figure 4: which tree – add name or latin name. Correc to " entry and exit holes". Photos where*

*done by who? Drilled or bored?*

Thanks for the clarification, tree name has been added, and corrections on the entry and exit holes. Photos were taken by the first author, something we must have missed out on referring to. Agree to change "drilled" to "bored" to keep the consistency.

*48) Table 1 – text in bold? Which infested tree –add name. Which site/area trees belong? Table*

*paragraph spacing. "exit and entry holes". There is no explanation what S1S1 etc mean? "Type*

*majority"-> "Hole type majority". If possible add spruce initial health status also – healthy, infested*

*etc.*

The text is kept in bold as the template has figure text in bold. Name has been added as well as two columns indicating plot and site. S1S1 etc are the tree IDs. Type majority was changed to Hole type majority. As indicated in the answer to comment 41, one paragraph could potentially be added indicating the start of the swarming and/or the time passes since infestation start.

*49) L190 – I suggest to add in this paragraph how much healthy and infected tree BVOC measurements*

*were done in total*

Thank you for the suggestion, this had been added.

*50) L191 –Tenax company? Carbograph company?*

The adsorbent tubes were bought packed with those two chemicals in a 2-bed configuration fromMarkes International as one product, so we prefer to give Markes International as reference.

*51) L194 – Why was the collecting time 5 to 6 L not exactly 5 L or 6 L?*

The liters collected varied because of the sample time variation. Because of unforeseen events sampling was varying +/- 5 minutes with the goal of it being 30 minutes. This is however accounted for in the sample analysis which is based on the sample volume and not on the sample duration.

*52) L196 – "twice a day" – can you say at least before mid-day or after mid-day? Or when?*

Clarification has been made "..from air entering the chamber once before the first sample and once after the last sample of the day to capture…"

*53) L197 – "at the start of the BVOC sampling and at the end"-> "at the start and at the end of the*

*BVOC sampling"*

Thank you for the suggestion, we have changed accordingly.

*54) L198 – "After sampling.." - does this sentence repeat the previous sentence? "During sampling period"?*

Thanks for pointing out this can be misunderstood. To clarify, the sentence before refers to the air temperature inside the chamber which was measured during the sampling period, i.e. during active sampling. After sampling was done we measured the bark temperature as well. We have clarified stating that the temperature measured inside the chamber was air temperature.

*55) L205 – He, which company He?; VOC or BVOC – use one style trough MS*

(Air Liquide Gas AB, Sweden). Changed the MS to use BVOC throughout.

*56) L210 – space between number and m/mm*

Corrected.

*57) L220 – Shimadzu... - country?*

Japan, corrected.

*58) 2.4 – ALL equations need numbers and indicated in the MS where necessary*

Some confusion on this comment. All equations have numbers in parenthesis on the right hand side. However not for all cases is there an indication of the equation prior to the equation itself. This is something we can easily correct.

*59) L225 – in equation the A is normal, but in describing text it is A (italic)*

Corrected

*60) L243 – For me you have three ways to describe 1) 100 days before/after; 2) majority entry and /or exit holes; 3) early and late season. And all these three are used trough MS. I suggest to explain this well in this paragraph and use one style trough MS and also on figures and figure legend text.*

We agree and have changed to keep it consistent.

*61) L256 - Q in italic?*

Corrected

*62) L258 – F in italic?*

Corrected

*63) L262 – F and Q in italic?*

Corrected

*64) 269 –Eq 4 – where, which one?*

See line 250 right hand side.

*65) L270 – Eq 5 – where, which one?*

See line 264 right hand side.

*66) L270 – I suggest explaining in Methods part: does constitutive mean healthy or control tree emission. And what is considered as control, because later on the figures you have a control.*

Control trees are the same as healthy trees, we have changed to use healthy only. Constitutive emissions mean emissions from healthy, unstressed trees as constitutive emissions are what is emitted constantly. But we will add a clarification.

*67) L274 – 3 m*

Corrected

*68) L275 – North, East? north? east? - Look trough MS that weather arc are correctly. I suggest that weather arcs info goes in BVOC collecting paragraph.*

Corrected.

*69) L276 – I'm not so fond of comparing bark emission with leaf emission. This should be extra study to compare with more measurements campaigns. For me this comparison does not give an extra value to this study (overall). For me the core of the study is bark emissions from infested and healthy trees and then finding important results to discuss on Discussion part and with SOA and climate change and modelling (MEGAN etc).*

See reply under the general comments. While we agree that the focus here is on the bark emissions, we also want to make a point that under infestation the emissions from bark are way larger than emissions from leaves – which normally are dominating the overall emissions from a tree.

*70) L282 – outliers from which sites/area? I suggest moving this paragraph closer to BVOC sampling paragraph. "Photography"->"photos"*

Clarification/corrections has been made.

*71) 2.5 Statistical studies – Needs a lot improving. In here also clarify which measurements, from what area/sites, what about constitutive etc, what about sub-study? Look my comment before. P in italic (p)? Same with F and Q. Clearly write out what did you compare and with what statistical test and which program did you use for it.*

Thanks for this comment as it seems this paragraph was confusing and may have been misleading. However, the requested information is listed. All measurements were included in the statistics, from all sites and plots. Clarifications has been made accordingly: Using a Kruskal-Wallis test (MATLAB R2021a, The MathWorks, Inc., MA, US) with a level of significance set to $p < 0.05$ we compared the following scenarios: 1) emission rates between the healthy tree plots, plot 1, 2 and 3 in HTM, 2) the difference in emission rates from healthy trees and infested trees, from all plots and sites, 3) the difference in emission rates from the two infested trees in the sub-study and 4) the difference between $Q_{10}$ and $F_0$ for healthy and infested trees.

We decided to only perform a Kruskal-Wallis test because of our small sample set. Our data was also not normally distributed which further limited our choice in statistics to perform. We decided that in order to test our hypothesis, we simply needed to know if there was a significant difference between the groups or not which we could find our using the Kruskal-Wallis test for our different scenarios.

*72) Should there be dot in each main/second headline, like 3. Results/ 3.1. zyx ?*

According to the online template (referred to above) there is no dot.

*73) 3.1. – Revise the tile if clarifying "constitutive" in previous MS parts*

This will be considered as stated above.

*74) L294 – constitutive=healthy? Is it temperature-standardized or temperature standardized? Use one style.*

This has been corrected to use temperature standardized throughout MS.

*75) L296 – indicate the results in table X if they are written out in table.*

Thank you for highlighting this, they are written in the table and this had now been indicated.

*76) L301 - The numbers here and on the figure are not well trackable/comparable and at the same time quickly understood on the figure. If I read the sentence and look at the same time on the figure, cannot distinguish quickly what is what and figure is hard to read for a reader who does not read the results part about this figure.*

The numbers on the second sentence at L301 is referring to the range of the emission rates at each plot (1-3) as visualized by the boxplot (Fig. 5), the medians are referring to the medians as visualized by the boxplot. The figure is describing all the emission rates measured throughout the season for the healthy trees at the different plots, the point of the boxplot is to show that there is no significant difference in emission rates between the plots, which we think is clear from this figure.

*77) L303 – p-value? p>0.3. Make changes through MS.*

Corrected.

*78) L304 – which sites in what area?*

Correction – mistake in MS, sites = plots at the same site in HTM.

*79) Add on Figure 5 HTM, so the reader can understand what area without reading figure-describing text. In describing text, indicate statistical test. What n means? What black circles mean?*

Thank you for the suggestion. A clarification on the parts of the box plot will be added in the figure caption.

*80) L311 – infested trees in which area?*

We indicate all infested trees in all areas, but this will also be clarified.

*81) L312 – drill or bored? "The season…" – you were describing it before in Methods part, in here use one style to indicate, but not repeating the same as in Methods. Look my comment before.*

See answer to comment before.

*82) L314 – I suggest to add total emission also to Table 1.*

Thank you for the suggestion, we will add a row with total emissions to Table 1.

*83) L315 – "..µg m-2 h-1 . MTs…" check space. " MTs…" indicate for table 2.*

Corrected.

*84) L319 – 42% or 42 %? Use one style thorough MS.*

See replies above.

*85) L322 – Which trees: name? Which area? Name? BVOC or VOC?*

See replies above. All trees are indicated to be from all sites but this will be clarified.

*86) L323 – healthy from which area?*

From all healthy trees, only measured at site HTM, which will be clarified.

*87) L328 – infested tree: I suggest to use infested bark samples, bark emissions, infested tree bark etc through MS, one style through MS.*

We have considered your suggestion but will continue to use the phrase "infested tree". Bark beetles attack the bark, but the entire tree is substantially and most often lethally affected by the infestation.

*88) 329 – "isoprene, (n = 17) compared " check the mistake. Infested trees in which area?*

Typo corrected. Infested trees from all sites.

*89) L331 – indicate table*

Corrected

*90) L337 – Which areas/sites? P value – how come 0.00???? "The median…" - Consider writing this sentence according to this what is shown on figure or change figure according to the sentence. I advise to start with healthy and then infested (early, late). I suggest to use less "respectively" through MS, not so easy to follow.*

All sites. P-value is a typo, corrected to 0.001. The median is shown in Figure 6 as the line in the boxplot. We will change the text to start with the healthy trees, and then continue with infested trees – thanks for pointing this out.. Thank you for the suggestion on finding an alternative to "respectively", we will change this.

*91) Figure 6 - please add also HTM or NOR on the figure. Which tree? Again use one style for early/late season, exit/entry. "The emission rate.." sentence doesn't belong to figure describing text. Pvalue?? Standardized to temperature – indicate equation in XX part. Explain yellow and blue.*

*Statistical test used?*

See earlier replies. "The emission rate…" sentence is corrected. Indication of Eq added.

*92) L349 – healthy trees and infested. Which sites/area???*

All plots and sites.

*93) L315 – I suggest rounding the numbers to proper value in table, visually better. Overall, it's kind of disturbing using so much % through MS.*

Your suggestion has been noted and the numbers will be rounded. See reply to general comments regarding %.

*94) Check VOC names trough MS.*

See earlier reply.

*95) L287 – Again, using % and even so big numbers….do you use 22678 or 22 678? Which area, which site trees? Which trees?*

See earlier replies regarding %. We will change to 22,678 throughout MS for consistency.

*96) L374 – when it's the first time indicating Appendix, write out Appendix, otherwise what Table A1 means? trans, trans-alloocimene - > maybe (4E,6E)-alloocimene better?*

Thank you for the suggestion. This has been corrected.

*97) L375 – gamma-terpinene or γ-terpinene? "in infested…"- revise end of the sentence, double Repeat*

This has been changed throughout MS, γ-terpinene is used. The sentence has been revised.

*98) Table 2 – paragraph spacing. Which tree? Which area? STDEV is only for groups, not for each compound. Explain MT, SQT. What about decrease (-65%), not explained, only increase. Again numbers: 22678 or 22 678? Check all the names in table, all the names need to be used correctly trough MS (tables, figures, etc). If possible add total emission.*

All trees, all plots, all sites. STDEV will be added for each compound. Other suggestions have been checked and corrected.

*99) 386 –Which area? Which trees? Number 13000?*

All sites and all infested trees. We noticed a large misinformation in the text, number 13000 refers to the highest temperature standardized emission rate measured from one single tree which can be seen in Figure 7. This is however not the seasonal average for all trees as it seems in the text. This has been revised to indicate the range in seasonal average for the individual trees, which ranged from 500 to 13000 as indicated by Figure 7. Thanks for spotting this!

*100) L390 - "…daily average standardized emission.." – check*

This is what we intended, the daily average temperature-standardized emission for clarity.

*101) L391 – "The average number of holes per square meter bark area found in this study was based on the values in Table 1." Revise or delete.*

If it does not bring any clarify to the following text we will delete it.

*102) L396 – Whole sentence "For…." – check and correct. Are here the results of sub-study?*

This is not the sub-study, this part is comparing the emission rates as only standardized to temperature with the emission rates when also standardized to bark beetle holes. This was done to exclude any difference in the results simply due to the variation in number of holes (20 holes might indicate higher emission rates than 10 holes). We will clarify this in the revision, thanks for pointing out.

*103) L400 – "The TS emission…" – I guess this sentence is to long for explanation, I suggest to divided*

*in two parts and describing clearly. I guess this kind of description goes to Methods part in*

*Statistics.*

We will move or add a section on this in the Methods part. But we do think it is necessary to keep where it is also as the continuation of the MS is based on this.

*104) Figure 7 – Which trees? What area? S3S2, S3S3, etc meaning? Early/late, 100 days before/after.*

*"The emission rates…" and "This is evident…" sentences do not belong to figure describing text.*

All infested trees in all sites and plots. S3S2 etc is the individual tree ID, will be clarified or changed to "tree 1" etc. Corrections have been made accordingly.

*105) 3.2 – Did you to statistics to evaluate influence of time?*

See answer to comment 4.

*106) L413 – Which site/area? Indicate table/figure.*

This was a general statement, but will be changed to avoid confusion.

*107) L415 – Where is 3.2.3? I suggest to put function to Statistics part and indicating on the figure*

*describing text where to find it.*

Thanks for spotting the wrong reference to 3.2.3. We will move the exp function to the methods/statistics section, and refer to it in the text here; thanks for this suggestion that will make the text more readable.

*108) L420 – use one style trough MS: 8b and 8d or 8b,d etc.*

We agree and will change this to ensure consistency.

*109) L421 – I guess "slightly" is not correct to use here. "..depending…" revise, double repeat.*

This has been revised.

*110) L425 – How long does it take from infestation to total damage/death to the tree? Maybe*

*something to discuss in Discussion and some info for introduction part?*

See reply above (comment 16).

*111) L428/429 – indicate figure. Again constitutive/healthy emission? Area?*

Corrected.

*112) L432 – "The tree measured.."- did it have a name SxSx? What area?*

Corrected.

Corrected.

Revised and corrected.

See previous comments. We would like to argue that the use of abcd before the intended figure part is not wrong but rather a matter of preference. As intended in the figure caption "All trees are included in the exponential fitted curve", also the yellow one. It is simply marked to show that it had lower emission rates compared to the other trees. No test is used due to low sample number here, and in Methods part there is indeed no specific indication of the measurements after 300 days since infestation had started, however, it was assumed from our part that it was obviously included in the measurements > 100 days since infestation has started. This will be clarified, and we thank the reviewer for spotting this inconsistency

Corrected.

Corrected.

Corrected.

See reply before.

Corrected

See previous replies for Figure 8. The "2" on the x-scale title is the same font size as the other text, but will be changed to a smaller font.

*122) 3.4 - sub-study?*

This is indeed the section where the results from the sub-study are presented. This will be indicated more clearly.

*123) L466 – Revise sentence "At plot 3…" and indicate figure. On another way L466-L469 is describing what you said before, I suggest revise this sentence. Which area?*

This will be revised.

*124) L470- P in italic.*

Corrected.

*125) L471 - Control trees – what does the control trees mean in MS? Control - not at all infested during campaign or beginning of campaign not infested and later experiencing bark beetle infestation? Indicate figure 8 correctly.*

See previous replies. Control trees = healthy trees, which were healthy all the time. Indication corrected.

*126) L473 –Which other trees? Other trees in plot 3 or somewhere else? Indicate a tableX?*

Other trees in plot 3. There is no table to indicate as we decided to only present this as the figure for better visualization.

*127) L576 –" a difference.." indicate table if possible*

Corrected. See reply above.

*128) L477 – check names; I suggest to remove 800 ug or use another number for comparing.*

Corrected and agreed regarding the 800 ug, seemed a bit random as it was.

*129) L479 - check names*

Corrected.

*130) L480 – indicate figure, I suggest that p<0,03. Delete "and close to zero", check names*

Thanks for the suggestion. Corrected.

*131) L483 – p italic*

Corrected.

*132) L484 – indicate figure*

Corrected.

*133) L485-495 – Again less % or even when presenting choose important results, because this part is interesting. I suggest this part to discuss in Discussion part. Check names. "..by mass" – what mass?*

See reply in general comments. Mass is the compound mass found in the sample, the %-age refers to the fraction of the respective compound regarding the total emitted mass. The results were narrowed down to what we thought were the most important compounds found, but a more elaborated list of

compounds found is shown in the appendix (Fig A1). We will clarify this point and explicitly refer to this figure in the revision, thanks for pointing this out. The ongoing infestation is discussed in 4.1.1., but we will make this more clear. Thanks for pointing this out.

*134) Figure 10 - Months names ith capital letter. On the figure, where is asterix-what does it indicate? Can't be seen. Check names! All figure plot need to have month names. Describing text what trees, what area. Why no SE? Add tick marks on the figure, other figures have. Remove the outer box from top of the figure, where are indicated control, S3 and S2 and add some space between them for clearer separation. Start with control in describing text, then healthy or infested etc. Add year in describing text. "For", I guess "from". In this figure, there is important info, but this does not come forward clearly because of figure negative sides. Maybe another alternative?*

See replies for other figure comments. Months have been capitalized. The asterix might be confusing, it is there to indicate that the trees were not infested in May. We will find an alternative. Month names will be moved as axis labels on the right to avoid repetition in the plot. SE is not included as the average is an average over the day, 3 measurements. This can of course be added if necessary but we did not think it would add any value other than cluttering the figure. We have added more space between labels in order to increase the readability and clearity of the figure which we think is also supported by the legend box that we have applied constantly throughout the figures in this MS. The control (changed to healthy) is shown in green, S3S3 is orange and S3S2 is blue in the figure. We hope the figure is more clear now. Revised caption text is inserted below where bold letters indicate changes:

"Figure 10: The average **temperature** standardized **BVOC** emission rates for all compounds **from Norway spruce at plot 3 in Hyltemossa: healhty trees (green), infested spruce with ID S3S2 (blue) and infested spruce with ID S3S3 (orange). Measurements were taken in 2019 during** May (a-b), June (c-d), July (e-f) and August (g-h). The graphs are horizontally separated for visibility due to large differences in scale. The **healthy** trees are included in all graphs but the emission rates are not visible on the same scale as the infested trees in June (c-d) or July (e-f). The bark beetle infestation had not started in May (a-b), however, the spruce S3S2 was already subjected to stress from late bark beetle attacks previous season before the bark beetle infestation started again in June (c-d), leading to higher emission rates in May."

*135) L506 – F in italic? Q in italic? Look through MS*

Corrected

*136) L507 – which area?*

All sites and plots.

*137) L508 – indicate a table*

Thanks for pointing out this was not clear enough. The reference for this is also Table A2, and we will make sure that Table A2 will be indicated here as well.

*138) L510-511 – double repeat*

Revised.

*139) L513 – 34900*

Changed to 34,900 µg m-2 h-1.

*140) L510-520 – too much %*

See replies on general comments; but we will reduce those high %-ages and refer to x-fold increase instead.

*141) L521- coefficient?*

This is meant to be the Q10 coefficient. Corrected.

*142) L526 – p<0.05 and p<0.01*

Corrected.

*143) 3.6 – As is said before: for me leaf emission does not give an extra value for the study. Maybe to leave this part out and present the result what are in this chapter? Again which area?*

See replies to general comments. The short paragraph on the comparison with leaf-scale emissions in our view adds valuable information and helps to assess the importance of the induced emissions as normally only leaf-emissions are considered as for healthy trees these are dominant.

*144) L537 – Ms or Ms?*

Corrected.

*145) L558 – well?*

We will revise this sentence as "**Emission rates from the bark of the infested trees were at least about 55 times higher than the total MT emission rate from both leaves and bark of a healthy tree.**"

*146) Figure 11 – pleases start with abc, revise figure then. Which tree, which area? Again: north, east, North, East? Meters -> m. Which ONE infested tree (SxSz?), which area? What green and purple shades are representing? Add tick marks on y-scale.*

Revised.

*147) 4. Discussion – needs a lot improving. Right now it's weak. Indicate table and figures where proper. Check names. What areas? More references and discussions. Somehow Discussion part repeats Results part, would be better to have more discussion about results.*

Thank you for your input. Tables and figures will be clearly indicated. Names have been checked. Sites will be clarified. The number of references used in our Discussion is low because of the low number of relevant studies to compare with. We decided to deliberately use only studies where a comparison will be relevant between Norway spruce bark BVOC emissions and stresses, particularly insect stress from the European spruce bark beetle. As the number of comparable studies remains low, we determined to keep our discussion very open as we thought it to be hard to draw any firm conclusions from our findings due to the limited dataset.

*148) L587 –what relationship? How did you calculate this? Assessing how? Based on what? No info in Methods. Any references about lichen cover related to isoprene emissions. Ghimire – how did they*

The relationship we are referring to would be a correlation, either positive or negative. The reference Zhang-Turpeinen et al., 2021 found a positive correlation with isoprene emission and lichen cover. We did, as Ghimire, asses the lichen cover visually from our bark photos. But you are correct that this is not included in Methods, this must have been missed and will be included around L170 where we describe looking at bark photos for checking the hole types. Ghimire tested the correlation using the Spearman's correlation analysis, which we did not do, we only assessed this visually by comparing the images with the emission rates of isoprene – where there were no clear indication of anything, there would be high isoprene emissions and no lichen cover at all, which made us determine not to move forward with that. It is included in the discussion only because we thought it interesting that there would be isoprene emitted from bark with no clear explanation why.

*149) L591 – healthy and infested trees – what previous findings? References? Some examples for*

*describing? Indicate figure.*

The previous findings referred to is the previously mentioned references – however this will be further elaborated in the revision.

*150) L591 – repeating of the results. I suggest to put these number into comparison with some*

*references if possible or to discuss about emissions. I suggest not to use "we".*

Thanks for pointing this out. This part will be revised or removed. We will revise all first person terms throughout the ms..

*151) L549 - And so what it is higher at the beginning? Connect the higher emission with atmospheric*

*processes, add references. What is different in the beginning? Indicate your figure/table if proper.*

We will add some more discussion to why this is interesting. There are more entry holes in the beginning and the spruce defense is more active, as time passes the vitality of the spruce decrease as does the emission rates. We will elaborate this in the revision of the discussion.

*152) L583 – 605: all compared to one reference.*

Yes, this is indeed all compared to one reference as this is, to our knowledge, the one recent study comparable to ours, in which the measured the bark emission rates from Norway spruce from healthy trees and infested trees. There are of course some other studies where they look at fungal inoculation and European spruce bark beetles on Norway spruce, however this is more targeted towards chemistry and they measure the absolute concentration of BVOCs using a different method, not calculating emission rates (Mageroy et al., 2020; Zhao et al., 2010, 2011). As mentioned previously we wanted to mainly compare our results to similar studies (bark bvoc emissions from Norway spruce trees infested by European spruce bark beetles and measured with chamber systems -> emission rates). However, our results will be discussed more widely but this was not our first intention. We will follow the recommendation by the reviewer and elaborate the discussion to include this.

*153) L599 - time...what about temperatures, weather conditions? And what time, measuring time*

*during day (morning, lunch time, evening) or campaign times (certain dates).*

Temperatures should not be an issue as they also used standardized temperatures in their study, and using monthly means for their results. The values are comparable in the manuscript as well.

*154) L600 – "study…" indicate figure*

Corrected.

*155) L601 – Ghimire – which holes in this reference?*

All holes supposedly, they did not specify entry or exit holes from inside their chamber area.

*156) L602 – "this allowed…" then what is done before? References? What stage to what stage?*

We could not find a single study measuring a spruce before infestation and following it through infestation, thus relation the emission rates to the stages of infestation. What has been done before is to measure healthy trees, travel to another site at another time to find bark beetle infested trees, and so on. There are also older studies on the very beginning stages and emissions from entry holes for about 2 weeks, this is mention later in discussion (L632) but we will re-structure and mention this here as well.

*157) L604 – "This finding.." – what finding exactly? Why it is important in bigger picture?*

Corrected – not referring to a specific finding, rather the design of the study.

*158) L605 – "We could see…"- is it really a trend or rather something normal to be expected when*

*more time passes since infestation.*

We expected an exponential decrease, we might have used the word "trend" falsely here. The sentence will be revised to "The results strongly indicates a decreasing exponential relationship with time since infestation for the total BVOCs and the selected compounds, supported by the goodness of fit which provided an RMSE with low error. ". As we did not find comparable studies for this part of our results we will elaborate further in our discussion to include this information and suggest follow up studies on this.

*159) L608 – check space*

Corrected.

*160) L609 – references, more to discuss with few sentences?*

This is a statement that the trend we saw in our data is only related to the start of the infestation, not taking into account the time of the season and in what swarming period the infestation started, while for e.g. leaf emissions seasonality and phenology is known to be important. But we agree that if it might be of interest there could be some added discussion on the potential influence of the swarming periods and time in season. However, finding data for this to discuss around might be difficult as there are only very few studies available, and this rather would be speculation

*161) L611 – indicate figure*

Corrected.

*162) L612 – which total holes, majority of entry or exit holes? Or entry+exit? To discuss what could be*

*the possibilities for no pattern. Any references where this kind of pattern occurs?*

Yes, total number of holes refers to all holes, entry+exit. We did not see any pattern in emission rates and the entry+exit holes until we separated them. This is discussed further in this section. We have not seen any studies done comparing number/type of holes and emission rates. We first expected a

positive correlation between increasing number of holes and higher emission rates, but as mentioned we did not see that pattern. Also to further stress this, we did not have a generous amount of data on the infested trees, which made it hard to make any conclusions, but will be suggested for further studies.

*163) L614 – indicate figure*

Corrected.

*164) L616-617 – references. Too long sentence, I suggest to make into two sentences. Double repeat.*

Will add reference on the entry holes appearing at the start of an infestation. Will revise sentence.

*165) L618 –VOC`*

Corrected.

*166) L619 – which individual tree, which area?*

Will be corrected.

*167) L620-629 – References.*

This is a discussion around our results only. As we did not find any comparable studies we decided only to focus this part around our own results as speculations and forming of hypotheses which can be tested in follow-up studies. We will clarify this and we will elaborate this more with the reference we found looking at BVOC emissions from entry holes of Norway spruce (Birgersson and Bergström, 1989).

*168) L624 – All trees – what area?*

All trees in the study. All sites, all plots.

*169) L628 – any reference to support your results and conclusion?*

See reply above (comment 167).

*170) L631 – indicate table/figure if proper*

Table will be indicated.

*171) L630-639 – add references and something to discuss more than these two references*

This is the one reference we found looking specifically at BVOCs emitted from entry holes on Norway spruce.

*172) L635 – oxygenated monoterpenes?*

Yes, corrected.

*173) L639 – any references to add?*

We will try to find one.

*174) L640 – 4.1.1.- Only 3 references per this part?*

See replies above regarding low amount of references.

*175) L682 – what other 16 BVOCs? Revise, no need to add BVOC names exactly.*

Will revise this, the 16 other BVOCs if from the referred study (Lee et al., 2006), but it should be "compared to 13 other BVOCs" as alpha-humulene, longifolene and beta-caryophyllene was already mentioned and the study compared in total 16 BVOCs. Thanks for pointing this out.

*176) L700- "As the…" double repeat.*

Corrected.

*177) 684 and 690 – both sentences are a repeat. " This could.." and "This change.."*

Revised.

*178) L685 – Myrcene sentence goes rather to Results part.*

Correct, we will move it to the result section.

*179) 4.2 – again % and numbers. Now here Ips typographnus – why not bark beetle as trough MS?*

*Only 5-6 references per this part?*

See previous replies. Will change to bark beetle for consistency.

*180) Table A1 – Names: numbers; MT, SQT explanation – check comment for table 1.*

Will be corrected per previous replies.

**References**

Birgersson, G. and Bergström, G.: Volatiles released from individual spruce bark beetle entrance holes Quantitative variations during the first week of attack, J. Chem. Ecol., 15(10), 2465–2483, doi:10.1007/BF01020377, 1989.

Celedon, J. M. and Bohlmann, J.: Oleoresin defenses in conifers: chemical diversity, terpene synthases and limitations of oleoresin defense under climate change, New Phytol., 224(4), 1444–1463, doi:10.1111/NPH.15984, 2019.

Krokene, P.: Conifer Defense and Resistance to Bark Beetles, Bark Beetles Biol. Ecol. Nativ. Invasive Species, 177–207, doi:10.1016/B978-0-12-417156-5.00005-8, 2015.

Kulmala, M., Suni, T., Lehtinen, K. E. J., Dal Maso, M., Boy, M., Reissell, A., Rannik, Ü., Aalto, P., Keronen, P., Hakola, H., Bäck, J., Hoffmann, T., Vesala, T. and Hari, P.: A new feedback mechanism linking forests, aerosols, and climate, Atmos. Chem. Phys. Discuss., 3(6), 6093–6107, doi:10.5194/acpd-3-6093-2003, 2003.

Lee, A., Goldstein, A. H., Kroll, J. H., Ng, N. L., Varutbangkul, V., Flagan, R. C. and Seinfeld, J. H.: Gas-phase products and secondary aerosol yields from the photooxidation of 16 different terpenes, J. Geophys. Res. Atmos., 111(17), doi:10.1029/2006JD007050, 2006.

Mageroy, M. H., Christiansen, E., Långström, B., Borg-Karlson, A. K., Solheim, H., Björklund, N., Zhao, T., Schmidt, A., Fossdal, C. G. and Krokene, P.: Priming of inducible defenses protects Norway spruce against tree-killing bark beetles, Plant Cell Environ., 43(2), 420–430, doi:10.1111/pce.13661, 2020.

Zhao, T., Krokene, P., Björklund, N., Lngström, B., Solheim, H., Christiansen, E. and Borg-Karlson, A. K.: The influence of Ceratocystis polonica inoculation and methyl jasmonate application on terpene chemistry of Norway spruce, Picea abies, Phytochemistry, 71(11–12), 1332–1341, doi:10.1016/j.phytochem.2010.05.017, 2010.

Zhao, T., Krokene, P., Hu, J., Christiansen, E., Björklund, N., Långström, B., Solheim, H. and Borg-Karlson, A. K.: Induced terpene accumulation in Norway spruce inhibits bark beetle colonization in a dose-dependent manner, PLoS One, 6(10), doi:10.1371/journal.pone.0026649, 2011.

Zhao, T., Krokene, P., Hu, J., Christiansen, E., Björklund, N., Långström, B., Solheim, H. and Borg-Karlson, A. K.: Induced terpene accumulation in Norway spruce inhibits bark beetle colonization in a dose-dependent manner, PLoS One, 6(10), doi:10.1371/journal.pone.0026649, 2011.

---

## Author Response (AR1)

**Response to the comment of Anonymous Referee #1**

We appreciate the time and effort from Referee #1 to provide comments and great suggestions on our paper. We address each comment below, where the reviewer's comments are shown in italics. The line numbers refer to the original document.

*Comments*

*This study investigates the effect of bark beetle attack on Norway spruce biogenic volatie organic compounds. While I think it is a interesting study subject, I found the manuscript difficult to evaluate because I could follow the experimental design: what was the n for all the different treatments, plots, sites, ect.? when was sampling performed i.e dates and time? I think it would be very helpful to make an experimental design figure.*

Thank you for your comment and helpful suggestion. An experimental design table (Table 1) have now been included in which we specify the campaign month and date as well as the site name and the number of collected samples at each plot divided into healthy and infested trees and the total samples collected during the campaign dates.

*My other major comment is about the statistics. I saw that there was a statistics paragraph in the methods, but I would like to have more details about how the statistics were perform. Also, no statistic were included in the result or figures.*

Thanks for this comment as it seems this paragraph was confusing and may have been misleading. However, the requested information is listed. All measurements were included in the statistics, from all sites and plots. Clarifications has been made accordingly: Using a Kruskal-Wallis test (MATLAB R2021a, The MathWorks, Inc., MA, US) with a level of significance set to $p < 0.05$ we compared the following scenarios: 1) emission rates between the healthy tree plots, plot 1, 2 and 3 in HTM, 2) the difference in emission rates from healthy trees and infested trees, from all plots and sites, 3) the difference in emission rates from the two infested trees in the sub-study and 4) the difference between Q10 and F0 for healthy and infested trees.

We decided to only perform a Kruskal-Wallis test because of our small sample set. Our data was also not normally distributed which further limited our choice in statistics to perform. We decided that in order to test our hypothesis, we simply needed to know if there was a significant difference between the groups or not which we could find our using the Kruskal-Wallis test for our different scenarios.

This has been further elaborated in the statistics section to clearly include what we did and how it fits in our study in relation to our aims and hypotheses.

Statistics is included throughout the Results section where the results of the statistical test is presented as the p value that was given. The figures only include statistics in the form of some of them being boxplots, which includes minimum, lower quartile, median, upper quartile and maximum as well as outliers. This has been revised to increase the clarity of where the statistics are applied throughout the results section.

*Finally, in the first sentence of the last paragraph of the introduction it is stated "The defense mechanism of Norway spruce is poorly understood." I don't think this is a fair assessment of the field. We know quite a bit about the induction of terpenes, phenolics, and traumatic resin ducts (e.g Krokene, 2015 Conifer Defense and Resistance to Bark Beetles in Bark Beetles: Biology and Ecology of Native and Invasive Species ; Celedon and Bohlmann, 2019 Oleoresin defenses in conifers: chemical diversity,*

*terpene synthases and limitations of oleoresin defense under climate change, New Phytologist).*
*Although, I agree that we still have a lot to learn.*

As this was also commented on from Referee #2, we agree and understand the poor choice on our wording. We changed the sentence to "There is still a lot to learn about the defense mechanism of Norway spruce, only a few studies have analyzed the induced BVOC emission….". And would like to thank you for contributing with valuable references on the matter.

**Response to the comment of Anonymous Referee #2**

We appreciate the time and effort from Referee #2 for their extensive work to provide detailed comments and great suggestions on our paper. We address each comment below, where the reviewer's comments are shown in italics. The line numbers refer to the original document.

*General comments*

*This study is interesting and has its own important value to supplement what has been done previously and giving new information about bark beetle infestation effect to BVOC emission from Norway spruce bark.*

Thank you for the positive comments regarding the value of this study.

*A lot of confusing information and weak Discussion is not helping to reveal this experiment value.*

Thank you for pointing this out. The MS is revised throughout to avoid confusion and the discussion have been re-arranged and revised to point out specifically when we discuss around our own results and speculations. Even though there is a large lack of comparable studies, we have broadened our view to include more comparisons in the discussion. See detailed responses below.

*Important part is also the new emerging volatiles, but these get so little attention or no attention in Discussion part.*

Thank you for your input, this have been elaborated further in the Discussion part, see details below.

*Discussion part is weak and needs more improving, more references and discussion that is connected with result and putting the results in bigger picture.*

Thank you for your comment. We found difficulties in increasing our reference list for our Discussion section as there are only very few studies available to discuss with. The manuscript have been revised by elaborating our Discussion to include a broader perspective and clarity to when we discuss and speculate around our own results to provide hypothesis to be tested and added on in follow-up studies. See detailed responses on the Discussion below.

*I suggest leaving out the comparison with leaf emission; this does not give an extra value to the study. This has opposite effect: to diminish your efforts.*

Thanks for the suggestion on leaving out the comparison with leaf emissions. We value the perspective you have given us but have decided to not leave this part out in the revisions as we believe that it brings value in comparison with the needle emissions – as they are believed to the main contributor of BVOC emissions.

*Through MS: commas are there where they are not supposed to be and at the same time missing where they should be. I suggest to ask some native speaker to check English (my English is also not so good, but based on my experience I guess I understand when English needs improving).*

Thank you for pointing this out, we have revised the MS to improve the grammar.

*A lot of using %, this does not make the MS better. Try to find better alternative for this to describe results.*

Thank you for this input, we appreciate the comment and have revised this to improve readability. We use % mainly to describe the relative occurrence of each compound in all samples and the mass contribution of each compound in the samples. Using % is necessary for this as we cannot quantify all of the detected compounds but we can compare the GCMS peak area to find the contribution of each compounds and compare the percentage within and between samples. As we believe this to be hard to describe any other way this have not been revised. We have however revised to convert "%" to "xx-fold increase/decrease" when we describe the increase/decrease in emission rate.

*Specific comments*

*1) Title says: "...cause up to 700 times higher emission..." – but compared to what?*

Title changed to: "Spruce bark beetle (*Ips typhgraphus*) infestation cause up to 700 times higher bark BVOC emission rates compared to healthy Norway spruce (*Picea abies*)"

*2) L14 – drilled – used one style trough MS: drilled or boring; instead "entry holes and exit holes" use*

*"entry and exit holes", check whole MS*

Corrected. Changed bored to drilled, replaced "entry holes and exit holes" to "entry and exit holes" throughout the MS.

*3) L15 – instead "healthy trees and infested trees" used "healthy and infested trees"; write out site*

*names in abstract*

Corrected. Replaced "healthy trees and infested trees" to "healthy and infested trees". Added site names in Abstract.

*4) L21 – decreasing relationship – is it statistically true?*

No corrections made. We test this by the 'goodness of fit' in terms of RMSE from which we found an exponentially decreasing relationship to best describe our data with the lowest error. Because of the few data points we did not further test this statistically and our intention is mainly to show the type of relationship in our data..

*5) L28 – " This study...." This kind if sentence fits better to Conclusions*

We agree and have removed this part of the Abstract.

*6) L54 – "..SOA." – add reference*

Reference inserted. (Kulmala et al., 2003)

*7) L55-56 – better use "negative or a positive feedback loop..."*

Corrected.

*8) L57-58 – form a better sentence*

Removed sentence, we regarded it as repetitive and must have been overlooked previously.

*9) L63 – any references from last 10 years? If yes, add.*

Yes, references added. (Celedon and Bohlmann, 2019; Krokene, 2015)

*10) L10 - any references from last 10 years? If yes, add.*

Assuming this refers to L70, no additional reference than (Everaerts et al., 1988) was found looking at the specific compounds mentioned. However, the references mentioned above was added after the section before "they have been shown to be toxic to spruce bark beetles".

*11) Trough MS – I suggest to use Norway spruce instead of (just) spruce when related to your*

*experiment and xyz spruce when related to specific reference, if possible*

Changed the use of only "spruce trees" to "Norway spruce trees" to avoid confusion of mixing it up with another kind of spruce.

*12) L78 - I guess it's not poorly understood, because every experiment gives valuable information, but*

*of course depending of the aims*

As this was also commented on by Reviewer #1, we agree and understand the poor choice on our wording. We changed the sentence to "There is still a lot to learn about the defense mechanism of Norway spruce, only a few studies have analyzed the induced BVOC emission….".

*13) L81 – Connections with what? Some words missing?*

We changed the sentence to clarify: ".. the **relation between** BVOC emission rate, number of bark beetle holes and time."

*14) L82 – For me the bark emission comparison with needle emission is not giving this MS much more*

*value. Needle emission and bark emission measurements from spruce should be standalone*

*experiment.*

This has not been corrected. We appreciate the new perspective that you bring on this and are willing to consider this part to be removed from the MS with the motivation you mention, needle emission and bark emission are in our case hard to compare as they are not measured at the same time. However, we still think it is important to compare emission rates from different parts of the tree to fully understand the whole tree dynamics. Our main reason for including this in our study was to highlight the importance of induced emission rates from infested bark, where the emission rates are so high that they greatly exceed those of leaf emissions, which normally is considered to be the dominant ones. As this is of importance to understand the high impact infested tree bark can have, we have decided to keep this part.

*15) L84-90 – I suggest not to use we/our etc for this part (check also trough MS). For the reader would*

*be helpful to find hypotheses easily when they are numbered 1,2,3 etc and hypotheses presented*

*clearly and properly.*

We have removed "we/our" in this part of the MS and have removed where suitable throughout the MS. The hypothesis part have been re-structured and numbers have been added to the different hypotheses to enable a clear and proper presentation of them.

*16) L87 – "...eventual death of the tree" – I assume you did not measure the tree until it's eventual*

*death. How long does it take when the tree is dead or considered dead after bark beetle*

*infestation?*

We consider this a matter of definition. From our perspective, we consider the tree dead when the needles have shed off because of the disruption in the vascular system of the tree, which can happen at different time scales depending on the extent of the infestation. To achieve tree death, the beetles kill the cambium which is the part capable of emitting VOCs as a defense (Krokene, 2015), so when we measure close to 0 emissions or at least lower than the emission from healthy trees, we would consider this emissions from a dead tree. To answer your first assumption, we did not measure one specific tree until its death, but we did measure different trees at different stages where one was for certain considered dead after almost a year since infestation.

*17) L89 – "total number of holes" – what do you mean in "total number of holes"? Entry+exit holes?*

Yes, "total number of holes" is meant as entry+exit holes. For clarity we changed to: "rather than a high amount of holes".

*18) L91 – You did not investigate blue stain fungi in this experiment – I suggest to use blue stain fungi*

*in discussion part.*

We agree, the mention of blue stain fungi is removed in this sentence.

*19) L92 – "tree individuals" – I suggest "two trees"*

Agreed and corrected!

*20) L99 – I suggest that in this sentence indicate table 1 for campaign dates*

At the request of Referee #1 to include a experiment/measurement design table we have now added a new Table 1 including the dates, sites, plots and samples to which we indicate in this sentence instead for the campaign dates.

*21) L96 –Based on this sentence I would expect that in a sub-study where more than one campaign*

*but it's written over time. For me it indicates more than one campaign.*

The sub-study we refer to in this case is indeed more than one campaign. We followed two tree individuals from May to August, from before to after infestation. So the indication of it being more than one campaign is accurate. The sentences in this case does not indicate specifically about the sub-study but about all campaigns conducted (six in total) in which the sub-study was a part. We have revised this part to clarify that the total number of measurement campaigns were six whereof five were in Hyltemossa and one in Norunda.

*22) L101 – use one style: sub-study or additional campaign trough MS*

Based on comment 21 above, we believe there to be a misunderstanding. The sub-study is the study in which we compare two trees with different health status over time before and after infestation. The additional campaign was a campaign carried out in Norunda on top of the campaigns in Hyltemossa, it

was an additional campaign as we did not plan it but acted on the opportunity to travel there to measure more bark beetle infested trees. To build on the reply to comment 21, we have changed the sentence and removed "additional campaign" to avoid confusion.

*23) Figure 1 – how are the rules for figure legends, is there a dot or semicolon after Figure 1? What is*

*the rule for figure legend text - needs to be in bold? I like the map for showing, but describing b-ca is not common in figure legends also in text (like Fig.1b). If possible, change maps etc. starting*

*with abc and in figure legends also describing abc. Missing proper explanations what yellow*

*squares and circles mean. "The Figure..." –why is there Figure not figure? If possible, write out in*

*the legends which site had healthy and infested trees, and where was sub-study done.*

The manuscript was conducted using the Copernicus_Word_template.docx in which the figure caption is in bold and has a semicolon after Figure number. The order of description is now changed to abc in the Figure caption, however we prefer to keep references like "Fig.1b" in text as it brings clarity to what is referred to in the text. We also appreciate that you noticed the lack of explanation in the figure caption, and the use of a capital F in "The Figure", this is not on purpose and have been corrected. Information about where the healthy and infested trees were measured is added as well as information on where the sub-study was conducted.

*24) L122 – "by visual examination" – meaning no entry holes?*

Yes, in the visual examination there would be no entry holes, but we also looked at the general health of the tree. The tree could for example have been stressed due to forest machinery ripping up the roots, some other pests like aphids and adelgids. If we saw signs of this the tree was not selected. This has been clarified and mentioned in the MS: "Healthy trees were selected by visual examination in close contact with the forest manager employed by the forest owner, Gustafsborgs Säteri AB, in May 2019. Trees that could have been stressed by forestry machinery or pests were not selected for the study."

*25) L123 – Indicate site name*

Gustafsborg säteri AB is the company that owns and manages the forest, thank you for pointing out that this information is missing here and may cause confusion. We have changes the sentence to now include specifically that we spoke to an employee at Gustafsborg and that they own the forest (see comment 24).

*26) L120-125 – write out here which was here considered as sub-study*

We have clarified this part as it was not intended to be mixed up with the sub-study but only indicate how the infested trees were selected. The sub-study is described below these lines in a new paragraph. This has also further been clarified to state where the sub-study was conducted and the method for it.

*27) In Methods part, please consider clearly explaining healthy, infested and control, because later you*

*show also control measurements and use the same style trough MS.*

We agree, it is currently inconsistent and it is now corrected in the MS. Control trees and healthy trees are the same and we already decided on using healthy trees only but obviously missed to change this through the MS. All indications of "control trees" have now been changed to "healthy trees".

*28) L130 – check names and ALL NAMES in MS must be correct, many mistakes: alpha-pinene, αpinene, beta-pinene, β-pinene etc and other names written not well.*

Thank you for highlighting this, we agree, consistency is important and this has been corrected for.

*29) L130 – Typosan- company name?*

The company name must have been missed, it is now corrected "(Typosan P306, Plantskydd AB, Ljungbyhed, Sweden)

*30) L132 – repeatedly: measurements dates are indicated where?*

Dates indicated by the new Table 1 (see comment 20) and have now been indicated in MS.

*31) L136 – indicate figure*

Corrected

*32) L137 – use one style trough MS 5°C to 22°C or 5° to 22°C*

This has been corrected to the first (5 °C to 22 °C).

*33) L138 – start with May and then with June*

Agree, this has been corrected.

*34) Figure 2 – in figure legends start describing with temperature as you have on the figure temp,*

*precip, campaign. Figure legend text in bold? I suggest writing out in a better way where data was*

*taken.*

Figure caption has been corrected to follow the structure of the legend. The figure legend text is kept in regular to decrease the number of fonts used in the figure to one according to online instructions (https://www.biogeosciences.net/submission.html#templates). The description of data has been changed to a full sentence.

*35) L147 – pump company? I guess L/min is visually better than lpm. 0.6-0.9 L.*

The pump box was custom made, but agree that the pump and flow meter information needs to be given here.. Corrections have been made accordingly. We decided to keep lpm as this is our preference. Changed liter to L.

*36) L148 – "Ad sorbent tubes…" goes to VOC collecting part*

Corrected, removed this part from instrument description.

*37) L153 - 40°C or 40 °C – use one style trough MS*

Corrected, a space between number and unit is intended as described in online instructions.

*38) L155 – I suggest here to use diameter, not inches; PTFE-tubing or PTFE-lines?*

Changed to metric system, ⌀ 6.35 mm (1/4"). Corrected to PTFE-tubing, not lines.

*39) L158 – "each sample" – do you mean here each VOC collection?*

For clarity, this has been corrected. "Each sample" is changed to "each BVOC collection".

*40) L160 – so the first VOC measurement was at 8:00 in the morning or you started to put the*

*chamber bases on tree?*

Yes, as indicated the sampling started at 8:00 in the morning. The chambers were placed onto the tree trunks prior to this to allow for flushing. This has been clarified by changing "with measurements typically starting..." to "with BVOC collection starting around..".

*41) L166 – Can you add the info when the infestation (swarming time) was detected. "For plot 3"*

*indicate plot name. "...witnessed on site." – when?*

Info on swarming times have be added as this adds important information, thanks for pointing this out. The swarming time was used later to calculate how long the infestation has been ongoing, so we missed to explicitly mention it at this point. Plot 3 is the name of Plot 3, the site is HTM which we have indicated for clarity. The date of the infestation start have been indicated more clearly by referring to the June campaign in the new Table 1 where the date is shown. We have added supplementary information to the swarming period time series to increase transparency on how accurately we were able to estimate the swarming date as we determined this from peaks during swarming periods taken from measurement stations located nearest to our measurement sites. We have also added information about the main swarms affecting our measured trees in the paragraph.

*42) L168 – based on data from Skogsstyrelsen database, when was the proposed swarming time?*

See answer to comment 41.

*43) L170 – Did you mark the entry/exit holes to distinguish new and previous entry/exit holes?*

No, the entry/exit holes were not marked, so it is possible that "old" entry/exit holes where measured, but we still believe it to be the time since the infestation start that matters in our case, as well as the distinguishing between entry and exit holes.

*44) "By looking at bark photographs..." – I suggest to use "photos" and it's the first time to indicate*

*figure 4, so, in the end of sentence, indicate figure, not in the next sentence.*

Thank you for this input, we have corrected accordingly.

*45) L173 – Rewise sentence – I suggest not to start the sentence with "table 1".*

Corrected: "The number of holes counted inside the chamber area is listed in Table 2 along with an extrapolation of the counted holes from the chamber area to square meter bark area."

*46) Figure 3: On figure you have VOC filter, but in Methods part there is no info about VOC filter. Is it*

*the same as nets? "A photograph of how the setup looked in the field is displayed to the right (b)." – I suggest revising. "The BVOC samples were.." - add here the pocket pump also. Add tree name or latin name.*

You are correct, in the Methods we do not mention "VOC filter" specifically but have only failed to be consistent in wording. In the Methods we named it hydrocarbon trap (L149). This has been changed to VOC filter to keep the consistency. In the figure caption, mention of the pocket pump as well as tree name have been added. The text was revised to mention (b) in the beginning: "The experimental schematics (a) and field photo (b) of the tree trunk chamber mounted on a tree...".

*47) Figure 4: which tree – add name or latin name. Correc to " entry and exit holes". Photos where*

Thanks for the clarification, tree name has been added, and corrections on the entry and exit holes. Photos were taken by the first author and as this is not a published photograph but taken for this study we will not include a reference to it. Agree to change "bored" to "drilled" throughout MS to keep the consistency.

*48) Table 1 – text in bold? Which infested tree –add name. Which site/area trees belong? Table paragraph spacing. "exit and entry holes". There is no explanation what S1S1 etc mean? "Type majority"-> "Hole type majority". If possible add spruce initial health status also – healthy, infested etc.*

(Now changed to Table 2). The text is kept in bold as the template has figure text in bold. Name have been added as well as two columns indicating plot and site to make it clearer to where the trees belong. S1S1 etc are the tree IDs which we believe is clear from the sentence "the Norway spruce with the ID…". Type majority was changed to Hole type majority. Thank you for the great suggestion of adding initial health status in the table. However, we have not added this information as we cannot tell this for more than two of the trees, the majority of the trees were measured after they were already infested – thus we had not chance of knowing their initial health status.

*49) L190 – I suggest to add in this paragraph how much healthy and infected tree BVOC measurements were done in total*

Thank you for the suggestion, this had been added.

*50) L191 –Tenax company? Carbograph company?*

The adsorbent tubes were bought packed with those two chemicals in a 2-bed configuration fromMarkes International as one product, so we prefer to give Markes International as reference.

*51) L194 – Why was the collecting time 5 to 6 L not exactly 5 L or 6 L?*

The liters collected varied because of the sample time variation. Because of unforeseen events sampling was varying +/- 5 minutes with the goal of it being 30 minutes. This is however accounted for in the sample analysis which is based on the sample volume and not on the sample duration.

*52) L196 – "twice a day" – can you say at least before mid-day or after mid-day? Or when?*

Clarification has been made "..from air entering the chamber once before the first sample and once after the last sample of the day to capture…"

*53) L197 – "at the start of the BVOC sampling and at the end"-> "at the start and at the end of the BVOC sampling"*

Thank you for the suggestion, we have changed accordingly.

*54) L198 – "After sampling.." - does this sentence repeat the previous sentence? "During sampling period"?*

Thanks for pointing this out, this was misunderstood. To clarify, the sentence before refers to the air temperature inside the chamber which was measured during the sampling period, i.e. during active

sampling. After sampling was done we measured the bark temperature as well. We have clarified stating that the temperature measured inside the chamber was air temperature and that the temperature taken after the sampling was taken after the chamber lid was removed and was taken onto the bark.

*55) L205 – He, which company He?; VOC or BVOC – use one style trough MS*

Company that distributed the helium has been added (Air Liquide Gas AB, Sweden). Changed the MS to use BVOC throughout.

*56) L210 – space between number and m/mm*

Corrected.

*57) L220 – Shimadzu… - country?*

Japan, corrected.

*58) 2.4 – ALL equations need numbers and indicated in the MS where necessary*

Some confusion on this comment. All equations have numbers in parenthesis on the right hand side. We have altered the text in this section to include indications of the equations before it appears like ".. according to Eq. (x):"

*59) L225 – in equation the A is normal, but in describing text it is A (italic)*

Corrected

*60) L243 – For me you have three ways to describe 1) 100 days before/after; 2) majority entry and /or*

*exit holes; 3) early and late season. And all these three are used trough MS. I suggest to explain*

*this well in this paragraph and use one style trough MS and also on figures and figure legend text.*

We agree and have changed to keep it consistent and will further in the MS refer to this as the early and late season: "As the two hole types were consistently occurring before or after 100 days since infestation start, the measurements with mainly entry holes is referred to as the early season and the measurements with mainly exit holes as the late season."

*61) L256 - Q in italic?*

Corrected

*62) L258 – F in italic?*

Corrected

*63) L262 – F and Q in italic?*

Corrected

*64) 269 –Eq 4 – where, which one?*

See line 250 right hand side. This has also now been indicated in the text before Eq. 4 appears.

*65) L270 – Eq 5 – where, which one?*

See line 264 right hand side. This has also now been indicated in the text before Eq. 5 appears.

*66) L270 – I suggest explaining in Methods part: does constitutive mean healthy or control tree*

*emission. And what is considered as control, because later on the figures you have a control.*

Control trees are the same as healthy trees, we have changed to use healthy only throughout the MS, including the figures. Constitutive emissions mean emissions from healthy, unstressed trees as constitutive emissions are what is emitted constantly. For clarity we decided to revise the document and removed the mention of constitutive emissions for our study as it can be replaced with "emissions from healthy bark" interchangeably. The sentence was thus revised to: "an estimation of the total BVOC emission rate from healthy Norway spruce bark and the…"

*67) L274 – 3 m*

Corrected

*68) L275 – North, East? north? east? - Look trough MS that weather arc are correctly. I suggest that*

*weather arcs info goes in BVOC collecting paragraph.*

Corrected. We kept the weather arc info in this section as it applies to the temperature measurements which the calculated emission rates are based on. However, the indication of the orientation in which the samples were taken was also added to the experimental design section for clarity. "The chamber bases were secured in place in the North or East orientation of the trunk every morning."

*69) L276 – I'm not so fond of comparing bark emission with leaf emission. This should be extra study*

*to compare with more measurements campaigns. For me this comparison does not give an extra*

*value to this study (overall). For me the core of the study is bark emissions from infested and*

*healthy trees and then finding important results to discuss on Discussion part and with SOA and*

*climate change and modelling (MEGAN etc).*

See reply under the general comments. While we agree that the focus here is on the bark emissions, we also want to make a point that under infestation the emissions from bark are way larger than emissions from leaves – which normally are dominating the overall emissions from a tree.

*70) L282 – outliers from which sites/area? I suggest moving this paragraph closer to BVOC sampling*

*paragraph. "Photography"->"photos"*

Clarification/corrections has been made. We agree that the outlier part can be moved and have placed it after the description of the lab procedure in the BVOC sampling section.

*71) 2.5 Statistical studies – Needs a lot improving. In here also clarify which measurements, from what*

*area/sites, what about constitutive etc, what about sub-study? Look my comment before. P in*

*italic (p)? Same with F and Q. Clearly write out what did you compare and with what statistical test*

*and which program did you use for it.*

Thanks for this comment as it seems this paragraph was confusing and may have been misleading. However, the requested information is listed. All measurements were included in the statistics, from all sites and plots. Clarifications has been made accordingly: Using a Kruskal-Wallis test (MATLAB R2021a, The MathWorks, Inc., MA, US) with a level of significance set to $p < 0.05$ we compared the

following scenarios: 1) emission rates between the healthy tree plots, plot 1, 2 and 3 in HTM, 2) the difference in emission rates from healthy trees and infested trees, from all plots and sites, 3) the difference in emission rates from the two infested trees in the sub-study and 4) the difference between Q10 and F0 for healthy and infested trees.

We decided to only perform a Kruskal-Wallis test because of our small sample set. Our data was also not normally distributed which further limited our choice in statistics to perform. We decided that in order to test our hypothesis, we simply needed to know if there was a significant difference between the groups or not which we could find our using the Kruskal-Wallis test for our different scenarios.

This has been further elaborated in the statistics section to clearly include what we did and how it fits in our study in relation to our aims and hypotheses.

*72) Should there be dot in each main/second headline, like 3. Results/ 3.1. zyx ?*

According to the online template (referred to above) there is no dot. We have thus not changed this.

*73) 3.1. – Revise the tile if clarifying "constitutive" in previous MS parts*

As mentioned under comment 66 we have revised the document to replace "constitutive" with "healthy". The title is thus revised to: "3.1 Bark BVOC emissions from healthy and infested Norway spruce"

*74) L294 – constitutive=healthy? Is it temperature-standardized or temperature standardized? Use*

*one style.*

See previous replies regarding constitutive emissions, it has been revised to "healthy". The "temperature standardized" has been corrected to use throughout MS.

*75) L296 – indicate the results in table X if they are written out in table.*

Thank you for highlighting this, they are written in the table and this have now been indicated.

*76) L301 - The numbers here and on the figure are not well trackable/comparable and at the same*

*time quickly understood on the figure. If I read the sentence and look at the same time on the*

*figure, cannot distinguish quickly what is what and figure is hard to read for a reader who does not*

*read the results part about this figure.*

The numbers on the second sentence at L301 is referring to the range of the emission rates at each plot (1-3) as visualized by the boxplot (Fig. 5), the medians are referring to the medians as visualized by the boxplot. The figure is describing all the emission rates measured throughout the season for the healthy trees at the different plots, the point of the boxplot is to show that there is no significant difference in emission rates between the plots, which we think is clear from this figure. The numbers appearing first in this paragraph is however the average emission rates for the different sites, which is not visible in the figure. As the main point was to show the difference between the plots (or lack of difference), the averaged numbers have been removed during the revision to avoid confusion as the aim of indicating similar emission rates are still fulfilled without them.

*77) L303 – p-value? p>0.3. Make changes through MS.*

Corrected p-values throughout MS.

*78) L304 – which sites in what area?*

Correction – mistake in MS, sites = plots at the same site in HTM. This has been revised.

*79) Add on Figure 5 HTM, so the reader can understand what area without reading figure-describing*

*text. In describing text, indicate statistical test. What n means? What black circles mean?*

Thank you for the suggestion. The figure caption has been revised to include: a clarification on the parts of the box plot, what statistical test was used and what n means. The figure have been revised to include what site the plots are located in.

*80) L311 – infested trees in which area?*

We indicate all infested trees in all areas. This has been revised to clearly state that: "For the bark beetle infested trees located in both sites (HTM and NOR)"

*81) L312 – drill or bored? "The season…" – you were describing it before in Methods part, in here use*

*one style to indicate, but not repeating the same as in Methods. Look my comment before.*

Thank you for this suggestion. The mention of the early/late season have been removed from this part as we revised the methods section (comment 60) to only use early and late season as an indication of mainly entry holes/earlier than 100 days since infestation. Thus there was no need to mention this again – as you pointed out.

*82) L314 – I suggest to add total emission also to Table 1.*

Thank you for the suggestion, in Table 3 (old Table 2) we have added a row at the end with the total emission (as well as the Table A1).

*83) L315 – "..µg m-2 h-1 . MTs…" check space. " MTs…" indicate for table 2.*

Corrected.

*84) L319 – 42% or 42 %? Use one style thorough MS.*

See replies above. This has been corrected to include a space between number and %.

*85) L322 – Which trees: name? Which area? Name? BVOC or VOC?*

See replies above. All trees are indicated to be from all sites but this will be clarified.

*86) L323 – healthy from which area?*

From all healthy trees, only measured at site HTM, which will be clarified.

*87) L328 – infested tree: I suggest to use infested bark samples, bark emissions, infested tree bark etc*

*through MS, one style through MS.*

We have considered your suggestion but will continue to use the phrase "infested tree". Bark beetles attack the bark, but the entire tree is substantially and most often lethally affected by the infestation.

*88) 329 – "isoprene, (n = 17) compared " check the mistake. Infested trees in which area?*

Typo corrected. Infested trees from all sites. Revised in paragraph.

*89) L331 – indicate table*

Corrected

*90) L337 – Which areas/sites? P value – how come 0.00???? "The median…" - Consider writing this*

*sentence according to this what is shown on figure or change figure according to the sentence. I*

*advise to start with healthy and then infested (early, late). I suggest to use less "respectively"*

*through MS, not so easy to follow.*

All sites. P-value is a typo, corrected to 0.001. The median is shown in Figure 6 as the line in the boxplot. The text have been changed to start with early and late season infested trees and later compared to healthy, as shown in figure. This was the intention with the order in the Figure (6). We have revised to reduce the number of times "respectively" is used.

*91) Figure 6 - please add also HTM or NOR on the figure. Which tree? Again use one style for early/late*

*season, exit/entry. "The emission rate.." sentence doesn't belong to figure describing text. Pvalue??*
*Standardized to temperature – indicate equation in XX part. Explain yellow and blue.*

*Statistical test used?*

See earlier replies. Early/late season is used throughout MS. The figure caption is revised to include tree name, indication of statistical test used and explanation of blue and yellow. Indication of equation is not necessary as this part explicitly mentioning this was determined to not be needed in the caption and was thus removed.

*92) L349 – healthy trees and infested. Which sites/area???*

All plots and sites.

*93) L315 – I suggest rounding the numbers to proper value in table, visually better. Overall, it's kind of*

*disturbing using so much % through MS.*

Your suggestion has been noted and the numbers in the table have been rounded for visual improvement. The numbers in the text have also been altered accordingly. See reply to general comments regarding %.

*94) Check VOC names trough MS.*

See earlier reply.

*95) L287 – Again, using % and even so big numbers….do you use 22678 or 22 678? Which area, which*

*site trees? Which trees?*

See earlier replies regarding %. We will change to the correct thousand separator throughout MS for consistency (10,000).

*96) L374 – when it's the first time indicating Appendix, write out Appendix, otherwise what Table A1*

*means? trans, trans-alloocimene - > maybe (4E,6E)-alloocimene better?*

Thank you for the suggestion. This has been corrected.

*97) L375 – gamma-terpinene or γ-terpinene? "in infested…"- revise end of the sentence, double*

*Repeat*

This has been changed throughout MS, γ-terpinene is used. The sentence has been revised.

*98) Table 2 – paragraph spacing. Which tree? Which area? STDEV is only for groups, not for each compound. Explain MT, SQT. What about decrease (-65%), not explained, only increase. Again numbers: 22678 or 22 678? Check all the names in table, all the names need to be used correctly trough MS (tables, figures, etc). If possible add total emission.*

The table description have been revised. Included now is also a description of the decrease, the location, tree name and an explanation of MT, SQT. STDEV have been added for each compound. And the numbers have been rounded (comment 93) and a thousand separator have been added (comment 95). We have also added a row at the bottom indicating the total emission (as requested in comment 82).

*99) 386 –Which area? Which trees? Number 13000?*

All sites and all infested trees. We noticed a large misinformation in the text, number 13000 refers to the highest temperature standardized emission rate measured from one single tree which can be seen in Figure 7. This is however not the seasonal average for all trees as it seems in the text. This has been revised to indicate the range in seasonal average for the individual trees, which ranged from 500 to 13000 as indicated by Figure 7. Thanks for spotting this!

*100) L390 - "...daily average standardized emission.." – check*

This is what we intended, we have added temperature standardized emission for clarity.

*101) L391 – "The average number of holes per square meter bark area found in this study was based on the values in Table 1." Revise or delete.*

This sentence have been deleted as it did not bring any further clarity or information in this paragraph.

*102) L396 – Whole sentence "For…." – check and correct. Are here the results of sub-study?*

This is not the sub-study, this part is comparing the emission rates as only standardized to temperature with the emission rates when also standardized to bark beetle holes. This was done to exclude any difference in the results simply due to the variation in number of holes (20 holes might indicate higher emission rates than 10 holes). We will clarify this in the revision, thanks for pointing out.

*103) L400 – "The TS emission..." – I guess this sentence is to long for explanation, I suggest to divided in two parts and describing clearly. I guess this kind of description goes to Methods part in Statistics.*

As the continuation of the MS is based on the information stated in this paragraph we have kept it where it is but revised it as an explanation to why we scale the emission rates by number of holes and not only temperature.

*104) Figure 7 – Which trees? What area? S3S2, S3S3, etc meaning? Early/late, 100 days before/after. "The emission rates…" and "This is evident…" sentences do not belong to figure describing text.*

The Figure caption have been revised to now also include a description of the tree ID appearing on the x-axis, location and tree name. The suggested sentences to be removed have been removed and the early/late and 100 days before/after have been revised for better readability.

*105) 3.2 – Did you to statistics to evaluate influence of time?*

See answer to comment 4.

*106) L413 – Which site/area? Indicate table/figure.*

This sentence has been fully revised to avoid confusion. We now start the section by describing what we aimed to do and when we started our first measurements to find the influence of time on emission rates of infested trees.

*107) L415 – Where is 3.2.3? I suggest to put function to Statistics part and indicating on the figure*

*describing text where to find it.*

Thanks for spotting the wrong reference to 3.2.3, this has been revised to the correct one (3.4). The exponential function have been moved and added as "Eq. 6" in the methods section under the statistics part. The function have been indicated in text and figure caption.

*108) L420 – use one style trough MS: 8b and 8d or 8b,d etc.*

This have been revised throughout the MS to have "Fig. 8b,d".

*109) L421 – I guess "slightly" is not correct to use here. "..depending..." revise, double repeat.*

This has been revised accordingly.

*110) L425 – How long does it take from infestation to total damage/death to the tree? Maybe*

*something to discuss in Discussion and some info for introduction part?*

See reply above (comment 16).

*111) L428/429 – indicate figure. Again constitutive/healthy emission? Area?*

Added indication of figure. See earlier replies on constitutive emissions.

*112) L432 – "The tree measured.."- did it have a name SxSx? What area?*

This have been revised to include the tree ID (S1S1) and the area.

*113) L434 – indicate figure*

Figure 8 is indicated.

*114) L435 – Too long sentence. Revise. Make it easier to read. And indicate figure.*

Revised and corrected.

*115) Figure 8 – VOC names!!! Which trees? What area? "..days passed since..". Correclty put abcd to*

*figure describing text, (a) is in a wrong place and (d). Blue is for what? Did you include yellow marked*

*tree also in the fitting? Revise sentence for clearer understanding. What test is used here? In*

*Methods part there is no indication about measurements after 300 days.*

VOC names have been revised in the figure. We have revised the figure caption to include tree name, location, specification of the legend and fitted curve. The caption is revised for increased understanding. As intended in the figure caption "All trees are included in the exponential fitted curve", also the yellow one. No test is used here, we only fitted an exponential curve. We have revised the methods part to include a sentence stating that the measurements occurring > 100 days since infestation had the last measurement 350 days after infestation.

*116) L445 - entry holes and exit holes*

Corrected.

*117) L450 – indicate tableX?*

Corrected.

*118) L452 – check names!*

Corrected.

*119) L453 – 9b,d – look my comments before.*

See reply before.

*120) L455 – indicate table*

Corrected

*121) Figure 9 – check names. Which area. Change the order top of the figure: first are entry holes and then exit holes. Relationship p value? Which test was used? Which trees? Again (a), (b), etc to correct order. All or total BVOC. The "2" on the x-scale title is to big. "High…" sentences does not belong to figure describing. Again: exit/entry, late/early, 100 days before/after. "There is a disctinion.." – revise.*

See previous replies for Figure 8. The figure caption has been revised as Figure 8 and also removed the mention of "100 days, late/early" and only focusing on the bark beetle hole type as this is the main focus of the figure. As mentioned previously, no p-value relationship is present as we did not do any statistical test on this data and no exponential curve is fitted as there was no good relationship.

*122) 3.4 - sub-study?*

This is indeed the section where the results from the sub-study are presented. This has been indicated in the first sentence in this section "As a part of the sub-study…".

*123) L466 – Revise sentence "At plot 3…" and indicate figure. On another way L466-L469 is describing what you said before, I suggest revise this sentence. Which area?*

This whole section has been revised, area and tree name is included and the repeating of description is removed.

*124) L470- P in italic.*

Corrected.

*125) L471 - Control trees – what does the control trees mean in MS? Control - not at all infested*

*during campaign or beginning of campaign not infested and later experiencing bark beetle*

*infestation? Indicate figure 8 correctly.*

See previous replies. Control trees = healthy trees, this have been revised. Figure 10 indicated.

*126) L473 –Which other trees? Other trees in plot 3 or somewhere else? Indicate a tableX?*

Other trees in plot 3, the sentence have been revised as it led to confusion. There is no table to indicate as we decided to only present this as the figure for better visualization.

*127) L576 –" a difference.." indicate table if possible*

Sentence have been revised. See reply above regarding table.

*128) L477 – check names; I suggest to remove 800 ug or use another number for comparing.*

The names are checked and corrected and the mention of 800 ug have been removed.

*129) L479 - check names*

Corrected.

*130) L480 – indicate figure, I suggest that p<0,03. Delete "and close to zero", check names*

Thanks for the suggestion. This has been revised accordingly.

*131) L483 – p italic*

Corrected.

*132) L484 – indicate figure*

No figure to indicate as we could not quantify the compound verbenone. We have however revised the sentence to indicate that this compound was not quantified and refer to "Table 3" where this is seen.

*133) L485-495 – Again less % or even when presenting choose important results, because this part is*

*interesting. I suggest this part to discuss in Discussion part. Check names. "..by mass" – what mass?*

As previously mentioned, we need to use (%) in order to present the BVOCs which we could not quantify, otherwise they would only be in the list of identified compounds but no further information. We think the change in compound blend is interesting and did not want to exclude the compounds which we could not quantify in this part, which is why we present them all as (%). For clarity in why we use (%) we have added a sentence about this in the Methods part where we describe the quantification: "For other BVOCs which could not be quantified during the study, the amount of the compound present on the sample was calculated as a percentage of the total amount on the sample.". We have revised the sentence to remove ".. by mass" as this seemed confusing and instead replaced it with "where the percentage represents the amount of the compound found in the sample relative to the total.". The results were narrowed down to what we thought were the most important compounds found, but a more elaborated list of compounds found is shown in the appendix (Fig A1). We will clarify this point and explicitly refer to this figure in the revision, thanks for pointing this out. The ongoing infestation is discussed in 4.1.1. which have been made more clear during the revision. Thanks for pointing this out.

*134) Figure 10 - Months names ith capital letter. On the figure, where is asterix-what does it indicate?*

*Can't be seen. Check names! All figure plot need to have month names. Describing text what trees, what area. Why no SE? Add tick marks on the figure, other figures have. Remove the outer box from top of the figure, where are indicated control, S3 and S2 and add some space between them for clearer separation. Start with control in describing text, then healthy or infested etc. Add year in describing text. "For", I guess "from". In this figure, there is important info, but this does not come forward clearly because of figure negative sides. Maybe another alternative?*

See replies for other figure comments. Months have been capitalized. The asterix have been removed. Month names have been moved as axis labels on the right to avoid repetition in the plot. SE is not included as the average is an average over the day, 3 measurements, we have decided not to add it to this figure as we think it does not add any value other than cluttering the figure. We have added more space between labels in order to increase the readability and clarity of the figure which we think is also supported by the legend box that we have applied constantly throughout the figures in this MS, and we have this not removed the "box". Tick marks have been added. The control (changed to healthy) is shown in green, S3S3 is orange and S3S2 is blue in the figure. We hope the figure is more clear now. Revised caption text is inserted below where bold letters indicate changes:

"Figure 10: The average **temperature** standardized **BVOC** emission rates for all compounds **from Norway spruce at plot 3 in Hyltemossa: healthy trees (green), infested spruce with ID S3S2 (blue) and infested spruce with ID S3S3 (orange). Measurements were taken in 2019 during** (a,b) May, (c,d) June, (e,f) July and (g,h) August. The graphs are horizontally separated for visibility due to large differences in scale. The **healthy** trees are included in all graphs but the emission rates are not visible on the same scale as the infested trees in (c,d) June or (e,f) July. The bark beetle infestation had not started in (a,b) May, however, the spruce S3S2 was already subjected to stress from late bark beetle attacks previous season before the bark beetle infestation started again in (c,d) June, leading to higher emission rates in May."

*135) L506 – F in italic? Q in italic? Look through MS*

Corrected

*136) L507 – which area?*

This has been revised to indicate trees from both sites.

*137) L508 – indicate a table*

Table A2 is now indicated here as well.

*138) L510-511 – double repeat*

Revised.

*139) L513 – 34900*

Changed to 34,900 µg m-2 h-1.

*140) L510-520 – too much %*

See replies on general comments; but we will reduce those high %-ages and refer to x-fold increase instead.

*141) L521- coefficient?*

This is meant to be the Q10 coefficient. Corrected.

*142) L526 – p<0.05 and p<0.01*

Corrected but kept P<0.03 rather than p<0.05 to keep the consistency in the MS.

*143) 3.6 – As is said before: for me leaf emission does not give an extra value for the study. Maybe to leave this part out and present the result what are in this chapter? Again which area?*

See replies to general comments. The short paragraph on the comparison with leaf-scale emissions in our view adds valuable information and helps to assess the importance of the induced emissions as normally only leaf-emissions are considered as for healthy trees these are dominant. We have decided to keep this section, but changed "leaf" emissions to "needle" emissions throughout MS. Area has been added.

*144) L537 – Ms or Ms?*

Corrected.

*145) L558 – well?*

We will revise this sentence as "**Emission rates from the bark of the infested trees were at least around 55 times higher than the total  MT emission rate from both leaves and bark of a healthy tree.**"

*146) Figure 11 – pleases start with abc, revise figure then. Which tree, which area? Again: north, east, North, East? Meters -> m. Which ONE infested tree (SxSz?), which area? What green and purple shades are representing? Add tick marks on y-scale.*

The figure caption have been revised to change the appearing order to "a,b,c". A clearer description of the figure parts is included as well as area and tree name.

*147) 4. Discussion – needs a lot improving. Right now it's weak. Indicate table and figures where proper. Check names. What areas? More references and discussions. Somehow Discussion part repeats Results part, would be better to have more discussion about results.*

Thank you for your input. Tables and figures have been indicated. Names have been checked. Sites have been clarified. The number of references used in our Discussion is low because of the low number of relevant studies to compare with. We decided to deliberately use only studies where a comparison will be relevant between Norway spruce bark BVOC emissions and stresses, particularly insect stress from the European spruce bark beetle. As the number of comparable studies remains low, we determined to keep our discussion very open as we thought it to be hard to draw any firm conclusions from our findings due to the limited dataset. We have revised the discussion to improve it according to the following comments.

*148) L587 –what relationship? How did you calculate this? Assessing how? Based on what? No info in Methods. Any references about lichen cover related to isoprene emissions. Ghimire – how did they assess? Statistically or also visually?*

The relationship we are referring to would be a correlation, either positive or negative. The reference Zhang-Turpeinen et al., 2021 found a positive correlation with isoprene emission and lichen cover, this has now been clarified in the MS. We did, as Ghimire, asses the lichen cover visually from our bark photos and this has now been included in Methods around L170 where we describe looking at bark photos for checking the hole types. Ghimire tested the correlation using the Spearman's correlation analysis, which we did not do, we only assessed this visually by comparing the images with the emission rates of isoprene – where there were no clear indication of anything, there would be high isoprene emissions and no lichen cover at all, which made us determine not to move forward with that. It is included in the discussion only because we thought it interesting that there would be isoprene emitted from bark with no clear explanation why. This has now been clarified.

*149) L591 – healthy and infested trees – what previous findings? References? Some examples for describing? Indicate figure.*

This section have been revised to indicate more clearly what the previous findings have found with added examples and references. Instead of only focusing on spruce and spruce bark beetles, we have also added one new reference (Heijari et al., 2011) to compare our results to looking at pine and weevils. The statement in the discussion remains the same however, our findings still indicate much greater increase in emission rates than the other studies. Figure 6 was indicated to visually show the higher emission rates from infested trees compared to healthy.

*150) L591 – repeating of the results. I suggest to put these number into comparison with some references if possible or to discuss about emissions. I suggest not to use "we".*

Thanks for pointing this out. We have revised this section and removed the repeating of results and only focusing on the increase which we have put in relation to other studies. The motivation for removing the numbers and only having the increase is the difficulty in comparing numbers, all studies have different ways of describing emissions/concentrations – thus is it easier to compare how much the emission rates increased as this is unaffected by different units. We have removed the use of we/our when suitable throughout the MS.

*151) L549 - And so what it is higher at the beginning? Connect the higher emission with atmospheric processes, add references. What is different in the beginning? Indicate your figure/table if proper.*

This part "higher earlier and decrease with time" have been removed as it caused some confusion. In this part we do not aspire to connect the emissions to something else but want to compare it to what others find. Comparison with atmospheric processes, SOA etc is discussed later, as well as what is different in the beginning (there are more entry holes in the beginning and the spruce defense is more active, as time passes the vitality of the spruce decrease as does the emission rates).

*152) L583 – 605: all compared to one reference.*

As replied in comment 149 we have elaborated the beginning of this section to include more comparisons with other results. However, the continuation where we are trying to find a reason for the big difference in emission rates are targeted towards the one reference (Ghimire et al., 2016) in which they measured the bark emission rates from Norway spruce from healthy trees and infested trees. The difference in emission rates compared to other studies is not as apparent as they are not looking at the same tree species or the same bark beetle. There are of course some other studies where they look at fungal inoculation and European spruce bark beetles on Norway spruce, which is something we did not do (Mageroy et al., 2020; Zhao et al., 2010, 2011b). Thus we will leave the

discussion of the potential difference in emission rates with this one reference, but have revised the manuscript to make this more clear.

*153) L599 - time...what about temperatures, weather conditions? And what time, measuring time*

*during day (morning, lunch time, evening) or campaign times (certain dates).*

Temperatures should not be an issue as they also used standardized temperatures in their study, and using monthly means for their results. The values are comparable in the manuscript as well. We do believe that the biggest indicator is the time which is supported by our findings that the emission rates decrease exponentially with time. Measurements done 1 week after infestation will thus have substantially higher emission rates compared to measurements done 4 weeks after infestation. This section have been revised to make it clear that the temperatures should not be an influencing factor.

*154) L600 – "study…" indicate figure*

Corrected.

*155) L601 – Ghimire – which holes in this reference?*

All holes supposedly, they did not specify entry or exit holes from inside their chamber area. A sentence stating this have been added for clarification.

*156) L602 – "this allowed…" then what is done before? References? What stage to what stage?*

We could not find a single study measuring a spruce before infestation and following it through infestation, thus relation the emission rates to the stages of infestation. What has been done before is to measure healthy trees, travel to another site at another time to find bark beetle infested trees, and so on. We have added an older study in which they measured the volatiles emitted from entry holes during the first week but not longer, the same reference used further down to discuss the difference in emission rates from entry and exit holes (Birgersson and Bergström, 1989). For clarity large parts of this section was revised and removed.

*157) L604 – "This finding.." – what finding exactly? Why it is important in bigger picture?*

Corrected – not referring to a specific finding, rather the design of the study. The sentence have been revised to include why this is important "to get a correct estimation of the spruce bark beetle impact". This part has also been moved further down in discussion.

*158) L605 – "We could see..."- is it really a trend or rather something normal to be expected when*

*more time passes since infestation.*

As we now mention this part earlier in this section as a support to why time might have played an important role in the different emission rates measured in our study compared to Ghimire, this has been removed.

*159) L608 – check space*

Corrected (removed this part).

*160) L609 – references, more to discuss with few sentences?*

This is a statement that the trend we saw in our data is only related to the start of the infestation, not taking into account the time of the season and in what swarming period the infestation started, while for e.g. leaf emissions seasonality and phenology is known to be important. It would be interesting to

discuss the potential influence of the swarming periods and time in season. However, finding data for this to discuss around might be difficult as there are only very few studies available, and this rather would be speculation. We have thus removed this part from this section as this is not the main focus of our study.

*161) L611 – indicate figure*

Corrected.

*162) L612 – which total holes, majority of entry or exit holes? Or entry+exit? To discuss what could be*

*the possibilities for no pattern. Any references where this kind of pattern occurs?*

Yes, total number of holes refers to all holes, entry+exit. We did not see any pattern in emission rates and the entry+exit holes until we separated them. This is discussed further in this section. We have not seen any studies done comparing number/type of holes and emission rates. We first expected a positive correlation between increasing number of holes and higher emission rates, but as mentioned we did not see that pattern. Also to further stress this, we did not have a generous amount of data on the infested trees, which made it hard to make any conclusions. This section have been revised for better readability and clarity.

*163) L614 – indicate figure*

Corrected.

*164) L616-617 – references. Too long sentence, I suggest to make into two sentences. Double repeat.*

Sentences have been revised to make it shorted and clearer. No references was added as this is results from our own study, which we have made clearer by indicating figures and tables. We also believe there to be a general consensus that entry holes appear at the start of an infestation and exit holes during the later stages and have thus not included a reference for this.

*165) L618 –VOC`*

Corrected.

*166) L619 – which individual tree, which area?*

Tree ID have been added and area. Table 2 is also indicated as this is where this info can be found.

*167) L620-629 – References.*

We have revised large parts of this section. We did not find any comparable studies to this part and have revised it to be written clearly as speculations of our own results. We have added a reference looking at resin decay over time in which they also found an exponential decay (Eller et al., 2013) and used as a comparison to our findings of exponential decay – motivating our claims that it is more due to bark beetles than simply resin exposure, which is further motivated by our results that the bark beetle hole type matter for the emission rates (entry holes have higher emissions than exit holes).

*168) L624 – All trees – what area?*

This part have been revised according to comment 167.

*169) L628 – any reference to support your results and conclusion?*

See reply above (comment 167).

*170) L631 – indicate table/figure if proper*

Table 3 have been indicated, thank you for spotting this.

*171) L630-639 – add references and something to discuss more than these two references*

We have revised this section for readability and connections to hole type and time. However, we only found one study looking specifically at BVOCs emitted from entry holes on Norway spruce which is the one we compare with in our discussion. We have however revised it to also include mentions of the blue stain fungi and the potential influence on the emission rates supported by (Mageroy et al., 2020)

*172) L635 – oxygenated monoterpenes?*

Yes, corrected.

*173) L639 – any references to add?*

Added: (Birgersson et al., 1988; Zhao et al., 2011a)

*174) L640 – 4.1.1.- Only 3 references per this part?*

We have revised this section by creating new angles to our data where we compare the stressed tree defenses to what have been tested in experimental studies (Mageroy et al., 2020; Zhao et al., 2011a) and also making it clearer that this is speculations and descriptions of the conclusions we can draw from the data we have collected in this study. As previous replies stated there is not many studies to compare with our specific findings, and we hope that it is now more clear that we are discussing around our data.

*175) L682 – what other 16 BVOCs? Revise, no need to add BVOC names exactly.*

We have revised this to not mention the 16 bvocs but only state that the SQTs …. Have been found to have the highest SOA yield.

*176) L700- "As the…" double repeat.*

Corrected.

*177) 684 and 690 – both sentences are a repeat. " This could.." and "This change.."*

Revised.

*178) L685 – Myrcene sentence goes rather to Results part.*

This have been revised and removed.

*179) 4.2 – again % and numbers. Now here Ips typographnus – why not bark beetle as trough MS?*

*Only 5-6 references per this part?*

This have been revised to reduce the use of %. However it could not be reduced completely as the referred review is presenting the increased emission rates as %. We have revised to spruce bark beetle for consistency. The same motivation for not using a high number of references as presented before is still valid in this case. We have however revised this section to make it clear that we are making speculations and discussing our own data.

*180) Table A1 – Names: numbers; MT, SQT explanation – check comment for table 1.*

This have been revised and corrected.

**References**

Birgersson, G. and Bergström, G.: Volatiles released from individual spruce bark beetle entrance holes Quantitative variations during the first week of attack, J. Chem. Ecol., 15(10), 2465–2483, doi:10.1007/BF01020377, 1989.

Birgersson, G., Schlyter, F., Bergström, G. and Föfqvist, J.: Individual Variation In Aggregation Phermomone Content Of The Bark Beetlem Ips typographus, J. Chem. Ecol., 14(9), 1988.

Celedon, J. M. and Bohlmann, J.: Oleoresin defenses in conifers: chemical diversity, terpene synthases and limitations of oleoresin defense under climate change, New Phytol., 224(4), 1444–1463, doi:10.1111/NPH.15984, 2019.

Eller, A. S. D., Harley, P. and Monson, R. K.: Potential contribution of exposed resin to ecosystem emissions of monoterpenes, Atmos. Environ., 77, 440–444, doi:10.1016/j.atmosenv.2013.05.028, 2013.

Ghimire, R. P., Kivimäenpää, M., Blomqvist, M., Holopainen, T., Lyytikäinen-Saarenmaa, P. and Holopainen, J. K.: Effect of bark beetle (Ips typographus L.) attack on bark VOC emissions of Norway spruce (Picea abies Karst.) trees, Atmos. Environ., 126, 145–152, doi:10.1016/j.atmosenv.2015.11.049, 2016.

Heijari, J., Blande, J. D. and Holopainen, J. K.: Feeding of large pine weevil on Scots pine stem triggers localised bark and systemic shoot emission of volatile organic compounds, Environ. Exp. Bot., 71(3), 390–398, doi:10.1016/j.envexpbot.2011.02.008, 2011.

Krokene, P.: Conifer Defense and Resistance to Bark Beetles, Bark Beetles Biol. Ecol. Nativ. Invasive Species, 177–207, doi:10.1016/B978-0-12-417156-5.00005-8, 2015.

Kulmala, M., Suni, T., Lehtinen, K. E. J., Dal Maso, M., Boy, M., Reissell, A., Rannik, Ü., Aalto, P., Keronen, P., Hakola, H., Bäck, J., Hoffmann, T., Vesala, T. and Hari, P.: A new feedback mechanism linking forests, aerosols, and climate, Atmos. Chem. Phys. Discuss., 3(6), 6093–6107, doi:10.5194/acpd-3-6093-2003, 2003.

Mageroy, M. H., Christiansen, E., Långström, B., Borg-Karlson, A. K., Solheim, H., Björklund, N., Zhao, T., Schmidt, A., Fossdal, C. G. and Krokene, P.: Priming of inducible defenses protects Norway spruce against tree-killing bark beetles, Plant Cell Environ., 43(2), 420–430, doi:10.1111/pce.13661, 2020.

Zhao, T., Krokene, P., Björklund, N., Lngström, B., Solheim, H., Christiansen, E. and Borg-Karlson, A. K.: The influence of Ceratocystis polonica inoculation and methyl jasmonate application on terpene chemistry of Norway spruce, Picea abies, Phytochemistry, 71(11–12), 1332–1341, doi:10.1016/j.phytochem.2010.05.017, 2010.

Zhao, T., Borg-Karlson, A. K., Erbilgin, N. and Krokene, P.: Host resistance elicited by methyl jasmonate reduces emission of aggregation pheromones by the spruce bark beetle, Ips typographus, Oecologia, 167(3), 691–699, doi:10.1007/s00442-011-2017-x, 2011a.

Zhao, T., Krokene, P., Hu, J., Christiansen, E., Björklund, N., Långström, B., Solheim, H. and Borg-Karlson, A. K.: Induced terpene accumulation in Norway spruce inhibits bark beetle colonization in a dose-dependent manner, PLoS One, 6(10), doi:10.1371/journal.pone.0026649, 2011b.

---

## Author Response (AR2)

**Response to the comment of Anonymous Referee #1**

We appreciate the time and effort from Referee #1 to provide comments and great suggestions on our paper. We have divided the comments into points and address each point below, where the reviewer's comments are shown in italics. The line numbers refer to the revised document.

*Comments*

*1) The writing needs to be improved for grammar, word usage and flow.*

Thanks for this comment. We have now tried to make the text more readable with a focus on improving the flow. We have specifically focused mainly on revising the text in the abstract, last three paragraphs in the introduction, the results paragraphs starting from line 392 and 413 and the discussion paragraphs staring from line 645, 669, 711, 733, 747 and 760. We have also re-structured the methods sections to make it easier to follow (see point 2). We hope it is more clear now. We have also run the manuscript by a native english speaker to check for grammar and word usage throughout the manuscript.

*2) I still find it difficult to follow the methods.*

We regret that the methods description obviously still was not as clear as we had hoped. To make the methods clearer we have re-structured the site description to clearly describe the location and vegetation description of the sites HTM and NOR followed by a description of the plots on each site with a more thorough explanation of the number of trees selected for the study (from line 110). We have also improved Table 1 by adding a row clearly stating the number of trees measured at each plot as an addition to the number of samples taken of these trees. We have also added a final row indicating how many samples were taken in total from healthy (n = 144) and infested (n = 40) trees and the total samples (n = 184). Table 1 have also been indicated more clearly throughout the site description to avoid any confusion.

We have also improved the paragraph from line 140 where we describe where and when we installed a bark beetle trap to make it clearer why this was done and that it only resulted in two infested trees at this plot.

The text in section 2.2 was improved for clarity and flow to make it easier to follow the experimental design of the study.

In order to improve the transparency and clarity we have also added information about the outliers removed (paragraph starting at line 251) and that it resulted in 33 samples which could not be used reducing the samples in the analysis from 184 to 151 (113 samples from healthy trees and 38 from infested trees).

*3) The reference list is not complete (e.g. Ortega & Helmig, 2008 is not included)*

Thanks for spotting this! We are sorry for having missed the Ortega & Helmig (2008) reference, which is now added. We have also double-checked the reference list once again.

*4) I think the sub-study with two trees should be removed from the manuscript as it is not possible to statistically compare two individuals.*

We thank the reviewer for enlightening us about this mistake in the manuscript. The intention was never to statistically compare two individuals but rather their individual sample pool and progress over time. However, we realize that the end product was indeed a statistical comparison of only two

individuals to which we agree is not possible. We do however not agree with the reviewer that this section should be removed. What we indented was to have those two individual trees as a case-study as those trees were infested by the bark beetles during the experiment. This means that we have measurements of the trees at either a completely healthy stage or a lowered defense stage and also measurements of the same tree individuals when they were infested. This was something we had hoped for when we had placed out the pheromone trap. To our knowledge, this is the only reported study of this kind, other studies analyzing Norway spruce BVOC emission rates from bark beetle infested trees are only comparing healthy and infested trees and not investigating the progress from healthy to infested from the same individuals in-situ. That we only got measurements from two individuals is unfortunate but nothing we could control for as we could not decide where the bark beetles should go.

We believe these results to be important for the knowledge of bark beetle impacts and rather than removing this section we have made major alterations. We acknowledge to have used a misleading expression here when phrasing this section as a 'sub-study' and have thus changed it throughout the manuscript to be phrased as a case-study clearly stating the use of only two individuals firstly mentioned in the methods (line 144). As this is nothing we can statistically compare we have aimed at presenting it as preliminary results and have thus removed the hypothesis around this in the introduction. All statistics have been removed from this section (3.4) and the text is now altered to present the results for the two trees in relation to each other but not drawing any major conclusions.

The discussion section around this case-study (4.1.1) have also been majorly revised to clearly state that this is a case study presenting initial results from when a tree goes from healthy/decreased defense to infested by spruce bark beetle and the progress over time. We also highlight the importance of follow-up studies on this in the future, which there will be none if no results are published despite small sample size. We have also substantially reduced the amount of discussion dedicated to this section as an indication of the small but important contribution is has to the study as a whole.

We are not drawing any major significant statistical conclusions from this section, but we want to highlight the promising initial results we found that deserves to be shared.

*5) The use of the word "campaign" is not correct. This needs to be changes in all instances.*

We have removed to word 'campaign' throughout the manuscript and replaced it with other, more appropriate terms.

*6) The paragraph beginning after line 55 needs more connection between the thoughts.*

Thanks for these suggestions! We have rephrased the section after line 55 (line 64 in the revised MS) to better guide the reader and to improve the clarity of the text.

*7) In the concluding paragraph of the introduction, I think it would be better to list all four aims in one sentence. It is confusing that aim (IV) is at the end of the paragraph.*

Thanks for this comment! We agree that the research aims and research questions should not be spread out in the text. Following the removal of the 'sub-study' raised earlier, we have also revised the aims (i-iii) and hypothesis (H1-H3) of the study to now cover one compact section (L85-L95).

*8) "Four Norway spruce trees were selected at each plot in HTM, and three trees at each plot in NOR. A total of 18 trees were measured, whereof 12 were measured repeatedly during the growing season in HTM.» I can't figure out how these statements fit with Tables 1 and 2.*

We are sorry for this confusion and would also like to thank the reviewer for highlighting this as it led to the discovery of a mistake in the manuscript. A typo had led to one tree being missed in this statement, the correct amount of total trees measured is 19 and this have now been corrected throughout the MS. It is correct that in total 19 trees were used in the study, growing in two different forests in Sweden. All trees were measured repeatedly (typically 3 times per sample day), and trees at HTM were measured repeatedly throughout the season, while all trees at NOR were only measured once as the forests are ca 700km apart. We have added information on number of trees now to Table 1 explicitly (see answer to comment 2). Table 2 however is referring to the number of bark beetle holes per tree, and thus only contains infested trees.

*9) How many total trees were included in the experiment?*

A total of 19 trees were used in this study, growing in two different forests. We have added this information more clearly to Table 1.

*10) From table 1 it seems that there were 31 infected trees, but table 2 indicates only 14.*

No, this is a misunderstanding. 19 individual trees have been used, of which 9 were infested or got infested during the study. Table 1 now lists the number of trees, and the number of samples for those trees. Table 2 provides the number of bark beetle holes for each individual tree that was infested during each field visit. This is clear from the tree ID which appears multiple times for some trees which were measured several times as indicated by the date. The total number of infested trees still remains 9.

*11) Were repeated measures performed only on some trees?*

All trees were measured repeatedly (typically 3 times per sample day), and trees at HTM were also measured repeatedly throughout the season, while all trees at NOR were only measured once as the forests are ca 700km apart. We have added information on number of trees now to Table 1 explicitly, see also line 110-116. In total 184 samples were taken on the 19 trees of the study, 144 from healthy and 40 from infested trees (line 219 to 220).

*12) Why were measurements at NOR only performed once and included only infected trees?*

Unfortunately this information seems to have gotten lost during the revision. Normal forest practice is to remove all bark beetle infested trees as quickly as possible to prevent spreading of bark beetles in the stand. At NOR we had the opportunity to increase the number of samples on infested trees when an outbreak was occurring there. Because of limitations in time and resources only infested trees were measured, as the forest is located ca 700km away from HTM.

*13) Are the tree genetics between these two sites the same?*

*14) How could genetics effect the BVOC results?*

Reply to both 13) and 14):

No, both forests were commercial plantations, and they were different in age and probably genetics. We have done pre-studies on (needle) emissions from both stands (van Meeningen et al., https://doi.org/10.1016/j.atmosenv.2017.09.045) which resulted in similar emission patterns between trees from both stands. This can be compared with a study from Bäck et al. 2012 which analysed the chemodiversity between 40 conifers in a single stand in Finland, so there is always some kind of variation in a forest stand. However, these variations caused by genetics in healthy

trees are orders of magnitude lower than the effect of bark beetle induced emissions. We have revised the discussion to add a paragraph mentioning this (line 637 to 643).

*15) Can you be sure that entry and exit holes were correctly designated without excavating the bark?*

Excavating the bark was impossible as this would have been a destructive sampling that was not possible to do as the aim was to repeatedly measure on the same trees as we were interested in the temporal development of emissions. As the drilling of entry holes is related to the swarming of bark beetles, new holes in the early season are entry holes. A couple of weeks later, after successful placement of eggs in the galleries, new formed holes are exit holes. So there exists a clear temporal pattern that, besides shape and structure of the holes, help identifying hole type. However, for a given (chamber) area it will be impossible to find a spot were only exit holes exist, so we did characterize after the majority of holes for the analysis, which is stated for example in Table 2 as well.

*16) Were any post-hoc test performed after the Kruskall-Wallis test?*

We did not perform any post-hoc test after the Kruskall-Wallis test as we never needed to do so. To our knowledge post-hoc tests are used for comparing several groups, so when for example an ANOVA was made with significant results – a post-hoc is made to see what groups had a significant difference. For this you need at least three groups. We never compare more than two: healthy or infested. In one case we do actually compare three groups – the different sites for healthy trees. But as that resulted in a not significant result (which was expected) there is no use for a post-hoc.

*17) Figure 6 I think early and late BVOCs in infected trees should be compared to early and late BVOCs in healthy trees rather than the full season.*

Thanks for this idea, which we have implemented in this revision (Fig 6)! We previously used the full season emission rate for the healthy trees because the differences between early and late season for the healthy trees is much smaller than the difference between healthy and infested during any season and the difference for early and late emissions from healthy trees are not statistically significant. But we agree that a more consistent way of comparison is also providing seasonal differences for the healthy trees even if it does not in any way change what we intended to show with this figure.